# Targeting alveolar epithelial cells with lipid micelle-encapsulated necroptosis inhibitors to alleviate acute lung injury
Zhi-ying Kang[1,3], Nan-xia Xuan[1,3], Qi-chao Zhou[1], Qian-yu Huang[1], Meng-jia Yu[1], Gen-sheng Zhang [1], Wei Cui[1], Zhao-cai Zhang [1] ✉, Yang Du [2] ✉ & Bao-ping Tian [1] ✉

Acute lung injury (ALI) or its more severe form, acute respiratory distress syndrome (ARDS), represents a critical condition characterized by extensive inflammation within the airways. Necroptosis, a form of cell death, has been implicated in the pathogenesis of various inflammatory diseases. However, the precise characteristics and mechanisms of necroptosis in ARDS remain unclear. Thus, our study seeks to elucidate the specific alterations and regulatory factors associated with necroptosis in ARDS and to identify potential therapeutic targets for the disease. We discovered that necroptosis mediates the progression of ALI through the activation and formation of the RIPK1/RIPK3/MLKL complex. Moreover, we substantiated the involvement of both MYD88 and TRIF in the activation of the TLR4 signaling pathway in ALI. Furthermore, we have developed a lipid micelle-encapsulated drug targeting MLKL in alveolar type II epithelial cells and successfully applied it to treat ALI in mice. This targeted nanoparticle selectively inhibited necroptosis, thereby mitigating epithelial cell damage and reducing inflammatory injury. Our study delves into the specific mechanisms of necroptosis in ALI and proposes novel targeted therapeutic agents, presenting innovative strategies for the management of ARDS.

Acute lung injury is a clinical disease with high morbidity and mortality, induced by multiple factors. Currently, the treatment strategies for ALI/ARDS in clinical practice mainly involve conservative supportive therapies such as low tidal volume positive pressure ventilation, prone position ventilation, extracorporeal membrane oxygenation, and restrictive fluid management[1–5]. Regarding pharmacological treatments, there are currently no specific drugs for ARDS. Dysregulation of the inflammatory response is a key factor in its development, with excessive inflammation leading to diffuse damage to the alveolar-capillary barrier[6]. Alveolar epithelial cell is a crucial component of the alveolar-capillary barrier, playing key roles in both barrier function and immune regulation. Epithelial cell damage and dysfunction are not only key features of alveolar injury in ARDS, but also play an important role in the development and progression of ARDS[7]. The disruption of the alveolar epithelial barrier leads to increased permeability, inflammatory cell infiltration, and pulmonary edema, ultimately resulting in impaired gas exchange and severe hypoxemia. Moreover, the release of pro-inflammatory mediators and cytokines by damaged epithelial cells can further exacerbate the inflammatory response and contribute to the pathogenesis of ARDS[2]. Therefore, strategies aimed at reducing the inflammatory response while simultaneously protecting and repairing the alveolar epithelium represent promising therapeutic approaches.

Necroptosis is in a special position among all cell death processes, it has been shown to be a type of regulated cell death (RCD), but exhibits non-RCD similar morphological features, triggers an inflammatory response in the body and is part of autoimmune or inflammatory diseases[8]. It is now well established that in LPS-induced ARDS, the inhibition of Caspase-8 and the activation of TLR4, MYD88, TRIF, RIPK1, RIPK3, and MLKL play significant roles at various levels. Among these, the formation of complexes between RIPK1, RIPK3, and MLKL, along with the membrane translocation of MLKL, represents the most critical steps in this process[9–16]. Notable advances on the role of necroptosis in disease progression and corresponding interventions have been made in the tumor, skin and cardiovascular diseases[9,11,17–25].

Activation of the necroptosis pathway during ARDS leads to the release of pro-inflammatory signals, which can contribute to the overall inflammatory response, amplifying the inflammatory cascade and exacerbating the condition[26–28]. However, despite these insights, targeted therapies focusing on necroptosis as a therapeutic approach for ARDS remain limited. Further research is needed to explore and develop strategies aimed at

[1]Department of Critical Care Medicine, The Second Affiliated Hospital, Zhejiang University School of Medicine, Hangzhou, Zhejiang, 310009, China. [2]Department of Hepatobiliary and Pancreatic Surgery, The Second Affiliated Hospital, Zhejiang University School of Medicine, Hangzhou, Zhejiang, 310009, China. [3]These authors contributed equally: Zhi-ying Kang, Nan-xia Xuan. ✉e-mail: 2313003@zju.edu.cn; yangdu@zju.edu.cn; TianBP@zju.edu.cn

inhibiting necroptosis to mitigate inflammation and improve clinical outcomes in ARDS patients.

In this study, we propose that necroptosis plays a key role in acute lung injury, characterized by the phosphorylation and complex formation of RIPK1, RIPK3, and MLKL. Additionally, the activation of TLR4 downstream signaling pathways, including TRIF and MYD88, is implicated. To target necroptosis as a therapeutic approach for acute lung injury, we have developed a lipid micelle-encapsulated MLKL inhibitor, GW806742X, which effectively alleviates airway inflammation and injury by selectively binding to alveolar type II epithelial cells. Our research reveals new pathological mechanisms underlying acute respiratory distress syndrome and paves the way for innovative targeted treatment strategies.

## Results

### Transcriptomic analysis revealed the activation of the necroptosis pathway in acute lung injury

To investigate whether the necroptosis pathway is activated in the pre-ARDS phase, we administered airway drops to mice using Lipopolysaccharides (LPS) to simulate the acute lung injury that occurs during the early phase of ARDS in the clinic. Lung tissue was taken for RNA-sequencing, western blotting and tissue immunofluorescence. Upon visualisation of the sequencing results, genes associated with necroptosis, *Myd88*, *Trif*, *Ripk1*, *Ripk3*, *Mlkl*, *Tlr4*, were not only all up-regulated, but were also located in bands with *p*-values <0.05 and log2 (fold change) > 1, as observed in the lung tissues of mice with ALI (Supplementary Fig. 1a). To gain a comprehensive understanding of the necroptosis pathway, a full list of genes associated with necroptosis and apoptosis were downloaded from the Kyoto Encyclopedia of Genes and Genomes (KEGG). This list was then cross-referenced with extensive literature review to ensure its accuracy and completeness. Subsequently, this curated gene list was utilized to generate a heat map representing mRNA expression levels in both ALI and control mice (Fig. 1a, Supplementary Fig. 1b). The results of the heatmap showed that the vast majority of the genes visualised showed a significant up-regulation in expression, suggesting the possibility of high activation of the pathway in which they are located. Following the bulk-RNA sequencing analysis of differential genes, a Gene Ontology (GO) enrichment analysis was conducted (Supplementary Fig. 1c). This analysis revealed the presence of eight functional clusters associated with death, three of which showed significant relevance to the necroptosis pathway. Gene Set Enrichment Analysis (GSEA) was performed on these gene sets, further confirming the presence and activation of the necroptosis and apoptosis (Fig. 1b, Supplementary Fig. 1d). Meanwhile, based on GSEA analysis of GO, we found five most statistically significant death-associated functional pathways, and the fifth one was extremely strongly associated with necroptosis (Supplementary Fig. 1e). However, neither KEGG nor GO-based GSEA enrichment analysis revealed any evidence of necrosis. Moreover, the representative genes such as *Myd88*, *Trif*, *Ikk*, *Irf3*, as well as *Ripk1*, *Ripk3*, *Mlkl* were analyzed for interaction using correlation analysis based on their Fragments Per Kilobase of exon model per Million mapped fragments (FPKM) (Fig. 1c). The results revealed a substantial correlation among these genes.

### Necroptosis activation and expression in acute lung injury

The western blotting analysis of mice lungs revealed a significant increase in the expression of phosphorylated RIPK1 (Fig. 2a-c), RIPK3 (Fig. 2d-f), MLKL (Fig. 2g-i), cleaved-Caspase-3 (Supplementary Fig. 2a-c) and cleaved-Caspase-7 (Supplementary Fig. 2d-f) in the ALI compared to the control mice. Cleaved-Caspase-8 did not show a significant increase and caspase-8 levels decreased as shown in the ALI model (Supplementary Fig. 2g-i). The western blotting analysis of lung tissues from mice with ALI induced by cecal ligation and puncture (CLP), a model of polymicrobial sepsis resulting in intestinal perforation and necrosis, revealed activation of p-RIPK1 (Supplementary Fig. 3a-c). And no statistically significant changes were observed in p-RIPK3 and p-MLKL expression (Supplementary Fig. 3d-i). Additionally, the paraffin immunofluorescence of lungs showed a marked increase in fluorescence intensity of p-MLKL, p-RIPK1 and

p-RIPK3 in the ALI in comparison to control mice (Fig. 2j), further supporting the activation of necroptosis in ALI. At the same time, the western blotting results show a significant decrease in the expression level of Caspase-8. Previous findings have suggested that phosphorylation modification of RIPK1, RIPK3, and MLKL proteins and decreased expression of Caspase-8 are essential and indispensable parts of the necroptosis pathway[11,12], and these results are consistent with previous studies suggesting that there is activation of necroptosis early in the development of ARDS[29,30]. Nevertheless, necroptosis is not activated in the early stages of sepsis.

### LPS induced phosphorylation of RIPK1/RIPK3/MLKL in epithelial cells

To investigate the presence of necroptosis within epithelial cells, human airway epithelial cell line (HBE) was exposed to LPS intervention[31]. Specifically, HBE was exposed to LPS at a concentration of 200 μg/ml for 24 h. CCK-8 results showed a decrease in HBE activity (Supplementary Fig. 4a), western blotting revealed a significant increase in the phosphorylation levels of RIPK1 and RIPK3 (Supplementary Fig. 4b-h). In contrast, the phosphorylation levels of MLKL did not show obvious changes (Supplementary Fig. 4i-l). We thought that the in vitro cultivation could not fully replicate the changes as in vivo. To further investigate the potential activation of necroptosis within HBE under LPS intervention, we extended the exposure time to 48 h using the same concentration of LPS. Western blotting revealed p-RIPK1 (Fig. 3a-c), p-RIPK3 (Fig. 3d-f), and p-MLKL (Fig. 3g-i) was significantly increased after LPS stimulation compared to the control. The same results were obtained from immunofluorescence staining of HBE after LPS intervention (Fig. 3j, Supplementary Fig. 5). These results provide additional support for the existence of the necroptosis pathway in LPS-intervened airway cells. However, note that there might be a delay in the activation of necroptosis in vitro.

### Both of RIPK1 and RIPK3 involve in MLKL phosphorylation in LPS treated epithelial cells

MLKL is located downstream of both RIPK1 and RIPK3 in the necroptosis pathway. Phosphorylation of either RIPK1 or RIPK3 proteins has now been found to initiate necroptosis in other diseases, but it has not been clarified in ALI exactly which initiates the process of necroptosis[32,33]. As expected, LPS induced the expression of p-RIPK1, p-RIPK3, and p-MLKL in HBE, indicating the activation of necroptosis (Fig. 3). Subsequently,we used Necrostatin-1 (Nec-1, A specific RIPK1 inhibitor) and GSK'872 (A specific RIPK3 inhibitor)[33–36], to inhibit RIPK1 and RIPK3 respectively. After confirming that these inhibitors did not affect cell viability (Supplementary Fig. 6a), we treated HBE cells with 10 μM Nec-1, 1 μM GSK'872, and 200 μg/ml LPS for 48 h. Western blotting showed a significant reduction in p-RIPK1 (Fig. 4a, d, g, j, m, p), p-RIPK3 (Fig. 4b, e, h, k, n, q), and p-MLKL (Fig. 4c, f, i, l, o, r) levels, indicating the inhibitory effects of Nec-1 and GSK'872. The expression of p-MLKL in HBE induced by LPS decreases after Nec-1 or GSK'872 treatment, respectively, according to fluorescence staining (Supplementary Fig. 6b). These results indicate that the phosphorylation of both RIPK1 and RIPK3 is essential for the activation of MLKL.

### The co-reaction between RIPK1, RIPK3 and MLKL in LPS induced necroptosis

Considering the currently proposed viewpoint that the combination of p-RIPK1 and p-RIPK3 recruits MLKL and triggers the phosphorylation process[37], and combining with the above findings showing that inhibition of both affects the phosphorylation of the other. We assume that there is a state in which p-RIPK1, p-RIPK3, and p-MLKL bind to each other to form a complex during the process of necroptosis. To validate this, we performed co-immunoprecipitation assays targeting p-MLKL, p-RIPK1, and p-RIPK3 proteins individually (Fig. 5a-c). The results demonstrated that all three proteins exhibited bands at their respective molecular weights, indicating their presence as a complex within the cells. These findings provide further

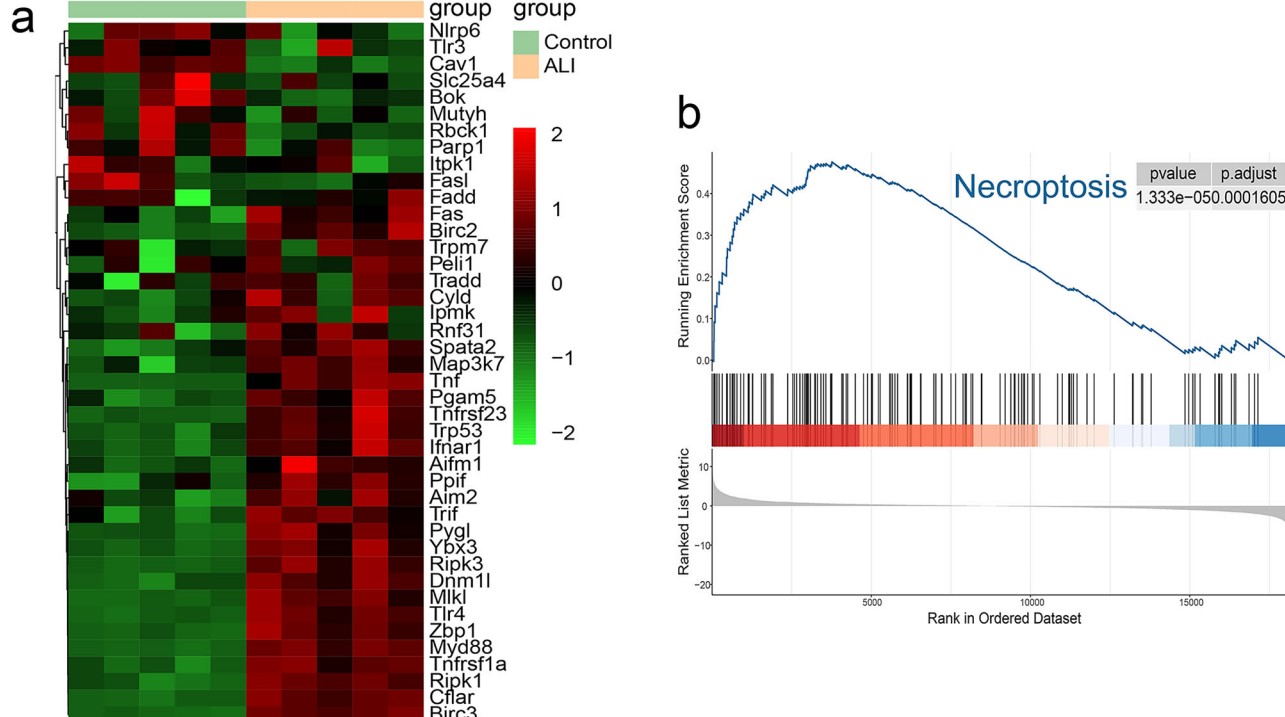

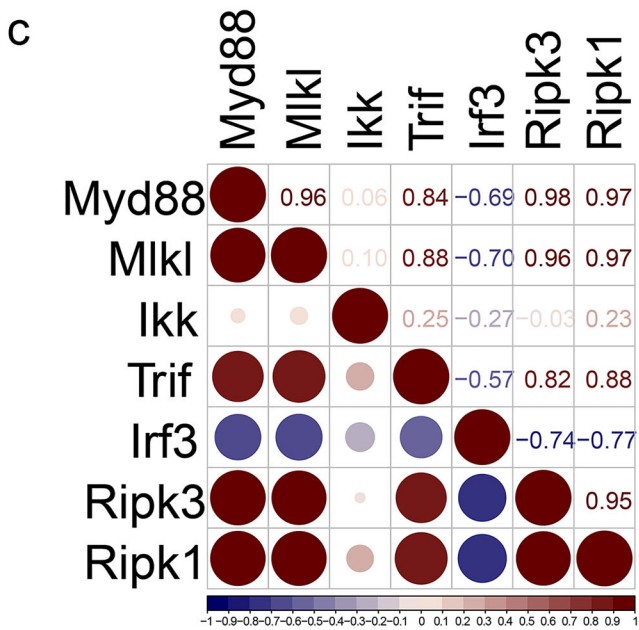

**Fig. 1 | Bulk RNA-seq analysis of lung tissues of ALI. a** Heat mapping of Bulk RNA-sequence results revealed that the expression of genes involved in the necroptosis pathway was altered in ALI group. **b** GSEA analysis using Bulk RNA-sequence data demonstrated that the necroptosis pathway was significantly activated in the lung tissue of mice in ALI group ($p < 0.05$). **c** Heatmap after correlation analysis based on Fragments Per Kilobase of exon model per Million mapped fragments (FPKM) of genes. Positive correlations are shown in red, negative correlations in blue, and the size of the circles and numbers represent the magnitude of the correlation. Samples from 5 mice per group were analyzed.

evidence that p-MLKL, p-RIPK1, and p-RIPK3 can form complexes, supporting the notion that they potentially interact with each other to regulate necroptosis. When co-immunoprecipitation was done with p-MLKL as the target, the expression of the three proteins was comparable in the co-precipitation group. In contrast, when co-immunoprecipitation was done with p-RIPK1 and p-RIPK3 as targets, the exposure of the bands of both in the co-precipitated group was significantly higher than that of the control group, whereas the bands of p-MLKL were equivalent or even lighter than that of the control group. Immunofluorescence staining performed on LPS-intervened HBE demonstrated the co-localization of p-RIPK1, p-RIPK3 and p-MLKL (Fig. 5d, Supplementary Fig. 7a). The protein structures of p-RIPK1, p-RIPK3, p-MLKL were obtained from the Protein Data Bank (PDB), and the 3D conformations of the complexes formed by their binding were predicted using the Cluspro website and the minimum binding

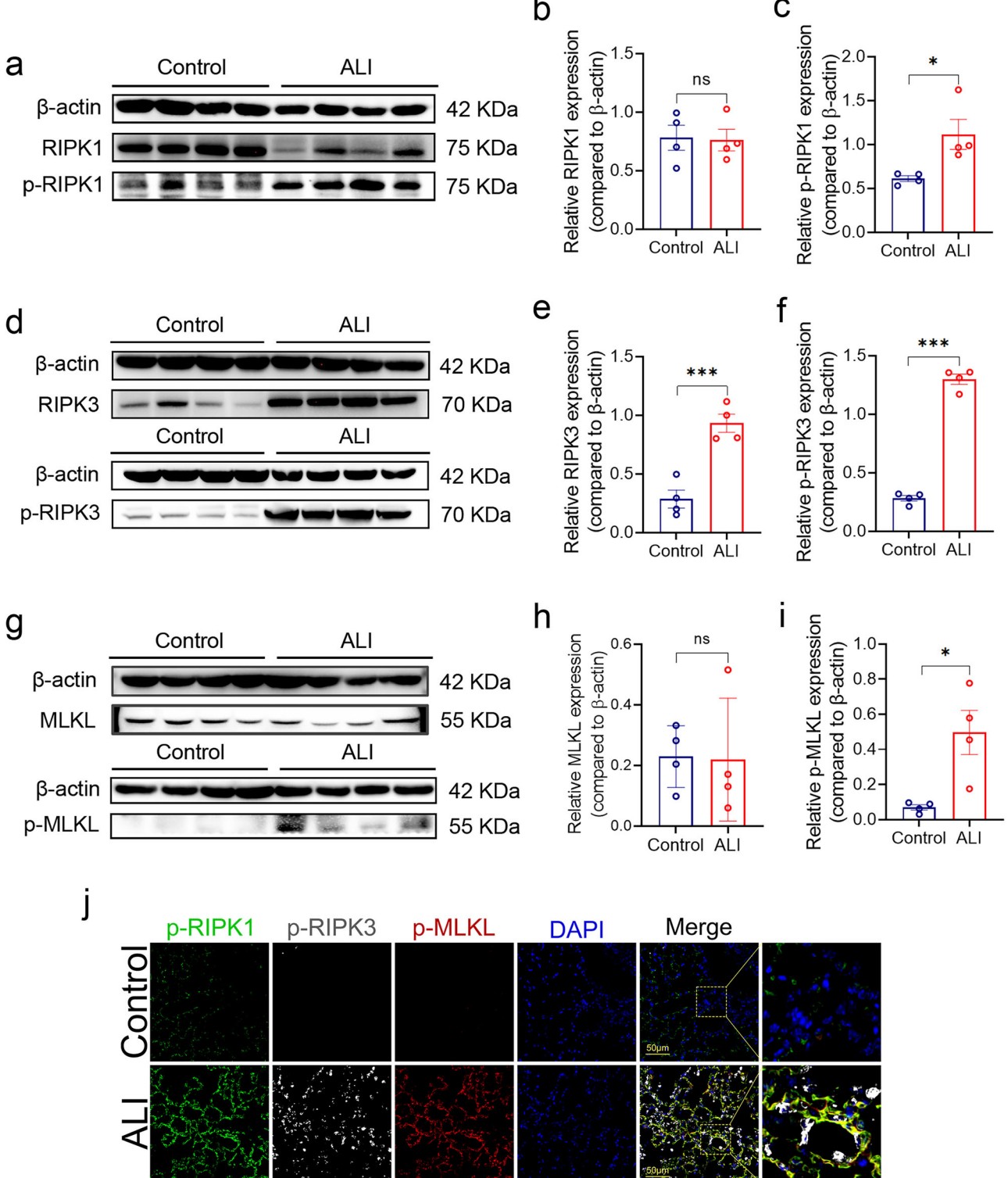

**Fig. 2 | Expression of necroptosis in LPS-induced ALI. a**, **d**, **g** Validation of protein expression levels of RIPK1, p-RIPK1, RIPK3, p-RIPK3, MLKL, and p-MLKL in mouse lung tissue with ALI using western blotting. **b**, **c**, **e**, **f**, **h**, **i** Quantitative statistical analysis of the western blotting results was performed by measuring the grayscale values of the bands using Image J, with the target bands normalized to the internal reference (β-actin). **j** Fluorescence microscope images of paraffin tissue sections from mouse lung tissues stained with anti-p-RIPK1 (green), anti-p-RIPK3 (grey), anti-p-MLKL (red), and DAPI (blue) for visualization, with a scale bar indicating 50μm. At least three samples were included in each group, with three randomly selected fields of view from each sample used to measure and analyze the corresponding fluorescence signal intensity. Significance was determined with $p$-values < 0.05, 0.01, and 0.001 by *, ** and ***, respectively, ns means not significant. The error bars represent the standard deviation (SD).

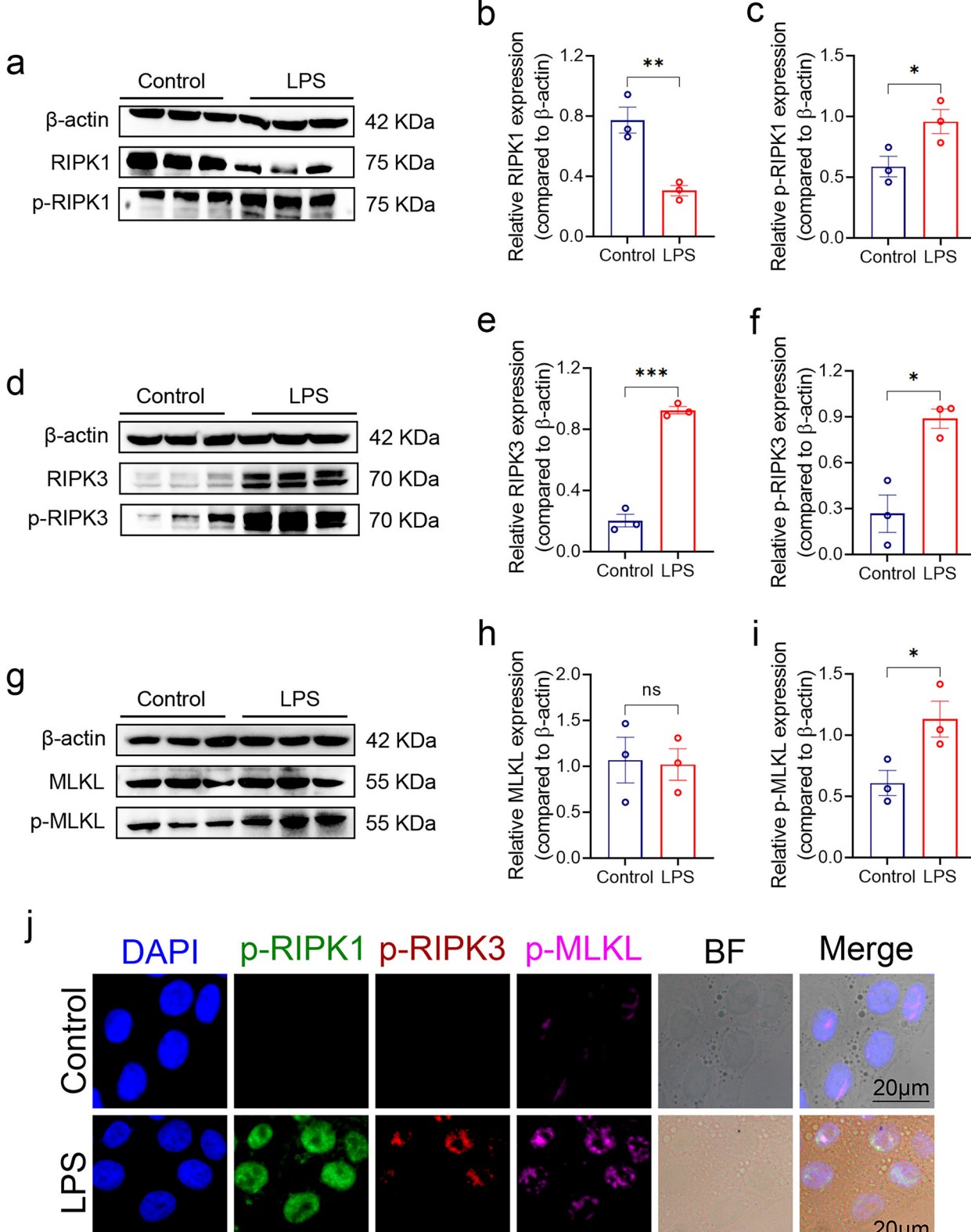

**Fig. 3 | RIPK1, RIPK3 and MLKL expression in LPS treated HBE in vitro.**
**a**, **d**, **g** Western blotting was performed to verify the expression of RIPK1, p-RIPK1, RIPK3, p-RIPK3, MLKL, and p-MLKL in HBE treated with 200 μg/ml of LPS for 48 h. **b**, **c**, **e**, **f**, **h**, **i** Quantitative analysis of the western blotting results was conducted by measuring the grayscale values of the bands using Image J. The target bands were normalized to the internal reference (β-actin). The data were then analyzed using GraphPad Prism 8, with an unpaired t-test applied to compare the control and ALI groups. **j** Fluorescence microscope images were captured for HBE treated with anti-p-RIPK1 (green), anti-p-RIPK3 (red), anti-p-MLKL (pink), and DAPI (blue) in both the control and LPS intervention groups. Scale markers of 20 μm were included in the images. Three samples were used for both groups in the experiment. Significance was determined with p-values < 0.05, 0.01, and 0.001 by *, ** and ***, respectively, ns means not significant. The error bars represent the standard deviation (SD).

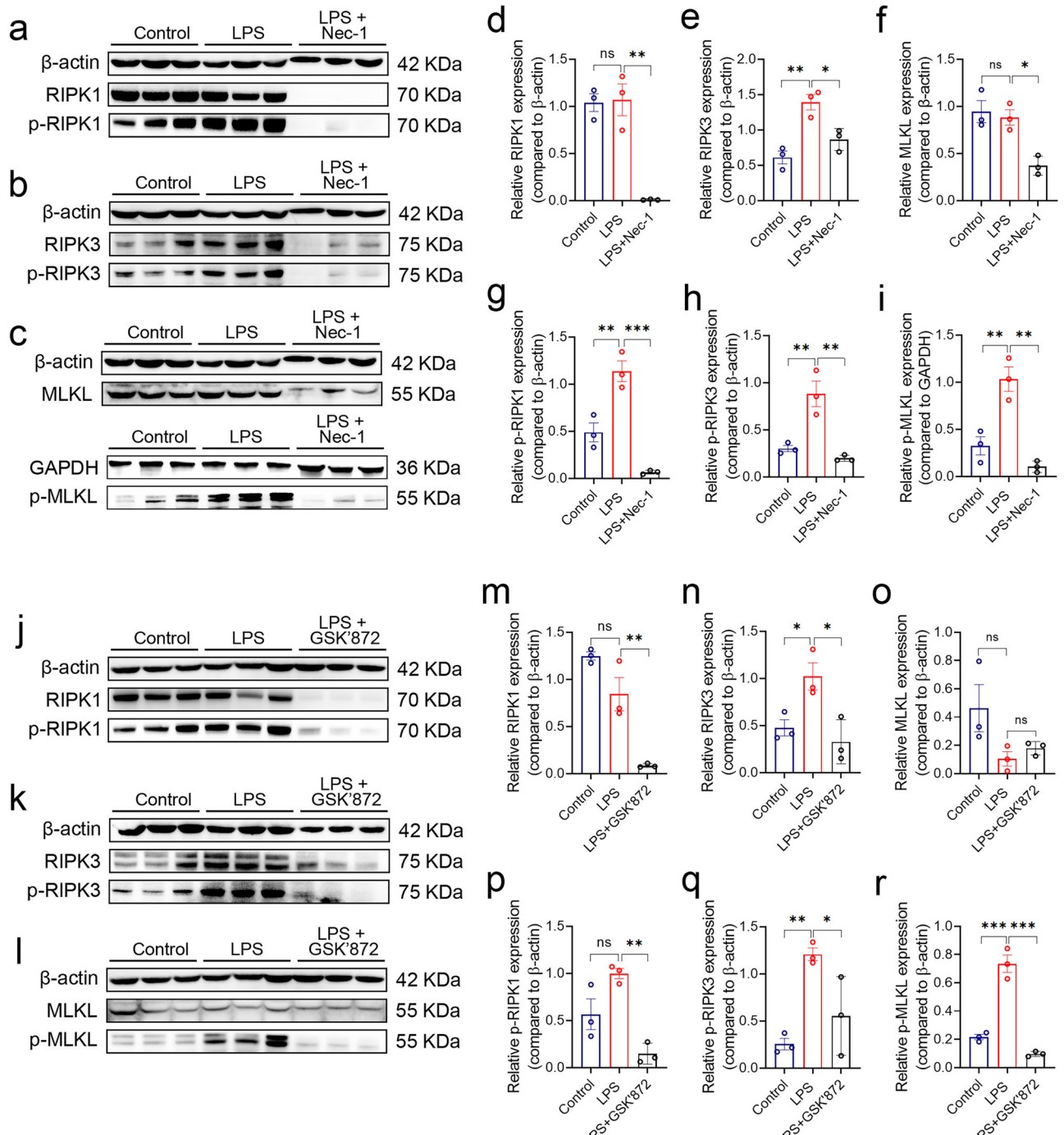

**Fig. 4 | The expression of necroptosis following the inhibition of RIPK1 and RIPK3 respectively. a**-**c** Western blotting was performed to verify the expression of RIPK1, p-RIPK1, RIPK3, p-RIPK3, MLKL, and p-MLKL in HBE treated with LPS at a concentration of 200 μg/ml or 200 μg/ml LPS + 10 μM Nec-1 intervention for 48 h. The bands for MLKL, RIPK1 and p-RIPK1 were obtained from the same membrane after multiple exposures, so the MLKL, RIPK1 and p-RIPK1 bands are on the same membrane and share the same internal reference protein band. **d**-**i** Quantitative statistical analysis of the western blotting results was conducted using Image J to measure the grayscale values of the bands. The target bands were normalized to the internal reference bands (β-actin or GAPDH). The data were then imported into GraphPad Prism 8 for statistical analysis, where a multifactor analysis of variance

was performed to compare the control, LPS, and LPS+Nec-1 groups. **j**-**l** Western blotting was performed to verify the expression of RIPK1, p-RIPK1, RIPK3, p-RIPK3, MLKL, and p-MLKL in HBE treated with LPS at a concentration of 200 μg/ml or 200 μg/ml LPS + 1 μM GSK'872 intervention for 48 h. The bands for MLKL, RIPK1, p-MLKL and p-RIPK1 were obtained from the same membrane after multiple exposures, so the MLKL, RIPK1, p-MLKL and p-RIPK1 bands are on the same membrane and share the same internal reference protein band. **m**-**r** Quantitative statistical analysis of the western blotting results was performed using Image J. Significance was determined with $p$-values < 0.05, 0.01, and 0.001 by *, ** and ***, respectively, ns means not significant. The error bars represent the standard deviation (SD).

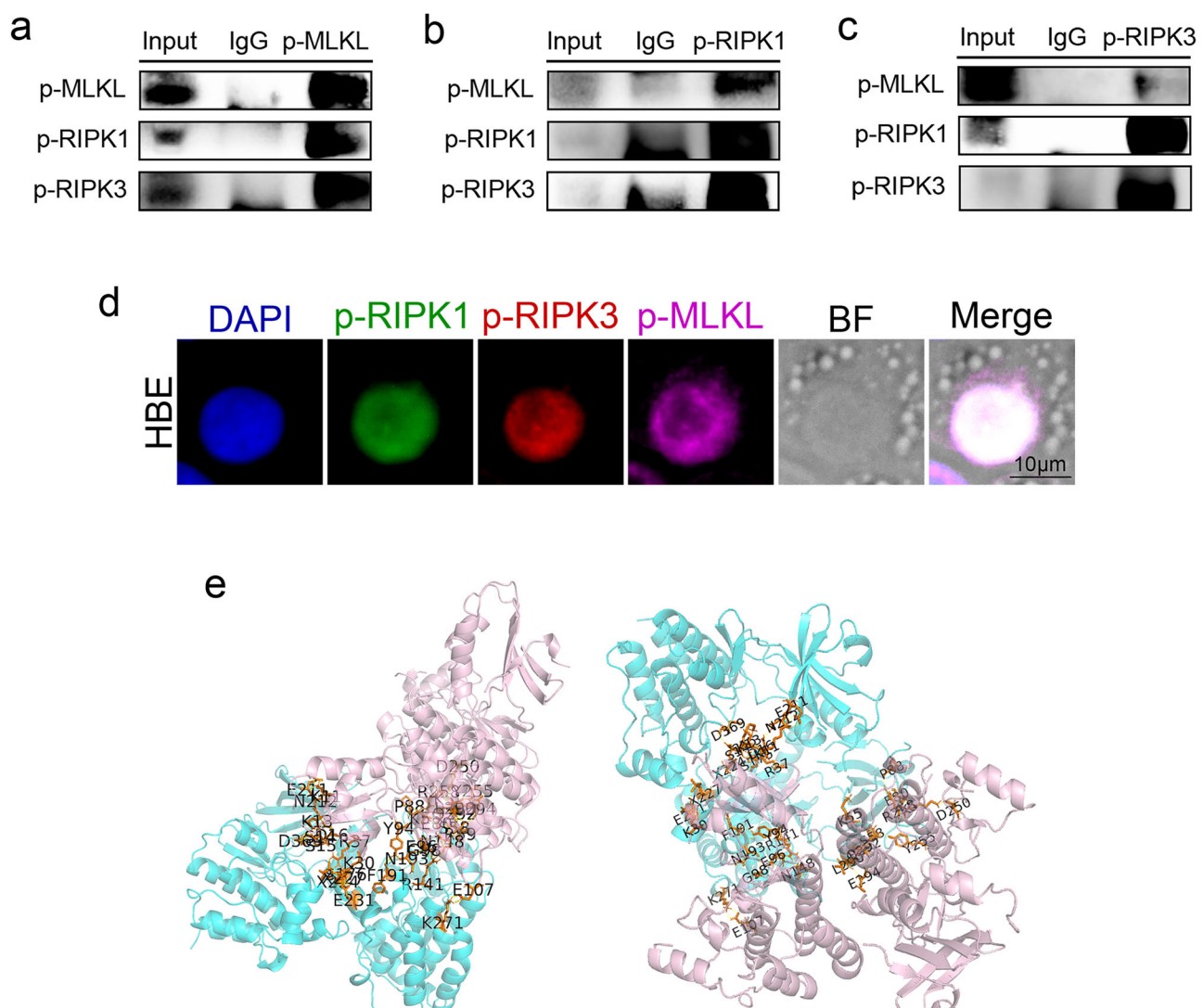

**Fig. 5 | The co-location of p-RIPK1, p-RIPK3 and p-MLKL during necroptosis process. a**-**c** Immunoprecipitation assays were conducted on HBE treated with LPS at a concentration of 200 μg/ml for 48 h, using p-MLKL, p-RIPK1, and p-RIPK3 as targets. p-RIPK1 and p-RIPK3 could be detected when protein complexes were precipitated with p-MLKL as the target. Similarly, the p-RIPK1 precipitation product could detect both p-MLKL and p-RIPK3, and the p-RIPK3 precipitation product could detect both p-MLKL and p-RIPK1. **d** Fluorescence microscopy images were captured for HBE treated with LPS, showing the localization of p-RIPK1 (green), p-RIPK3 (red), p-MLKL (pink), and DAPI (blue) as markers. The images revealed the co-localization of these proteins within the cells. The scale length is 10 μm. **e** A 3D model was generated to simulate the binding states of RIPK1 (blue) and the RIPK3-MLKL complex (pink). The model was obtained from the Protein Data Bank (PDB) database using the ClusPro website, which predicts protein-protein interactions and complex structures.

energies were obtained for the relevant structures (Supplementary Fig. 7b), and finally their protein-binding sites were demonstrated by Pymol (Fig. 5e). All of these results support that the protein of p-RIPK1, p-RIPK3, p-MLKL bind to each others.

### MYD88 and TRIF are essential for necroptosis activation

MYD88 and TRIF are different adapter proteins that are recruited upon TLR4 activation which enable signal transduction by mediating the recruitment and activation of downstream effector molecules. These two factors contribute significantly to the overall cellular response to LPS stimulation[38,39]. In order to investigate the role of TRIF and MYD88 in LPS induced necroptosis, the expression was silenced with siRNA individually (Supplementary Fig. 8a-b). Samples were collected and subjected to a western blotting after 48 h, revealed a decrease in the expression of p-RIPK1 (Fig. 6a-c), p-RIPK3 (Fig. 6d-f), and p-MLKL (Fig. 6g-i) in cells treated with siRNA-*Myd88* or siRNA-*Trif* (Fig. 6j-r) compared to LPS treated alone. HBE was treated with 200 μg/ml LPS for 6, 12, and 24 h for Western blotting

analysis, indicated a significant increase in the levels of p-IKK and p-IRF3 at 6 h post-treatment (Supplementary Fig. 8c-h). These findings suggest that both MYD88 and TRIF play a role in the activation of the necroptosis pathway.

### Alveolar type II-targeted delivery of GW806742X via SPC antibody-modified lipid micelle

Alveolar type II epithelial cells (AT II) can differentiate into alveolar type I epithelial cells (AT I), and take on the responsibility of secreting alveolar surface-active substances and reducing alveolar surface tension. Therefore, we aimed to target AT II and treat lung damage through cell repair. GW806742X, an MLKL inhibitor, has demonstrated the ability to inhibit necroptosis, indicating its potential as a treatment for inflammatory injuries[40–44]. However, its clinical application is hindered by its limited water solubility and low bioavailability. We thought to prepare an lung SPC (surfactant protein C). which is an extremely hydrophobic surfactant protein essential for lung function and homeostasis after birth,

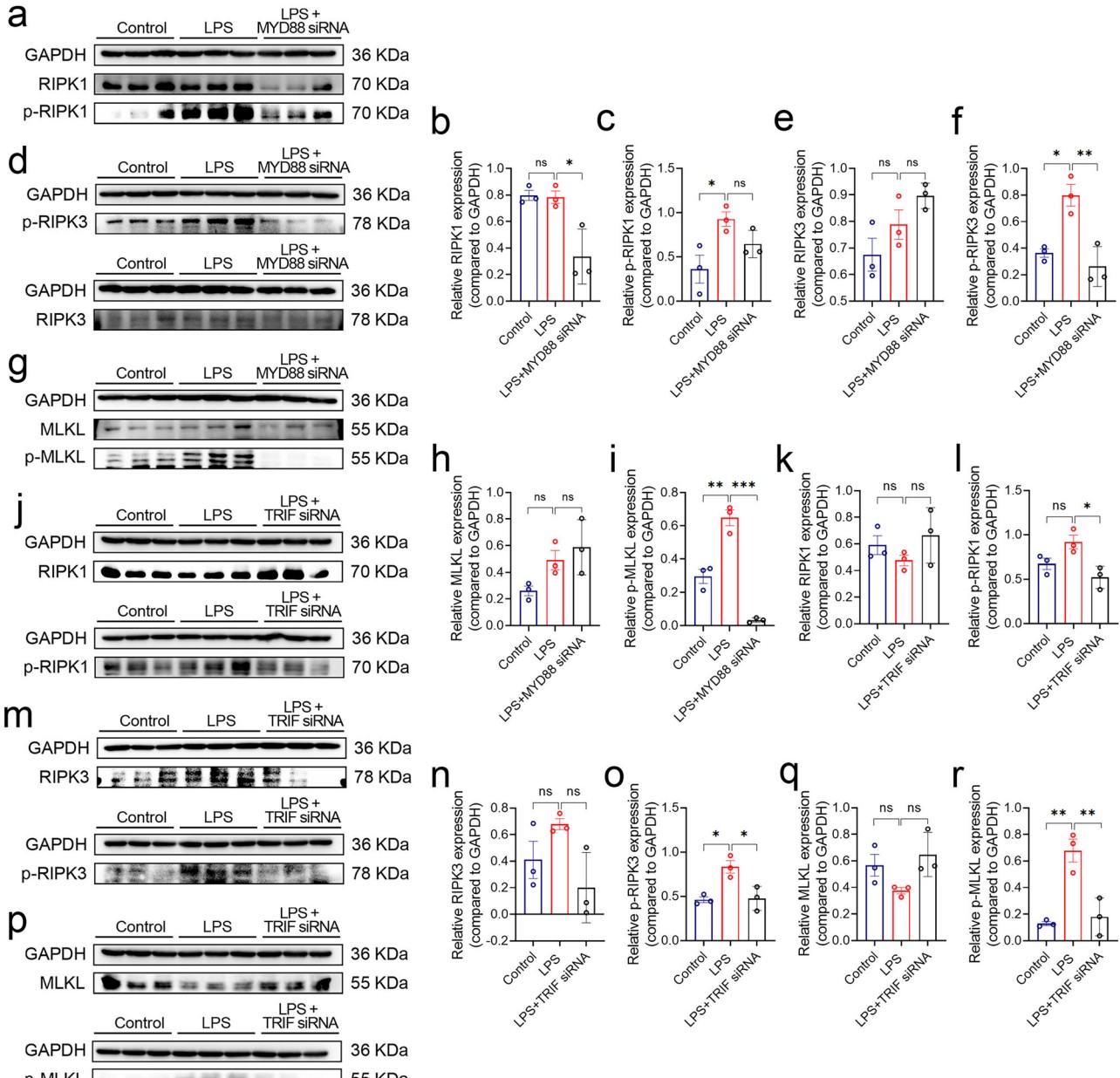

**Fig. 6 | The downstream signaling in LPS induced necroptosis. a, d, g** Western blotting to verify RIPK1, p-RIPK1, RIPK3, p-RIPK3, MLKL and p-MLKL expression in HBE with LPS 200 µg/ml or LPS + MYD88 siRNA-treatment. **b, c, e, f, h, i** Grayscale values of the western blotting bands were quantified using Image J, and the target bands were normalized to the internal reference (GAPDH). **j, m, p** Western blotting to verify RIPK1, p-RIPK1, RIPK3, p-RIPK3, MLKL and p-MLKL expression in HBE with LPS 200 µg/ml or LPS + TRIF siRNA. The bands for MLKL and RIPK1 were obtained from the same membrane after multiple exposures, so the MLKL and RIPK1 bands are on the same membrane and share the same internal reference protein band.
**k, l, n, o, q, r** Quantitative statistical analysis of western blotting using Image J and GraphPad Prism 8. Statistical significance was determined with *p*-values of 0.05, 0.01, and 0.001, which are represented by *, **, and ***, respectively, with ns indicating no significant difference. The error bars represent the standard deviation (SD).

antibody-modified and GW806742X-loaded lipid nanoparticle targeting AT II, abbreviated as SPC-M-GW, with the abbreviation used to represent this targeted drug. The GW806742, DSPE-mPEG2000, and DSPE-PEG2000-MAL powders were combined and dissolved in methanol. The lipids were then self-assembled around the inhibitor by dialysis displacement using PBS buffer. Subsequently, the antibodies were coupled to the lipids, and any unbound SPC antibodies were removed through dialysis (Fig. 7a). In which, SPC antibodies can target AT II, so it was attached to lipid micelles. Finally, the lipid micelle was concentrated using ultrafiltration tubes, and its SPC-M-GW loading efficiency was determined using high-performance liquid chromatography (HPLC). The electron microscopy analysis revealed the distinctive features of the lipid micelle particles. They exhibited a spherical shape, maintaining a uniform and consistent size throughout (Fig. 7b). Additionally, the hydrodynamic particle size were evenly dispersed without any signs of aggregation. The nanosizer of micelles was determined to be an average of 110 nm using a nanosizer point analyser. Furthermore, micelles conjugated with antibodies exhibited a slightly larger diameter of 140 nm (Fig. 7c). Additionally, the micelles possessed a negative potential, indicating a balanced and stable system (Fig. 7d). The use of fluorescent dye Cy5 to label micelles allows for visualization of their localization. CY5-labeled modified and unmodified lipid nanoparticles were administered to mice via airway instillation. To validate the targeted effect of the prepared lipid micelle, we conducted immunofluorescence staining of lung tissue and found that SPC-modified DSPE-PEG micelle containing

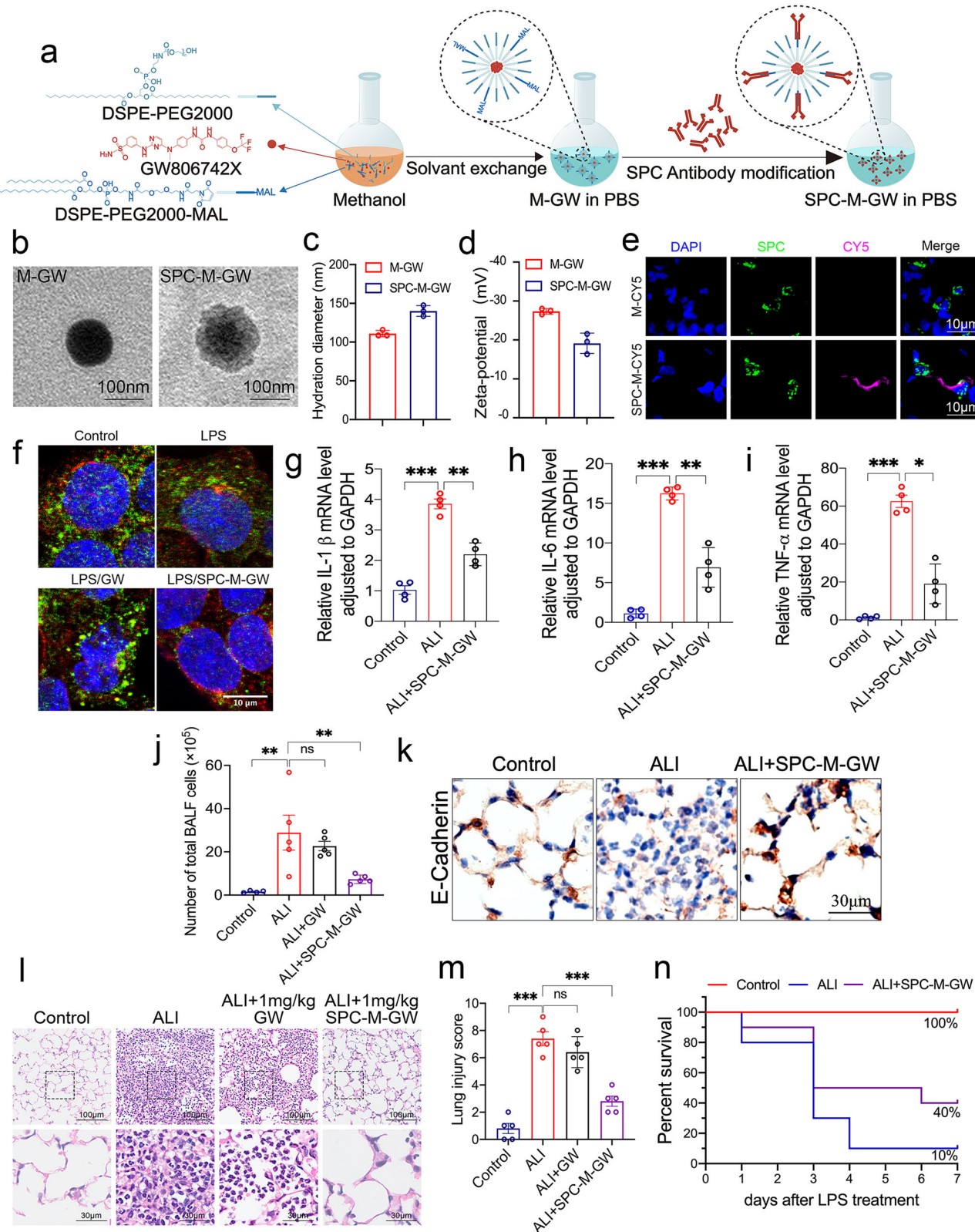

CY5 can specifically bind to AT II (Fig. 7e). The SPC protein on the membrane surface of MLE was stained by immunofluorescence, showing a certain level of SPC expression on the surface (Supplementary Fig. 9), which is consistent with previous studies[45]. The GW806742X and SPC-M-GW at 1 μM were used to treat the LPS-stimulated MLE cells. Tubulin was employed to stain the cytoskeleton, revealing the distribution of MLKL

protein within cells. In images captured by confocal microscopy, it can be observed that in SPC-M-GW treated LPS-stimulated cells, MLKL rarely accumulates on the cell membrane compared with control or LPS alone treated MLE cells (Fig. 7f). In the LPS-induced ALI mouse model, SPC-M-GW (1 mg/kg) or GW806742X (1 mg/kg) was administered via the airway 6 h after LPS administration and samples were collected 24 h after the model

**Fig. 7 | The effect of lipid micelle-encapsulated GW806742X targeting MLKL in AT II in ALI. a** The process of preparing SPC-modified PEG-DSPE micelle containing GW806742X(SPC-M-GW) targeting AT II. **b** Transmission electron micrographs of lipid nanomicelles with unconjugated antibody (M-GW) and conjugated antibody (SPC-M-GW), the scale length is 100 nm. **c, d** Statistical plots of the hydration diameters and zeta potentials of the M-GW and the SPC-M-GW. **e** Fluorescence microscope images of mouse lung tissue stained with DAPI, SPC and Cy5 can be seen to appear in aggregates in AT II. **f** Confocal microscopy images of MLE, with or without treatment, where blue (DAPI) represents the cell nucleus, red (Tubulin) represents the cytoskeleton, and green illustrates the distribution of MLKL within the cells, the scale bar is 10 μm. **g-i** The changes of mRNA levels for IL-1β, IL-6, TNF-α in the lung tissues in Control, ALI and ALI/SPC-M-GW 1 mg/kg treatment groups. **j** The total cell count in BALF. **k** Immunohistochemical staining images of E-cadherin protein in mouse lung tissue. **l** Microscope images of HE-stained lung tissues of mice with LPS, GW (GW806742X) or SPC-M-GW, the dose for two groups of GW is 1 mg/kg, and lung injury was scored (**m**). **n** Survival curve of ALI and SPC-M-GW treatment mice. After airway instillation for modeling and treatment, the number of deaths and body weight of mice were recorded daily. Mice were considered deceased when their body weight dropped more than 20% from the original weight. Significance was determined with $p$-values < 0.05, 0.01, and 0.001 by *, ** and ***, respectively, and ns means not significant. The error bars represent the standard deviation (SD).

was established. The mRNA levels in lung tissues revealed a significant increase in inflammatory factors such as IL-1β, IL-6, and TNFα in ALI, and a marked decrease after treatment with SPC-M-GW (Fig. 7g-i). The total inflammatory cells in the alveolar lavage fluid were significantly elevated in ALI mice, but decreased significantly after treatment with 1 mg/kg of SPC-M-GW (Fig. 7j). Immunohistochemical staining of E-cadherin protein in the lung tissues of Control, ALI, and ALI + SPC-M-GW groups revealed a significant reduction in E-cadherin expression between cells in the lung tissues of ALI mice, indicating a marked disruption of the intercellular barrier structure. In contrast, the expression of E-cadherin protein in the lung tissues of the treatment group was significantly higher compared to the ALI group, suggesting that the alveolar barrier structure was preserved to some extent (Fig. 7k). According lung HE staining, the ALI mice showed significant alveolar collapse, loss of epithelial cells, accumulation of inflammatory cells in the alveolar lumen, and disruption of the continuity of the hyaline membrane, along with localized thickening (Fig. 7l-m). No inhibitory effect of GW806742X was observed in alleviating inflammation-induced damage in lung tissues in 1 mg/kg, and unmodified antibody M-GW only has a weak effect (Fig. 7i, Supplementary Fig. 10). Surprisingly, administration of SPC-M-GW at a dose of just 1 mg/kg significantly inhibited inflammatory cell aggregation, reduced interstitial edema, and improved capillary dilation, congestion, and hemorrhage (Fig. 7j-m). 90% of the ALI mice died within 4 days, while the survival rate of ALI mice treated with 1 mg/kg SPC-M-GW was 50% at 4 days and 40% at 7 days (Fig. 7n). These results indicated that AT II-targeted delivery of necroptosis inhibitors via SPC antibody-modified lipid micelles can significantly alleviate airway inflammatory injury, thereby improving mouse survival rates.

## Disscussion

ALI/ARDS is a condition characterized by widespread damage to the airways and an inflammatory response. At the same time, necroptosis refers to a type of cell death that can be modified and is also accompanied by a severe inflammatory response. As a result, the damage to the airways and the inflammatory reactions mutually exacerbate each other, leading to the progression of the disease[23,46]. In this study, we demonstrated that necroptosis in the LPS-induced ALI model is mediated by the activation of RIPK1 and RIPK3, as previously established in other studies. We also confirmed that the simultaneous phosphorylation of RIPK1 and RIPK3 triggers downstream responses (Fig. 8). We proved the presence of p-RIPK1, p-RIPK3, and p-MLKL complexes, further indicating the interaction of these phosphorylated proteins. Additionally, our findings indicated that in LPS-induced ALI, TLR4 triggers necroptosis by the combined action of two downstream factors, MYD88 and TRIF. Finally, we developed a delivery system targeting the key necroptosis factor MLKL in AT II, lipid micelle-encapsulated GW806742X, which effectively alleviated acute lung injury through local airway administration.

It was proposed that when cells undergo necroptosis, it is LPS that activates its downstream TIRAP and MYD88 in the plasma membrane or activates the TRAM and TRIF pathways by stimulating the TLR4 receptor[47–49]. However, it is noteworthy that dual signalling mediated by TRIF and MYD88 has received little attention in previous studies[50–52]. Moreover, it has been clearly demonstrated in previous studies that TRIF-mediated signalling is non-MYD88-dependent and that the signalling

cascade it mediates occurs after endocytosis, whereas TLR4-mediated receptor endocytosis blocks MYD88-dependent signalling[47,53]. Nevertheless, we observed inhibition of the necroptosis pathway in HBE when either MYD88 or TRIF was blocked. Additionally, in our study, we observed an increase in the phosphorylation of IKK and IRF3 downstream of MYD88 and TRIF after the intervention with LPS. The concurrent activation of both pathways may be associated with the ability of LPS to evade endosomes created through TLR4-mediated endocytosis, or with the reduction of intracellular CD14. Previous findings have revealed that the activation of TRIF relies heavily on intracellular CD14, and when intracellular CD14 levels are depleted, cellular TLR4 signaling shifts towards downstream transmission via the MYD88 pathway[47,54,55]. However, the sequencing of activation for the MYD88 and TRIF pathways in the two aforementioned theories contradicts the proposition put forth by Reynoso et al. They suggest that MYD88 is responsible for initiating early responses, while TRIF is involved in later responses[56]. These disparities in results could arise from variances in cell types, the timing of interventions, or even variations in the dosage of medication, potentially contributing to divergent outcomes.

In the present study, we demonstrate that phosphorylation of both RIPK1 and RIPK3 is an integral part of necroptosis in the development of ARDS, and that intervention in either of the two can have a dramatic effect on the other. Not only RIPK1 is not essential for necroptosis, but also inhibition of the kinase activity of RIPK1, which converts the necroptosis pathway to RIPK1-independent necroptosis, results in the production of large amounts of DAMP by the cells, which induces a severe inflammatory response[15,57–60]. It has also been pointed out that RIPK3 is involved in the transmission and activation of various signalling pathways, and its inhibition will promote the inflammation-related STING pathway, while the inhibition of MLKL membrane translocation can inhibit the STING signalling pathway, which can inhibit the body's inflammatory response to a certain extent[60,61].

Currently, lipid micelles serve as a widely employed drug delivery technology due to their lipophilic properties. This technology significantly enhances the efficiency of cellular drug uptake and simultaneously minimizes the undesired degradation of drugs within the body[62,63]. Lipid micelles targeted delivery also enhances the absorption efficiency of drugs by the target cells, leading to lower dosages and reduced side effects in the treatment of diseases, and can be further enhanced in targeting specificity through chemical modification by attaching targeting ligands[64]. In our study, GW806742X, an MLKL inhibitor, was incorporated into SPC antibody-modified lipid micelles for targeted delivery to AT II using SPC antibody named SPC-M-GW. The micelles allowed for efficient release of a concentrated dose of GW806742X into the cytoplasm of these cells, enhancing intracellular uptake. In addition, due to the targeted function of the micelles, SPC-M-GW can effective suppress of MLKL membrane translocation, thereby achieving the purpose of inhibiting the process of necroptosis. And finally, it was demonstrated that this targeted medicine successfully alleviated airway inflammation-associated damage by specifically inhibiting necroptosis in alveolar epithelial cells. Nevertheless, as we have previously validated, these conclusions cannot be applied to sepsis models. The conclusions drawn from our study are specific to lung injury caused by direct factors (pulmonary factors). Although we only used LPS to model ALI in this study, previous research has already explored the presence

**Fig. 8 | Schematic illustrating the potential targeted therapy for necroptosis in acute lung injury.** The pathological characteristics of ALI involve diffuse inflammatory injury of the airway epithelial/endothelial cells, which is mediated by necroptosis. During this process, phosphorylation and complex formation of RIPK1, RIPK3, MLKL, as well as activation of TLR4 downstream signaling pathways involving TRIF and Myd88 are involved. The lipid micelle-encapsulated GW806742X (SPC-M-GW) that specifically targets MLKL in AT II has successfully used it to treat acute lung injury.

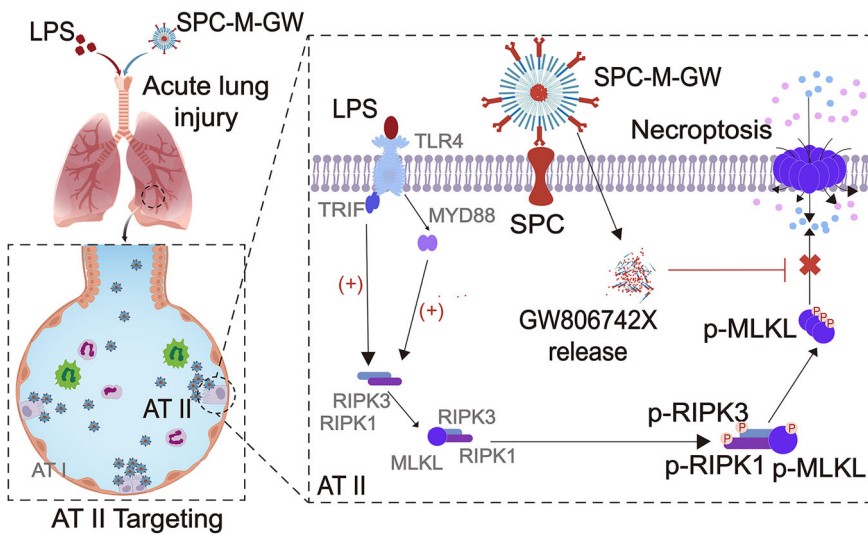

of necroptosis in ALI induced by H7N9 virus infection, corn oil, or ischemia-reperfusion[65–69].

In conclusion, we demonstrated the activation of p-RIPK1, p-RIPK3, and p-MLKL, as well as the formation of their complexes in LPS-stimulated acute lung injury. Additionally, we highlighted the involvement of MYD88 and TRIF in the necroptotic signaling pathway within this model. Furthermore, we have developed a drug that inhibits MLKL oligomerization and its translocation to the membrane, allowing for specific targeting of alveolar type II epithelial cells. This approach reduces damage and inflammatory progression during lung injury, offering a novel strategy for the treatment of acute lung injury or acute respiratory distress syndrome.

## Method
### Mice
Male C57BL/6 mice, aged 6-8 weeks, weighing between 16-24 grams, were purchased from SHANGHAI SLAC LABORATORY ANIMAL CO.LTD., and were acclimated for one week in the animal laboratory of the Clinical Research Center of the Second Affiliated Hospital of Zhejiang University School of Medicine. The mice were housed under specific pathogen-free (SPF) conditions, with a temperature of $25 \pm 1°C$, humidity of $60 \pm 5\%$, a 12 h light/dark cycle, and free access to water. All animal experiments were approved by the Ethics Review Committee of the Second Affiliated Hospital of Zhejiang University School of Medicine (No. 2022.050). We have complied with all relevant ethical regulations for animal use.

### Preparation of SPC-M-GW
Since GW806742X (Selleck, E1395) is not readily dissolved in water, a solvent exchange method was used to prepare the lipid micelle. The lipid compositions used were as follows: DSPE-PEG2000 (Tanshui Technology Co, a80010201-2000-1g) and DSPE-PEG2000-MAL (Yusi, YS-D223025) (mass ratio, 4:1). All lipid materials and GW806742X (mass ratio, 20:1) was dissolved in methanol, and put in dialysis bag (Molecular weight cut-off, MWCO 100-500, Yuanye, SP131057-0.5 m) at 4°C to be dialyzed against PBS, pH 7.5. The outer phase was replaced with fresh buffer solution at 2, 4, 6, 8, and 12 h. After 24 h, the lipid materials and GW806742X were self-assembled into lipid micelle. The SPC Antibody (Affinity, #DF6647) were dissolved in the HEPES buffer (1 M, pH 7.5), and the lipid micelle was added to a 50% glucose solution so that it ultimately contains 5% glucose[70]. Antibody dilutions were mixed with glucose-containing lipid micelle in the ratio of 1:2 molar ratio of antibody to DSPE and placed at 4°C on 3D suspension instrument. Sample was collected after 6 h and placed in a dialysis bag (MWCO 1000 K, Yuanye, SP131486-0.5 m) at 4 °C for 24 h. The outer phase was also replaced with fresh buffer solution at 2, 4, 6, 8, and 12 h. The collected

samples were finally concentrated and purified by placing them in ultrafiltration tubes with a pore size of 3KDa (Merck, UFC800308).

### Characterization of SPC-M-GW
The drug loading of SPC-M-GW was determined by high performance liquid chromatography (HPLC). The SPC-M-GW were solubilized with methanol before HPLC analysis. The Daojin LC-20AD HPLC system equipped with a Ultimate Plus-C18 4.6*100 mm 3.5um column was used. The column was eluted with 0.1%TFA acetonitrile at a flow rate of 1.0 mL/min. Sample was detected at 254 nm. The SPC-M-GW drug loading capacity was determined using a calibration curve obtained in the same conditions using standard concentrations of GW806742X (ranging between 10 and 60 μg/mL); the correlation coefficient $R^2$ was 0.9981. The particle size and zeta potential were measured using a laser particle sizer (Malvern, ZSV3100) and photographed using an electron microscope (HITACHI, H-7650), and the particle sizes were measured to be 110 and 140 nm, and the potentials to be-27.35 mV and -19.13 mV for M-GW and SPC-M-GW respectively. SPC-M-GW loading efficiency (LE) was calculated as follows:

LE (%)=(Mass of GW806742X in 1 ml of solution /Mass of SPC-M-GW in 1 ml of solution)×100%

To confirm the successful connection of SPC antibody, a BCA detection kit (yamei, ZJ101) was used. Calculations of grafting rate (GR) were performed by the following methods:

GR (%)=(Total mass of antibody/initial mass of antibody input) × 100%.

The final test results were: LE = 28.67%, GR = 30.23%.

### Preparation and characterization of SPC-M-CY5
In accordance with the above method, DSPE-PEG2000 (Tanshui Technology Co, a80010201-2000-1g), DSPE-PEG2000-MAL (Yusi,YS-D223025), DSPE-PEG2000-NHS (Ponsure Biological, PS2-HE1-2K-100mg) and CY5-NH2 (MeloPEG, 431201-2) were added according to a molar ratio of 20:5:1:1 to prepare and assay SPC-M-CY5.

### Acute respiratory distress syndrome (ARDS) or pharmacological treatment animal model construction
Mice were anesthetized with 200 μL of 1% pentobarbital administered intraperitoneally and were administered a 50 μL solution of LPS (Sigma-Aldrich, L9143) at a concentration of 10 mg/kg via the airway to induce the ALI model. To treat ALI model, mice were administrated with 60ul solution of SPC-M-GW in a concentration of 2 mg/kg, 1 mg/kg, 0.5 mg/kg via the airway after LPS intervention 6 h. For the control group, mice were treated with an equal volume of PBS. Lung tissues were collected after 24 h of LPS, GW806742X or PBS treatment for the further experiments.

## Cecal ligation and puncture (CLP) model construction

Male C57BL/6 mice (6-8 weeks old, weighing approximately 25 g) were used. After shaving the abdominal hair, a 1 cm incision was made along the midline of the abdomen. The skin and abdominal muscles were bluntly separated, and the free cecal tip was located in the abdominal cavity. A 5-0 polyester suture (Yuanlikang) was used to ligate the cecum 2 cm from the tip. A 12-gauge syringe needle was then used to perforate the ligated cecum, and pressure was applied to the cecum at the ligation site until fecal material was observed to flow from the perforation. After the cecum was returned to the abdominal cavity, the abdominal muscles and skin were sutured sequentially[71,72].

## Cell culture and LPS stimulation

The human bronchial epithelial cell line (HBE), which were purchased from American Type Culture Collection (ATCC, CRL-2741), propagated and maintained in 1640 medium (Gibco, C11875500BT) supplemented with 10% fetal bovine serum (FBS, Meilunbio, 9048-46-8). The mouse alveolar epithelial cell line (MLE), were obtained from the American Type Culture Collection (ATCC, CRL-2110), and were cultured in DMEM/F12 medium (Gibco, C11330500BT) supplemented with 10% fetal bovine serum. The cells were incubated in a constant temperature $CO_2$ incubator at 37°C with 5% $CO_2$. The experimental group was treated with 200 μg/ml LPS (Sigma-Aldrich, L9143), while the control group received the same volume PBS.

## siRNA transfection

The siRNA of *Myd88* and *Trif* were transfected into cells using GP-transfect-Mate. After 6 h, the liquid was changed to 1640 medium containing 10% FBS to continue incubation for 12 h, then 1640 medium containing 200 μg/ml LPS and 10% FBS was given to the LPS intervention group and siRNA intervention group as described above, and the samples were collected after 48 h of incubation, and stored at −80 °C. (*Trif*-S: CGGAACAGAAAUUCUAUAAUT; *Trif* - AS: UUA UAGAAUUUCUGUUCCGTT; *Myd88*- S: GACUUUGAGUACUUGGAGATT; *Myd88*-AS: UCUCCAAGUACUCAA AGUCTT).

## Quantitative real−time polymerase chain reaction

The lung tissues or cells modelled and obtained by "*In vivo animal experiments*" or "*Cell Culture and LPS Stimulation*" were used for PCR assays. Trizol reagent (Vazyme, R411-01) was used to extract total RNA from lung tissue according to the manufacturer's instructions. The PrimeScript RT Reagent Kit was used to make cDNA (AG, AG11706). On QuantStudio 5 PCR equipment, qRT-PCR was performed using AceQ Universal SYBR qPCR Master Mix (Vazyme, Q111). The following were the primers utilized in this study: m*Gapdh*-F: CATCACTGCCACCCAGAAGACTG; m*Gapdh*-R: ATGCCAGTGAGCTTCCCG TTCAG; m*Il-6*-F: TACCACTTCACAAGTCGGAGGC; m*Il-6*-R: CTGCAAGT GCATCATCGTTGTTC; m*Tnfα*-F: GGTGCCTATGTCTCAGCCTCTT; m*Tnfα*-R: GCCATAGAACTGATGAGAGGGAG; m*Il-1β*-F: TGGACCTTCCAGGATGAGGACA; m*Il-1β*-R:GTTCATCTCGGAGCCTGTAGTG; h*Gapdh*-F: GTCTCC TCTGACTTCAACAGCG; h*Gapdh*-R: ACCACCCTG TTGCTGTAGCCAA; h*Myd88*-F: GACUUUGAGUACUUGGAGATT; h*Myd88*-R: UCUCCAAGUACU CAAAGUCTT; h*Trif*-F: CGCCATAGACCACTCAGCTT; h*Trif*-R: TGAGG TTGCTCATCATGGCTT; Relative quantification is determined using the $2^{-\Delta\Delta Ct}$ method.

## Inhibitor intervention

The HBE cells were randomly divided into control, LPS intervention and GSK'872 or Necrodtatin-1 intervention group. GSK'872 (Selleck, S8465) and Necrodtatin-1 (Selleck, S8037) was dissolved in DMSO at a concentration of 50 mM and diluted to 5 μM and 10 μM in 1640 medium containing 200 μg/ml LPS and 10% FBS for intervention group. Cells were treated with 5 μM GSK'872 or 10 μM Necrodtatin-1 for 48 h and were subsequently collected for the next experiments.

## Western blotting

To extract total protein, HBE cells and lung from mice were lysed using RIPA buffer (Sparkjade, EA0002) containing protease inhibitor (Beyotime, ST507-10ml) and phosphatase inhibitor (Biosharp, BL615A) and quantified using the BCA Protein Quantification Kit (Yamei, ZJ101). The concentration of protein stock solution was adjusted to 3ug/ul after fully denaturing the protein using loading buffer (Fude Biological, FD002). The sample was separated by 10% sodium dodecyl sulfate-polyacrylamide gel electrophoresis (SDS-PAGE) and transferred onto 0.45 μm polyvinylidene difluoride (PVDF) membranes (Merck, IPVH00010). The membranes were blocked with 5% FBS and incubated with primary antibodies to RIPK1 (CST, #3493), p-RIPK1 (Affinity, af2398), RIPK3 (Abcam, ab62344), p-RIPK3 (Abcam, ab209384), MLKL (Merck, SAB5700808), p-MLKL (Abcam, ab196436 and Affinity, AF7420), Caspase-8 (Proteintech, 13423-1-AP), Caspase-3 (Abcam,ab184787), Caspase-7 (Abcam,ab255818). Then the membranes were incubated with HRP-conjugated Affinipure Goat Anti-Mouse IgG (H + L) (Proteintech, SA00001-1) and HRP-conjugated Affinipure Goat Anti-Rabbit IgG (H + L) (Proteintech, SA00001-2). The antibody to β-Actin (Sigma, A2228), GAPDH (Abcam, ab8245) were used as internal control for protein quantification. Band intensity was quantified using a chemiluminescence imaging system (BIO-RAD) and Image J software.

## Co-immunoprecipitation

HBE were cultured in 1640 medium (Gibco, C11875500BT) containing 10% FBS (Meilunbio) and 200ug/ml LPS in 10 cm culture dishes. After 48 h, cells were washed with cold PBS and fully lysed using IP lysis solution (Beyotime, P0013) containing protease inhibitor (Beyotime, ST507-10ml) and phosphatase inhibitor (Biosharp, BL615A). After centrifuged at 13000×g for 15 min at 4 °C, per 750ug protein was added with 25 μL Pierce™ protein A/G Agarose (Thermofisher, 88802) and IgG (CST, 3900S) or p-MLKL (Abcam, ab196436 and Affinity, AF7420) or p-RIPK1 (Affinity, af2398) or p-RIPK3 (Abcam, ab209384) overnight at 4 °C. After 24 h, the immunoprecipitants were washed five times and boiled at 95 °C for 5 min with 2 × SDS Loading Buffer. Western blotting assay was conducted according to the protocol above.

## Hematoxylin-eosin staining

Hematoxylin-eosin (HE) staining was performed to explore inflammation and damage in the lung tissue. Prior to histological analysis, lungs were immersed in 4% paraformaldehyde for 24 h. Paraffin-embedded Section (3 μm thickness) were prepared and subjected to HE staining. The sections were placed in aqueous hematoxylin solution for 3-5 min and 1% eosin staining solution for 2 sec. Then, they were washed with running water for 1-2 min, dehydrated in alcohol for 1-2 min. Finally, sections were treated with clarification and fixation. Light microscopy images were obtained using an microscope (Olympus, CX-31). The degree of congestion, edema, exudation, and tissue damage in the lung tissue could be evaluated.

## Lung injury score

The scoring system considered five factors to assess the extent of pathological injury, including alveolar and interstitial inflammation.alveolar and interstitial hemorrhage, edema, alveolar fusion, and alveolar septal thickening. Each of these indicators was assigned a severity grade based on the following scale: no damage (0 points),25% damage (2.5 point), 50% damage (5 points), 75% damage (7.5 points), and diffuse damage (10 points). The final score was obtained by summing the scores of various factors and dividing the total by 5. The final score ranged from 0 to 10 points, with 0 representing no injury and 10 representing the highest level of injury[73].

## Lung immunofluorescence

Lung samples from mice were treated following a previously protocol (The animal model preparation protocol is the same as described in the "*In vivo*

*animal experiments*" section, and the sample paraffin embedding and sectioning methods are the same as in the HE staining section). The sectioned coronally at a thickness of 3 μm using a microtome, and then deparaffinized with a series of xylene and graded ethanol washes. The slides were heated for 90 s at 100 ℃ in pressure cooker. Endogenous peroxidase activity was blocked by incubating the sections in 3% H$_2$O$_2$ for 30 min. After brief TBS washes, the sections were incubated in blocking solution (10% normal goat serum) for 30 min. The sections were incubated overnight at 4 ℃ with anti-p-RIPK1 (Affinity, af2398) diluted in TBS. After washing three times for 5 min in TBST, the slides were incubated with Goat Anti-rabbit IgG (HRP) (Abcam, ab205718) for 45 min at 37 ℃, washed three times with TBST and then subjected to signal amplification using iFluor® 488 tyramide (Aatbio, 11060) for 10 min at room temperature. Following three washes in TBST, the slices were subjected to microwave treatment in citrate buffer pH 6.0, (1 h in 37 ℃). The sections were incubated in blocking solution (10% normal goat serum) for 30 min. The slices were subjected to staining using anti-p-RIPK3 (Abcam, ab209384) or anti-p-MLKL (Abcam, ab196436), employing the same staining procedure as described for p-RIPK1 staining but using Cy3 tyramide (Aatbio, 11065) and Cy5 tyramide (Aatbio, 11066). The slides were treated with DAPI (Vector Laboratories) and mounting medium (Southern Biotech CAT NO 0100-01). Images were taken on fluorescence microscope (Olympus#BX53) and processed by CaseViewer (2.4). The average fluorescence intensity was calculated using Image J.

### Lung immunohistochemistry
Lung tissue sections were obtained using the same method as above protocol (The animal model preparation protocol is the same as described in the "*In vivo animal experiments*" section, and the sample paraffin embedding and sectioning methods are the same as in the HE staining section). Tissue sections were stained with anti-E-cadherin (Proteintech, 20874-1-AP) overnight at 4 ℃. Goat Anti-rabbit IgG (HRP) (Abcam, ab205718) was used as secondary antibody. After the final staining, the samples were scanned using fluorescence microscope (Olympus, CX-31) and processed by CaseViewer (2.4).

### Bronchoalveolar lavage fluid collection (BALF)
Administer an intraperitoneal injection of 1% pentobarbital to the mice to achieve an overdose of anesthesia and death. Expose the trachea and both lungs, and after completely ligating the right lung, make a "V"-shaped incision on the trachea. Carefully insert a 20 mL syringe needle into the trachea through the incision, secure it with a cotton thread, and inflate the left lung with 0.4 mL of PBS. With draw the PBS and collect it in a 1.5 mL Eppendorf tube. Repeat the above steps three times (using a total of 1.2 mL of PBS to inflate the lung tissue), and store the collected lavage fluid at 4 ℃.

### Survival curve
According to the " *In vivo animal experiments* " section, mice were modeled, and at 0, 24, 48, 72… hours post-modeling, each mouse was weighed and deaths were recorded. Subsequently, a survival curve was plotted. During the recording period, in accordance with animal ethics requirements and humanitarian principles, until the end of the observation, the weight of the mice was calculated daily. If the weight loss exceeded 20% of the pre-modeling weight, the mice were euthanized, and the deaths were recorded on the same day[74–76].

### Cell immunofluorescence
HBE from LPS, Nec-1, GSK'872 and control were treated following a previously protocol (The cell model preparation protocol is the same as described in the "*Cell Culture and LPS Stimulation*" section). The cell was fixed in 4% paraformaldehyde and treated with 0.1% TritonX-100 for membrane permeabilization. After three washes in TBS, endogenous peroxidase activity was blocked by incubating the cell in 3% H$_2$O$_2$ for 10 min. The cells were incubated in blocking solution (10% normal goat serum) for 30 min, and subsequently were incubated overnight at 4 ℃ with anti-p-RIPK1 (Affinity, af2398) diluted in TBS. After washing three times for 5 min

in TBST, the slices were incubated with Goat Anti-rabbit IgG(HRP) (Abcam, ab205718) for 45 min at 37 ℃, washed three times with TBST and then subjected to signal amplification using iFluor® 488 tyramide (Aatbio, 11060) for 10 min at room temperature. Following three washes in TBST, the slices were subjected to microwave treatment in citrate buffer pH 6.0, (1 h in 37 ℃). The slices were subjected to staining using anti-p-RIPK3 (abcam, ab209384) or anti-p-MLKL (Affinity, af7420), employing the same staining procedure as described for p-RIPK1 staining but using Cy3 tyramide (Aatbio, 11065) and Cy5 tyramide (Aatbio, 11066). The slices were stained with DAPI (Vector Laboratories) and mounting medium (Southern Biotech CAT NO 0100-01). Images were taken on fluorescence microscope (Olympus#BX53) and processed by CaseViewer (2.4). The average fluorescence intensity was calculated using Image J.

MLE cells were divided into four groups: untreated, treated with 200 μg/ml LPS for 48 h, treated with 200 μg/ml LPS and 1 μM GW806742X for 48 h, and treated with 200 μg/ml LPS and 1 μM SPC-M-GW for 48 h[41]. These cells were then fixed, permeabilised, and blocked using 4% paraformaldehyde, 0.5% Triton X-100, and 3% BSA, respectively. Cells were then incubated at room temperature for 3 h with anti-SPC (Affinity, DF6647, 1:500) or MLKL (Merck, SAB5700808, 1:200) and Tubulin (Abcam, ab6160, 1:1000). Following this, cells were incubated for 1 h at room temperature with Multi-rAb CoraLite® Plus 594-Goat Anti-Rabbit Recombinant Secondary Antibody (H + L) (Proteintech, RGAR004) or Goat anti-Rat IgG (H + L) Cross-Adsorbed Secondary Antibody, Alexa Fluor™ 594 (Thermofisher, A-11007) and Goat anti-Rabbit IgG (H + L) Cross-Adsorbed Secondary Antibody, Alexa Fluor™ 488 (Thermofisher, A-11008). The nucleus were stained with SlowFade™ Diamond Antifade Mountant with DAPI (Thermofisher, S36964) and the samples were mounted. After incubating for 10 min at room temperature, images were captured using an fluorescence microscope (Leica, DM5500) or a laser confocal microscope (Leica, STELLARIS5).

### Cell viability assay
200 μg/ml LPS, 10 μM GSK'872, or 100 μM Nec-1 were added to the HBE culture medium, and cells were cultured for 24 or 48 h. Cell viability was then measured using the Cell Count Kit-8 (CCK-8, UElandy, C6005L) at 450 nm under white light absorbance, following the manufacturer's instructions.

### Protein-protein docking
The 3D structures of RIPK1(6nw2) protein[77]and RIPK3-MLKL (7mon) protein[78] complex and their related information were obtained by searching the PDB database[79–81]. The local files of the two 3D structures were uploaded to the Cluspro website, and the protein docking results containing 10 docking models with Job Details of 991258 were obtained, and their docking model scores were compared, and the model with the lowest energy value and the lowest central energy value were selected[82–85]. The docking keys were visualised using PyMOL (Anaconda 2, 64-bit).

### Bulk RNA-sequencing
C57BL/6 mice were treated as the method described in the "*In vivo animal experiments*" section, and after obtaining lung tissues, RNA was extracted for transcriptome sequencing. The data presented in the study are deposited in the GEO-NCBI repository, accession numbers GSE263867. Raw reads were obtained using the BGISEQ platform and filtered to remove low-quality reads, adapter contamination, and an unknown base N content. Then, clean reads were then mapped to gene symbols obtained from the National Center for Biotechnology Information (NCBI) database. "DESeq2" package in R (version 4.2.0) was used to identify differentially expressed genes (DEGs) and their values. Genes related to the Necroptosis pathway were obtained from the KEGG database. Heatmaps were generated by intersecting the Necroptosis pathway genes with the DEGs and plotting their FPKM values. Volcano plots were generated using the "ggplot2" package to visualize DEGs with log2 Fold Change < -1 or >1 and *p*.value < 0.05. Key genes in the necroptosis pathway were fluorescently

labeled. GSEA enrichment analysis of differential genes was performed using "clusterProfiler" and "ReactomePA" package, and the data of the pathway named necroptosis was extracted and visualised using the gseaplot2 function to obtain a GSEA biplot, while the pathway related to death was extracted and plotted as a bolliplot.

### Statistics and reproducibility

Data were presented as mean ± standard deviation of minimum three replicates unless otherwise indicated[86]. The data were analyzed with either Student's t-test (two-tailed) or one-way ANOVA followed by Tukey's post hoc test using GraphPad Prism 8 software (GraphPad Software, La Jolla, CA, USA). The bar charts are generated using the mean values, with a 95% confidence interval. The error bars represent the standard deviation (SD). Differences were considered statistically signicant at $P < 0.05$. The reported values are from ≥3 separate experiments. For Bulk RNA sequencing data, statistical analyses were described as above.

### Reporting summary

Further information on research design is available in the Nature Portfolio Reporting Summary linked to this article.

### Data availability

The data that support the findings of this study are available from the corresponding author upon reasonable request. The RNA-sequencing data presented in the study are deposited in the GEO-NCBI repository, accession numbers GSE263867.

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

## Acknowledgements

This work was supported by the National Key Research and Development Program of China (2021YFC2501800), the National Natural Science Foundation of China (82172163, 82272182, 82372185, 82302485), the Zhejiang Provincial Natural Science Foundation of China (LMS25H150002, LQN25H150005), the Zhejiang Provincial Traditional Chinese Medicine Science and Technology Plan (2022ZQ055). We thank Ms Yuqiong Xie, Ms Meirong Yu, Ms Yonglan Zhu, Ms Amin Liu, Ms Yuelan Chen, Clinical Research Center, The Second Affiliated Hospital of Zhejiang University for laboratory management and technical support.

## Author contributions

T.B.P. contributed to the overall conceptualization and design of the research, D.Y. was responsible for designing experiments related to pharmaceutical materials. K.Z.Y., Z.Q.C., H.Q.Y. and Y.M.J. carried out the animal experiments, in vitro cell experiments and bioinformatics analysis. T.B.P., D.Y., Z.Z.C., X.N.X., and K.Z.Y. analyzed the data and provided explanations. K.Z.Y., X.N.X. and T.B.P. prepared the figures and drafted the manuscript. Z.Z.C., C.W., Z.G.S. and X.N.X. performed the critical reading of the manuscript. T.B.P. and Z.Z.C. edited and revised the manuscript. All authors have read and agreed to the published version of the manuscript.

## Competing interests

The authors declare no competing interests.
