## [Transparent Peer Review file · Communications Biology]

Targeting Alveolar Epithelial Cells with Lipid Micelle-Encapsulated Necroptosis Inhibitors to Alleviate Acute Lung Injury

Corresponding Author: Dr Baoping Tian

This manuscript has been previously reviewed at another journal. This document only contains information relating to versions considered at Communications Biology.

Version 0:

Reviewer comments:

Reviewer #2

(Remarks to the Author)

This manuscript, "Lipid micelle-encapsulated necroptosis inhibitor targets epithelial cells to effectively alleviate acute airway inflammatory injury", comprehensively describes an investigation into the cell signaling pathways that underpin acute lung injury (ALI) and acute respiratory distress syndrome (ARDS). Specifically, the authors provide fundamental insights into the precise characteristics and mechanisms of necroptosis, a cell death pathway implicated in ALI and identified the RIPK1/RIPK3/MLKL complex. This identified a new drug target for the treatment of ALI.

Finally, the authors exploited this knowledge to design and prepare a lipid-based nanomedicine table to deliver a drug targeted to MLKL. The drug of choice was GW806742X, a MLKL inhibitor, while targeting of ATII cells was achieved through the surface modification of the lipid particle with SPC antibodies that can target ATII cells.

Comments on the manuscript: The manuscript was excellently prepared and edited with only minor errata which I have marked in the uploaded PDF

This work would find a wide readership and interest in Communications Biology. In light of my minor comments above I recommend this manuscript for publication after the minor comments indicated are fixed.

It is excellent work scientifically, presented in a highly polished manuscript and I very much enjoyed reading it.

General comments:

1. I am not a pharmacologist but the biological experiments were clearly described and the results were well presented. In my humble opinion the data supports the conclusions of the manuscript.
2. The use of colours in the structures of the lipid components may not translate well.
3. Some of the figures are crowded but I expect these will improve when the proofs are generated.
4. The animal ethics approval number should be cited clearly in the manuscript.

Reviewer #3

(Remarks to the Author)

Overview: Acute lung injury (ALI) is an inflammatory disease mediated by multiple factors. The authors propose that

necroptosis, a form of regulated cell death, is a significant driver of this disease. They conducted extensive experimental studies using an LPS-induced ALI mouse model to examine the role of necroptosis mediators, specifically the RIPK1/RIPK3/MLKL complex, in ALI. Subsequently, they explored using the MLKL inhibitor GW806742X to inhibit or reverse disease progression. While demonstrating necroptosis involvement in ALI/ARDS is an exciting finding, many concerns with the study diminished enthusiasm for the work.

Major Concerns:

1. Incremental advance:

This study represents an incremental advance in the field of ALI. Previous research has already demonstrated the involvement of necroptosis in pulmonary diseases such as ALI (e.g., Wang L, et al., Necroptosis in Pulmonary Diseases: A New Therapeutic Target. *Front Pharmacol.* 2021, 12:737129. doi: 10.3389/fphar.2021.737129; Ning and Qiao, The role of necroptosis in common respiratory diseases in children. *Front Pediatr.* 2022, 10:945175. doi: 10.3389/fped.2022.9451752022). Surprisingly, these articles were not cited, and we hope this omission was not intentional. Therefore, the claim "We discovered that necroptosis mediates the progression of ALI through the activation and formation of the RIPK1/RIPK3/MLKL complex" is not novel. Additionally, the use of GW806742X for treating ALI has been previously reported, although not in nanoparticle formulations.

2. Alternative cell death pathways:

The study did not test alternative cell death pathways. Describing apoptosis and necroptosis as regulated and non-regulated forms of cell death, respectively, is incorrect; both are regulated. The manuscript does not mention necrosis, a non-regulated form of cell death with similar characteristics to necroptosis. Without distinguishing between apoptosis, necrosis, and necroptosis, attributing the observed effects solely to necroptosis does not paint a complete picture of the complexity of ALI. The authors may have missed an opportunity to leverage these complementary mechanisms for a more comprehensive treatment strategy for ALI.

3. Necroptosis induction:

It is unclear if necroptosis is inherently present in ALI or if LPS itself induces necroptosis. Many studies have shown that LPS can induce necroptosis. Using non-LPS-induced ALI models would help demonstrate that LPS is not the primary driver of the observed necroptosis. Perhaps staining human tissue from ALI or ARDS patients could validate necroptosis's role in these diseases.

4. Comparison with standard therapies:

The authors did not discuss existing ALI treatments and their associated limitations that motivated this study. How does GW806742X compare with standard therapies? Does inhibiting MLKL activation offer better outcomes than current treatments?

5. Physiological relevance:

The study aims to "mitigate lung injury, improve gas exchange, and ultimately improve the outcomes of patients with ARDS," but no physiological tests address these points. The authors could demonstrate improvements in gas exchange and survival in ALI mice treated with necroptosis inhibitors compared to standard therapies.

Minor Comments:

1. Introduction:

The introduction reads like a review article. The authors can significantly shorten this section, focusing on current findings relating ALI/ARDS to necroptosis and associated treatment models, including standard therapies and their deficiencies. This will clarify the novelty and impact of the current study.

2. Figures and images:

- o Please include a bright field image for topography visualization in Figures 3J and S4.
- o For Figure 5D, include original images from which the zoomed cells were derived, along with a scale bar, bright field, and untreated cell images to help understand protein distribution before and after LPS treatment.

3. Abbreviations and clarity:

- o Define "SPC" upon first use.
- o Clarify the sentence on page 17: "The lung tissues were obtained in animal experiences for PCR assay."

4. Reproducibility:

Review all experimental methods to ensure they are reproducible. For example, "The next day, the immunoprecipitants were washed five times and boiled with 2 × SDS Loading Buffer. "next day" is not precise, and "boiled" without any indication of duration and temperature is highly subjective. Please specify the precise timing and temperature for boiling immunoprecipitants (page 19).

5. Relevance of Procedures:

- o Ensure all described procedures are relevant to the study. Some procedures described in the manuscripts may not be relevant to the current study. For example, "Goat Anti-rabbit IgG (HRP) (Abcam, ab205718), FTIC (aatbio, 11060), CY3 (aatbio, 11065) and CY5 (aatbio, 11066) were used as fluorescent labelling" (page 19). First, what is FTIC? Do you mean FITC? Second, where are the images obtained with "FTIC" and CY3? Please clarify the usage of fluorescent labeling throughout the manuscript and include images obtained with FITC and CY3 if mentioned.
- o Clarify some experimental procedures. For example, on page 19, the authors state that "Hematoxylin-eosin (HE) staining was performed to explore damage in the lung tissue. Prior to histological analysis, lungs were immersed in 4% paraformaldehyde for 24 hours." Was this H&E staining applicable to injured lungs? If yes, please briefly explain why the permeation and fixation steps were performed at the end. Include Eosin staining time in the protocol.

6. Literature citation:

- o On page 7, the authors acknowledged that "necroptosis is present in ALI" but omitted citing the relevant literature (page 7).
- o There are many other instances with similar issues in the experimental section. For example, "Tissue Immunofluorescence: Samples from both ALI and control were treated following a previously protocol" (page 19). Please cite all previous methods mentioned in the manuscript.

7. Figures:

- o In Figure 5, "The results showed that p-RIPK1, p-RIPK3, and p-MLKL could be used as targets to precipitate the other two proteins, indicating their interaction or complex formation" (page 22). What are the "other two proteins"?
- o Figure 7: Please provide clearer zoomed-in images to display infiltrating cells more distinctly; this will facilitate determining the cell types involved in ALI.

8. Spelling and Grammar:

- o Multiple spelling and grammatical errors in the manuscript need correction. Examples include:
 - Alleviates → Alleviate (Title page)
 - necroptosis → necroptosis (page 4)
 - Danger-associated molecular patterns → Damage-associated molecular patterns (page 4)
 - Remove double periods (..) after "Figure 1B" (page 6)
 - necrodtatin-1 → necrostatin-1 (pages 8, three places on page 18)
 - PIPK1 → RIPK1 (page 22)

Reviewer #4

(Remarks to the Author)

Fig 2. There is no mention of the timepoints in the results section or in the figure legend.

Figure 2J- quantification is missing across multiple animals as well as multiple images per animal.

Fig 3. Details regarding the HBE cells used are lacking from the methods. Are these primary cells? Obtained from where? How were they grown eg in ALI? Can the authors show the cells undergo cell lysis and/or cell death?

Figure 3J- quantification is missing across multiple images.

Fig 4 – why is RIPK1 no longer expressed in nec1 treated cells. They should still express total RIPK1. Are the inhibitors toxic? Can the authors show toxicity isn't an issue? Inhibitor alone controls are missing.

Fig 6 – can the authors demonstrate TRIF and Myd88 knockdown?

Fig 7 – analysis of lung injury is very limited to HandE sections. More detailed analysis is warranted. A dose response would also be useful The methods lacks details regarding how lung injury was scored. Eg what does 1-9 mean? Was it done blindly? And by how many readers?

Fig 7 – critical data showing treatment inhibits MLKL is missing. Also details as to how mice were treated e.g. the route of administration is lacking.

Details regarding the statistical analysis tests are missing from the legends.

Overall, the methods and figure legends lack detail, making it difficult to assess the data and the conclusions.

The introduction is very long, almost 3 pages in length.

Version 1:

Reviewer comments:

Reviewer #3

(Remarks to the Author)

Reviewer #4

(Remarks to the Author)

The authors have not adequately addressed the following comments:

Fig 7 – analysis of lung injury is very limited to HandE sections. More detailed analysis is warranted.

Fig 7 – critical data showing SPC-M-GW treatment inhibits MLKL in vivo is missing. This data is very critical with regard to validating the conclusions as there is currently no evidence it does inhibit MLKL.

Please note, Figure S6 to S9 are incorrectly labelled in the figure legends.

Version 2:

Reviewer comments:

Reviewer #4

(Remarks to the Author)

Zhejiang University

Department of Critical Care Medicine of the Second Affiliated Hospital
Zhejiang University School of Medicine, Hangzhou, China

25 November 2024

Re: Resubmission of Revised Manuscript (COMMSBIO-24-3412-T)

Dear Reviewers,

We are pleased to resubmit our revised manuscript, titled “**Targeting Alveolar Epithelial Cells with Lipid Micelle-Encapsulated Necroptosis Inhibitors to Alleviate Acute Lung Injury,**” for consideration for publication in *Communications Biology*. We sincerely appreciate the thoughtful and constructive feedback provided by you, which has greatly contributed to the improvement of our study.

As detailed in our Response to Comments document, we have provided comprehensive replies to each question of yours. Corresponding changes have been made in the revised manuscript, where modifications are highlighted in blue text for clarity.

We believe these modifications have substantially improved the rigor and clarity of our study, and we are confident that the revised manuscript now meets the high standards of *Communications Biology*. Thank you again for your time and consideration.

Sincerely yours,

Bao-ping Tian, M.D., Ph.D.

Dept. of Critical Care Medicine, The Second Affiliated Hospital, Zhejiang University School of Medicine

Email: TianBP@zju.edu.cn

Reviewer #1 (Remarks to the Author):

This manuscript, “Lipid micelle-encapsulated necroptosis inhibitor targets epithelial cells to effectively alleviate acute airway inflammatory injury”, comprehensively describes an investigation into the cell signalling pathways that underpin acute lung injury (ALI) and acute respiratory distress syndrome (ARDS). Specifically, the authors provide fundamental insights into the precise characteristics and mechanisms of necroptosis, a cell death pathway implicated in ALI and identified the RIPK1/RIPK3/MLKL complex. This identified a new drug target for the treatment of ALI.

Finally, the authors exploited this knowledge to design and prepare a lipid-based nanomedicine table to deliver a drug targeted to MKL. The drug of choice was GW806742X, a MLKL inhibitor, while targeting of ATII cells was achieved through the surface modification of the lipid particle with SPC antibodies that can target ATII cells.

Comments on the manuscript: The manuscript was excellently prepared and edited with only minor errata which I have marked in the uploaded PDF

This work would find a wide readership and interest in Communications Biology. In light of my minor comments above I recommend this manuscript for publication after the minor comments indicated are fixed.

It is excellent work scientifically, presented in a highly polished manuscript and I very much enjoyed reading it.

RESPONSE:

Zhejiang University

Department of Critical Care Medicine of the Second Affiliated Hospital
Zhejiang University School of Medicine, Hangzhou, China

We appreciate the insightful comments from your review. We have carefully revised and extensively edited our manuscript, making significant updates to the content. These modifications have been implemented in the revised manuscript, with changes highlighted in blue font for clarity. We hope these improvements enhance readability and effectively convey our findings to you and your readers.

General comments:

1. I am not a pharmacologist but the biological experiments were clearly described and the results were well presented. In my humble opinion the data supports the conclusions of the manuscript.

RESPONSE:

We sincerely appreciate your thorough review of our manuscript and the valuable concluding remarks you provided.

2. The use of colours in the structures of the lipid components may not translate well.

RESPONSE:

Thank you for your valuable suggestion. We apologize for the weak CY5 fluorescence signal in our original immunofluorescence images, which resulted from the strong oxidative effects and subsequent quenching of fluorescent molecules during sample preparation. In response, we have optimized the experimental protocol, repeated the co-localization immunofluorescence staining, and updated the images accordingly. The revised images can be found in **Figure 7E** of the original text. We hope these improvements enhance the visual experience for you and readers.

Zhejiang University

Department of Critical Care Medicine of the Second Affiliated Hospital
Zhejiang University School of Medicine, Hangzhou, China

3. Some of the figures are crowded but I expect these will improve when the proofs are generated.

RESPONSE:

We sincerely appreciate your feedback. Indeed, the organization of our figures needed improvement. In the re-submitted figures of the revised manuscript, including **Figure 1-8 and Figure S1-S9**, we have carefully reorganized and arranged the figures to enhance both readability and visual appeal.

4. The animal ethics approval number should be cited clearly in the manuscript.

RESPONSE:

Thank you for bringing this issue to our attention. In our study, all animal experiments were approved by the Ethics Review Committee of the Second Affiliated Hospital of Zhejiang University School of Medicine (Approval No. 2022.050). This approval has now been included in the relevant methodology section of the revised manuscript, as shown in **Page 13, Line 368-370**.

Reviewer #2 (Remarks to the Author):

Overview: Acute lung injury (ALI) is an inflammatory disease mediated by multiple factors. The authors propose that necroptosis, a form of regulated cell death, is a significant driver of this disease. They conducted extensive experimental studies using an LPS-induced ALI mouse model to examine the role of necroptosis mediators, specifically the RIPK1/RIPK3/MLKL complex, in ALI. Subsequently, they explored using the MLKL inhibitor GW806742X to inhibit or reverse disease progression. While demonstrating necroptosis involvement in ALI/ARDS is an exciting finding, many concerns with the study diminished enthusiasm for the work.

RESPONSE:

Thank you very much for your valuable suggestions and feedback on our work. As the referee mentioned, previous studies have shown that necroptosis is involved in the pathological process of ARDS. However, there is a lack of targeted therapeutic methods or strategies based on this mechanism. In this study, we demonstrated the activation of p-RIPK1, p-RIPK3, and p-MLKL, as well as the formation of their complexes in LPS-induced acute lung injury. Additionally, we highlighted the involvement of MyD88 and TRIF in the necroptotic signalling pathway within the pathological process of this model. Furthermore, we developed a drug that inhibits MLKL oligomerization and its translocation to the membrane, allowing for specific targeting of alveolar epithelial type II cells. This method reduces damage and inflammatory progression during lung injury, offering a novel strategy for the treatment of acute lung injury or acute respiratory distress syndrome. Based on your suggestions, we have made substantial revisions and added additional experiments to address your comments, and we hope that these changes meet your expectations and the requirements of your journal. Please see the detailed responses below.

Major Concerns:

1. Incremental advance:

This study represents an incremental advance in the field of ALI. Previous research has already demonstrated the involvement of necroptosis in pulmonary diseases such as ALI (e.g., Wang L, et al., Necroptosis in Pulmonary Diseases: A New Therapeutic Target. *Front Pharmacol.* 2021, 12:737129. doi: 10.3389/fphar.2021.737129; Ning and Qiao, The role of necroptosis in common respiratory diseases in children. *Front Pediatr.* 2022, 10:945175. doi: 10.3389/fped.2022.9451752022). Surprisingly, these articles were not cited, and we hope this omission was not intentional. Therefore, the claim "We discovered that necroptosis mediates the progression of ALI through the activation and formation of the RIPK1/RIPK3/MLKL complex" is not novel. Additionally, the use of GW806742X for treating ALI has been previously reported, although not in nanoparticle formulations.

RESPONSE:

We sincerely appreciate your feedback. As cited by the referee, necroptosis has indeed been reported to be involved in the pathological process of ALI. Due to our oversight, this reference was not included in the original manuscript. We have now incorporated the relevant literature and discussed it in the revised manuscript, as shown on **Page 6 Line 141-146**, and in **References 29, 30**.

Currently, there is a lack of studies that experimentally demonstrate the presence of p-RIPK1, p-RIPK3, and p-MLKL complexes in ALI-associated necroptosis. We have provided evidence for the existence of these complexes, which are described in detail on **Page 7-8 Line 185-210** ("**2.5 The co-reaction between RIPK1, RIPK3 and MLKL in LPS induced Necroptosis**" in the results) of the revised manuscript. It is important to note that AT II cells are responsible for synthesizing and secreting surfactants and endogenous antimicrobial peptides.

They also possess significant regenerative capacity, as they can differentiate into AT I cells, thereby promoting the repair of damaged lung tissue (Ruaro, Salton et al. 2021). Therefore, we focus on alleviating AT II cell damage by targeting the inhibition of necroptosis within these cells and aim to reduce AT II cell damage, thereby preventing the progression of inflammation in ARDS. We have demonstrated that this treatment effectively reduces inflammatory cytokines, mitigates the inflammatory response, and improves mouse survival outcomes. These results are discussed and analyzed in detail on **Page 10 Line 262-281** ("*2.7. Alveolar type II-targeted delivery of GW806742X via SPC antibody-modified lipid micelle*" in the results) of the manuscript.

The effect of GW806742X in treating ALI by inhibiting necroptosis has not been reported. The final and most critical step of necroptosis is the oligomerization of MLKL and its translocation to the cell membrane, which is why we chose MLKL as an inhibition target (Wang, Sun et al. 2014, Petrie, Birkinshaw et al. 2020), a process that GW806742X effectively prevents (Hildebrand, Tanzer et al. 2014, González-Juarbe, Gilley et al. 2015, Zhan, Jeon et al. 2022, Sun, Revach et al. 2023, Liu, Lu et al. 2024). By modifying GW806742X into a lipid-encapsulated nanomedicine, it achieves targeted delivery, reduces side effects, and optimizes therapeutic effects. Ultimately, this lipid-encapsulated nanomedicine targets AT II cells to inhibit their necroptosis, reducing the aggregation of alveolar inflammatory cells, the secretion of inflammatory factors, and the destruction of alveolar structure, thereby improving the survival rate of ALI mice. The relevant treatment effects are presented in **Figure 7**, with detailed descriptions provided on **Page 10 Line 264-281** of the revised manuscript.

Gong, T., X. Zhang, Z. Peng, Y. Ye, R. Liu, Y. Yang, Z. Chen, Z. Zhang, H. Hu, S. Yin, Y. Xu, J. Tang and Y. Liu (2022). "Macrophage-derived exosomal aminopeptidase N aggravates sepsis-induced acute lung injury by regulating necroptosis of lung epithelial cell." *Commun Biol* **5**(1): 543.

Zhejiang University

Department of Critical Care Medicine of the Second Affiliated Hospital
Zhejiang University School of Medicine, Hangzhou, China

González-Juarbe, N., R. P. Gilley, C. A. Hinojosa, K. M. Bradley, A. Kamei, G. Gao, P. H. Dube, M. A. Bergman and C. J. Orihuela (2015). "Pore-Forming Toxins Induce Macrophage Necroptosis during Acute Bacterial Pneumonia." *PLoS Pathog* **11**(12): e1005337.

Hildebrand, J. M., M. C. Tanzer, I. S. Lucet, S. N. Young, S. K. Spall, P. Sharma, C. Pierotti, J. M. Garnier, R. C. Dobson, A. I. Webb, A. Tripaydonis, J. J. Babon, M. D. Mulcair, M. J. Scanlon, W. S. Alexander, A. F. Wilks, P. E. Czabotar, G. Lessene, J. M. Murphy and J. Silke (2014). "Activation of the pseudokinase MLKL unleashes the four-helix bundle domain to induce membrane localization and necroptotic cell death." *Proc Natl Acad Sci U S A* **111**(42): 15072-15077.

Huang, L., X. Tan, W. Xuan, Q. Luo, L. Xie, Y. Xi, R. Li, L. Li, F. Li, M. Zhao, Y. Jiang and X. Wu (2024). "Ficolin-A/2 Aggravates Severe Lung Injury through Neutrophil Extracellular Traps Mediated by Gasdermin D-Induced Pyroptosis." *Am J Pathol* **194**(6): 989-1006.

Li, Y., Y. Liu, X. Yao, L. Zhu, L. Yang and Q. Zhan (2024). "MLKL Protects Pulmonary Endothelial Cells in Acute Lung Injury." *Am J Respir Cell Mol Biol* **70**(4): 295-307.

Liu, M., J. Lu, J. Hu, Y. Chen, X. Deng, J. Wang, S. Zhang, J. Guo, W. Li and S. Guan (2024). "Sodium sulfite triggered hepatic apoptosis, necroptosis, and pyroptosis by inducing mitochondrial damage in mice and AML-12 cells." *J Hazard Mater* **467**: 133719.

Petrie, E. J., R. W. Birkinshaw, A. Koide, E. Denbaum, J. M. Hildebrand, S. E. Garnish, K. A. Davies, J. J. Sandow, A. L. Samson, X. Gavin, C. Fitzgibbon, S. N. Young, P. J. Hennessy, P. P. C. Smith, A. I. Webb, P. E. Czabotar, S. Koide and J. M. Murphy (2020). "Identification of MLKL membrane translocation as a checkpoint in necroptotic cell death using Monobodies." *Proc Natl Acad Sci U S A* **117**(15): 8468-8475.

Ruaro, B., F. Salton, L. Braga, B. Wade, P. Confalonieri, M. C. Volpe, E. Baratella, S. Maiocchi and M. Confalonieri (2021). "The History and Mystery of Alveolar Epithelial Type II Cells: Focus on Their Physiologic and Pathologic Role in Lung." *Int J Mol Sci* **22**(5).

Sun, Y., O. Y. Revach, S. Anderson, E. A. Kessler, C. H. Wolfe, A. Jenney, C. E. Mills, E. J. Robitschek, T. G. R. Davis, S. Kim, A. Fu, X. Ma, J. Gwee, P. Tiwari, P. P. Du, P. Sindurakar, J. Tian, A. Mehta, A. M. Schneider, K. Yizhak, M. Sade-Feldman, T. LaSalle, T. Sharova, H. Xie, S. Liu, W. A. Michaud, R. Saad-Beretta, K. B. Yates, A. Iracheta-Vellve, J. K. E. Spetz, X. Qin, K. A. Sarosiek, G. Zhang, J. W. Kim, M. Y. Su, A. M. Cicerchia, M. Q. Rasmussen, S. J. Klempner, D. Juric, S. I. Pai, D. M. Miller, A. Giobbie-Hurder, J. H. Chen, K. Pelka, D. T. Frederick, S. Stinson, E. Ivanova, A. R. Aref, C. P. Paweletz, D. A. Barbie, D. R. Sen, D. E. Fisher, R. B. Corcoran, N. Hacohen, P. K. Sorger, K. T. Flaherty, G. M. Boland, R. T. Manguso and R. W. Jenkins (2023). "Targeting TBK1 to overcome resistance to cancer immunotherapy." *Nature* **615**(7950): 158-167.

Wang, H., L. Sun, L. Su, J. Rizo, L. Liu, L. F. Wang, F. S. Wang and X. Wang (2014). "Mixed lineage kinase domain-like protein MLKL causes necrotic membrane disruption upon phosphorylation by RIP3." *Mol Cell* **54**(1): 133-146.

Wen, H., W. Miao, B. Liu, S. Chen, J. S. Zhang, C. Chen and M. Y. Quan (2024). "SPAUTIN-1 alleviates LPS-induced acute lung injury by inhibiting NF- κ B pathway in neutrophils." *Int Immunopharmacol* **130**: 111741.

Yang, H. H., H. L. Jiang, J. H. Tao, C. Y. Zhang, J. B. Xiong, J. T. Yang, Y. B. Liu, W. J. Zhong, X. X. Guan, J. X. Duan, Y. F. Zhang, S. K. Liu, J. X. Jiang, Y. Zhou and C. X. Guan (2022). "Mitochondrial

Zhejiang University

Department of Critical Care Medicine of the Second Affiliated Hospital
Zhejiang University School of Medicine, Hangzhou, China

citrate accumulation drives alveolar epithelial cell necroptosis in lipopolysaccharide-induced acute lung injury." *Exp Mol Med* **54**(11): 2077-2091.

Yang, T., C. G. Xiang, X. H. Wang, Q. Q. Li, S. Y. Lei, K. R. Zhang, J. Ren, H. M. Lu, C. L. Feng and W. Tang (2024). "RIPK1 inhibitor ameliorates pulmonary injury by modulating the function of neutrophils and vascular endothelial cells." *Cell Death Discov* **10**(1): 152.

Zhan, Q., J. Jeon, Y. Li, Y. Huang, J. Xiong, Q. Wang, T. L. Xu, Y. Li, F. H. Ji, G. Du and M. X. Zhu (2022). "CAMK2/CaMKII activates MLKL in short-term starvation to facilitate autophagic flux." *Autophagy* **18**(4): 726-744.

2. Alternative cell death pathways:

The study did not test alternative cell death pathways. Describing apoptosis and necroptosis as regulated and non-regulated forms of cell death, respectively, is incorrect; both are regulated. The manuscript does not mention necrosis, a non-regulated form of cell death with similar characteristics to necroptosis. Without distinguishing between apoptosis, necrosis, and necroptosis, attributing the observed effects solely to necroptosis does not paint a complete picture of the complexity of ALI. The authors may have missed an opportunity to leverage these complementary mechanisms for a more comprehensive treatment strategy for ALI.

RESPONSE:

Thank you very much for pointing out this issue. Indeed, both apoptosis and necroptosis are tightly regulated processes, each involving a complex network of signaling pathways that govern cell fate decisions in response to various stressors. While apoptosis is typically characterized by controlled, programmed cell death, necroptosis represents a form of regulated necrosis that occurs when apoptotic pathways are compromised or inhibited. This form of necroptosis plays a crucial role in inflammation and tissue damage, particularly under pathological conditions. We have made the corresponding revisions in the manuscript, aiming for a more precise description, as shown in **Page 3 Line 67-74** (the second paragraph of the introduction). In the pathological process of ALI, multiple forms of programmed and non-programmed cell death events may

be involved. In our study, we established a mouse model of ALI induced by LPS airway instillation. RNA sequencing of lung tissue revealed activation of both apoptosis and necroptosis pathways, with key biomarkers such as *Ripk1*, *Ripk3*, *Mkl1*, *Aim2*, *Casp4*, *Casp3* showing increased expression (**Page 4-5 Line 104-109, Figure 1 and Figure S1B**). This result was further validated at the protein level (**Page 5 Line 128-133, Figure 2 and Figure S2**). While a variety of studies have reported the involvement of apoptosis in the pathogenesis of lung inflammation, such as ARDS (**Reference 15, 43, 45**), the detailed mechanisms of necroptosis remain poorly understood. Consequently, therapeutic strategies targeting necroptosis are still lacking in ALI/ARDS. Nevertheless, necrosis is a form of non-programmed cell death, and like necroptosis, its process involves inflammation and further tissue damage. During necrosis, the integrity of the cell membrane is compromised, leading to the leakage of cellular contents into the surrounding environment. This can trigger local inflammatory responses and further damage adjacent tissues. Given this characteristic of necrosis, we also performed enrichment analysis on pathways related to necrosis in lung tissue. Despite using various methods, we did not obtain obvious evidence of necrosis, as mentioned on **Page 5 Line 120-121** of the revised manuscript.

3. Necroptosis induction:

It is unclear if necroptosis is inherently present in ALI or if LPS itself induces necroptosis. Many studies have shown that LPS can induce necroptosis. Using non-LPS-induced ALI models would help demonstrate that LPS is not the primary driver of the observed necroptosis. Perhaps staining human tissue from ALI or ARDS patients could validate necroptosis's role in these diseases.

RESPONSE:

We extend our heartfelt appreciation to the reviewer for highlighting this issue. Airway instillation of lipopolysaccharide was used to establish a mouse model of

acute lung injury (ALI), mimicking the inflammatory response and tissue damage associated with this condition. This is a well-established and widely used method for constructing ALI models, as employed in our previous studies (Zhao et al., IJMS, 2023; Kang et al., *Frontiers in Immunology*, 2024). As suggested, it is of broader significance to clarify the role of necroptosis in ALI models established by different methods. Therefore, we established a cecal ligation and puncture (CLP) model of sepsis-induced ALI. We performed cecal ligation, puncture, and extrusion of feces from the intestinal lumen in C57 mice to simulate abdominal infection and intestinal necrosis. The specific experimental steps are described in detail in the methodology section under ***"Cecal ligation and puncture (CLP) model construction"*** (Page 15 Line 426-434) (Ruiz, Vardon-Bouines et al. 2016, Bojalil, Ruíz-Hernández et al. 2023). Western blot analysis of lung tissues showed that phosphorylation levels of RIPK1 were significantly increased, while phosphorylation levels of RIPK3 and MLKL did not show obvious elevation (Figure S3, Page 5-6 Line 133-137).

HBE cells treated with LPS for 24 hours showed a significant increase in the phosphorylation levels of RIPK1 and RIPK3, while no noticeable increase in MLKL phosphorylation was observed. This is consistent with the findings from our CLP mouse model. Therefore, we believe this could be due to extensive organ failure and premature death in the septic model mice, where necroptosis pathways in lung tissues were not fully activated before death. Which was detailedly described in Page 6 Line 151-155 (Figure S4B-L) of the article.

It has been reported that necroptosis occurs in lung tissues from various ALI models, including corn oil instillation and hyperoxic ventilation in rats (Han, Guan et al., 2018; Qin, Sai et al., 2019), ischemia-reperfusion injury in lung transplantation (Zhao, Ning et al., 2015), and lung tissues from patients who died

from H7N9 infection (Qin, Sai et al., 2019). Additionally, knockout of RIPK3 effectively protected lung tissues from injury in mechanically ventilated mice (Siempos, Ma et al., 2018). On the other hand, using RIPK1 phosphorylation inhibitors significantly improved survival rates in peritoneally injected adult cecal plasma-induced neonatal septic mice (Bolognese, Yang et al., 2018). Therefore, we conclude that necroptosis in the LPS-induced ALI model is not directly caused by LPS itself but is an inherent feature of acute lung injury models induced by direct factors. As shown in **Page 13-14 Line 353-360**.

Confirming the role of necroptosis through samples from ARDS patients, such as lung tissue, is of great significance. It is difficult to obtain lung tissue from patients with pure ARDS. In cases where patient samples are from individuals with comorbidities, it may be challenging to exclude confounding factors. Furthermore, this approach also raises ethical concerns related to human research. However, there are reports in the literature indicating that phosphorylated MLKL levels were elevated in the plasma of patients infected with COVID-19 (Li, Zhang et al. 2020, Karki, Sharma et al. 2021, Koupenova, Corkrey et al. 2021).

Bojalil, R., A. Ruíz-Hernández, A. Villanueva-Arias, L. M. Amezcua-Guerra, S. Cásarez-Alvarado, A. M. Hernández-Dueñas, V. Rodríguez-Galicia, L. Pavón, B. Marquina, E. Becerril-Villanueva, R. Hernández-Pando and R. Márquez-Velasco (2023). "Two murine models of sepsis: immunopathological differences between the sexes-possible role of TGFβ1 in female resistance to endotoxemia." *Biol Res* **56**(1): 54.

Karki, R., B. R. Sharma, S. Tuladhar, E. P. Williams, L. Zalduondo, P. Samir, M. Zheng, B. Sundaram, B. Banoth, R. K. S. Malireddi, P. Schreiner, G. Neale, P. Vogel, R. Webby, C. B. Jonsson and T. D. Kanneganti (2021). "Synergism of TNF-α and IFN-γ Triggers Inflammatory Cell Death, Tissue Damage, and Mortality in SARS-CoV-2 Infection and Cytokine Shock Syndromes." *Cell* **184**(1): 149-168.e117.

Koupenova, M., H. A. Corkrey, O. Vitseva, K. Tanriverdi, M. Somasundaran, P. Liu, S. Soofi, R. Bhandari, M. Godwin, K. M. Parsi, A. Cousineau, R. Maehr, J. P. Wang, S. J. Cameron, J. Rade, R. W.

Zhejiang University

Department of Critical Care Medicine of the Second Affiliated Hospital
Zhejiang University School of Medicine, Hangzhou, China

Finberg and J. E. Freedman (2021). "SARS-CoV-2 Initiates Programmed Cell Death in Platelets." *Circ Res* **129**(6): 631-646.

Li, S., Y. Zhang, Z. Guan, H. Li, M. Ye, X. Chen, J. Shen, Y. Zhou, Z. L. Shi, P. Zhou and K. Peng (2020). "SARS-CoV-2 triggers inflammatory responses and cell death through caspase-8 activation." *Signal Transduct Target Ther* **5**(1): 235.

Ruiz, S., F. Vardon-Bounes, V. Merlet-Dupuy, J. M. Conil, M. Buléon, O. Fourcade, I. Tack and V. Minville (2016). "Sepsis modeling in mice: ligation length is a major severity factor in cecal ligation and puncture." *Intensive Care Med Exp* **4**(1): 22.

4. Comparison with standard therapies:

The authors did not discuss existing ALI treatments and their associated limitations that motivated this study. How does GW806742X compare with standard therapies? Does inhibiting MLKL activation offer better outcomes than current treatments?

RESPONSE:

We appreciate the insightful comments from the referee. Currently, the treatment strategies for ALI/ARDS in clinical practice mainly involve conservative supportive therapies such as low tidal volume positive pressure ventilation, prone position ventilation, extracorporeal membrane oxygenation (ECMO), and restrictive fluid management (Bos and Ware 2022, Gorman, O'Kane et al. 2022, Grasselli, Calfee et al. 2023, Ramji, Hafiz et al. 2023, Qadir, Sahetya et al. 2024). Regarding pharmacological treatments, there are currently no specific drugs for ARDS. Commonly used medications include corticosteroids, neuromuscular blockers, pulmonary vasodilators, and other adjunctive therapies. However, all these available methods have not shown ideal efficacy in treating ARDS.

AT II cells are responsible for synthesizing and secreting surfactants and endogenous antimicrobial peptides, and also possess significant regenerative capacity, as they can differentiate into AT I cells, thereby promoting the repair of damaged lung tissue (Ruaro, Salton et al. 2021). Thereby preventing the release

of related inflammatory factors within these cells and addressing the excessive inflammatory response in lung tissue in ARDS patients at its root. In this study, ALI mice treated with a lipid-encapsulated nanomedicine showed significant inhibition of necroptosis, decreased levels of inflammatory cytokines (IL-1 β , IL-6, TNF α), reduced inflammatory cell recruitment, alleviated alveolar epithelial cell damage, and improved survival (**Figure 7, Page 10 Line 262-272**). Therefore, we believe that lipid-encapsulated nanomedicine, by targeting AT II cells and inhibiting their necroptosis, effectively achieves the goal of treating ALI.

Bos, L. D. J. and L. B. Ware (2022). "Acute respiratory distress syndrome: causes, pathophysiology, and phenotypes." *Lancet* **400**(10358): 1145-1156.

Gorman, E. A., C. M. O'Kane and D. F. McAuley (2022). "Acute respiratory distress syndrome in adults: diagnosis, outcomes, long-term sequelae, and management." *Lancet* **400**(10358): 1157-1170.

Grasselli, G., C. S. Calfee, L. Camporota, D. Poole, M. B. P. Amato, M. Antonelli, Y. M. Arabi, F. Baroncelli, J. R. Beitler, G. Bellani, G. Bellington, B. Blackwood, L. D. J. Bos, L. Brochard, D. Brodie, K. E. A. Burns, A. Combes, S. D'Arrigo, D. De Backer, A. Demoule, S. Einav, E. Fan, N. D. Ferguson, J. P. Frat, L. Gattinoni, C. Guérin, M. S. Herridge, C. Hodgson, C. L. Hough, S. Jaber, N. P. Juffermans, C. Karagiannidis, J. Kesecioglu, A. Kwizera, J. G. Laffey, J. Mancebo, M. A. Matthay, D. F. McAuley, A. Mercat, N. J. Meyer, M. Moss, L. Munshi, S. N. Myatra, M. Ng Gong, L. Papazian, B. K. Patel, M. Pellegrini, A. Perner, A. Pesenti, L. Piquilloud, H. Qiu, M. V. Ranieri, E. Riviello, A. S. Slutsky, R. D. Stapleton, C. Summers, T. B. Thompson, C. S. Valente Barbas, J. Villar, L. B. Ware, B. Weiss, F. G. Zampieri, E. Azoulay and M. Cecconi (2023). "ESICM guidelines on acute respiratory distress syndrome: definition, phenotyping and respiratory support strategies." *Intensive Care Med* **49**(7): 727-759.

Qadir, N., S. Sahetya, L. Munshi, C. Summers, D. Abrams, J. Beitler, G. Bellani, R. G. Brower, L. Burry, J. T. Chen, C. Hodgson, C. L. Hough, F. Lamontagne, A. Law, L. Papazian, T. Pham, E. Rubin, M. Siuba, I. Telias, S. Patolia, D. Chaudhuri, A. Walkey, B. Rochweg and E. Fan (2024). "An Update on Management of Adult Patients with Acute Respiratory Distress Syndrome: An Official American Thoracic Society Clinical Practice Guideline." *Am J Respir Crit Care Med* **209**(1): 24-36.

Ramji, H. F., M. Hafiz, H. H. Altaq, S. T. Hussain and F. Chaudry (2023). "Acute Respiratory Distress Syndrome; A Review of Recent Updates and a Glance into the Future." *Diagnostics (Basel)* **13**(9).

Ruaro, B., F. Salton, L. Braga, B. Wade, P. Confalonieri, M. C. Volpe, E. Baratella, S. Maiocchi and M. Confalonieri (2021). "The History and Mystery of Alveolar Epithelial Type II Cells: Focus on Their Physiologic and Pathologic Role in Lung." *Int J Mol Sci* **22**(5).

5. Physiological relevance:

Zhejiang University

Department of Critical Care Medicine of the Second Affiliated Hospital
Zhejiang University School of Medicine, Hangzhou, China

The study aims to "mitigate lung injury, improve gas exchange, and ultimately improve the outcomes of patients with ARDS," but no physiological tests address these points. The authors could demonstrate improvements in gas exchange and survival in ALI mice treated with necroptosis inhibitors compared to standard therapies.

RESPONSE:

We greatly appreciate the referee for bringing this to our attention. The description in the original manuscript, 'mitigate lung injury, improve gas exchange, and ultimately improve the outcomes of patients with ARDS,' was indeed imprecise. Therefore, in the revised manuscript, it has been modified to: 'Therefore, strategies aimed at reducing the inflammatory response while simultaneously protecting and repairing the alveolar epithelium represent promising therapeutic approaches,' as seen on **Page 3 Line 64-66**.

As mentioned in the previous response, there are currently lacking of effective treatments for ARDS in clinical practice, and the supportive therapies commonly used in clinical settings are difficult to replicate in mouse models (Bos and Ware 2022, Gorman, O'Kane et al. 2022, Grasselli, Calfee et al. 2023, Ramji, Hafiz et al. 2023, Qadir, Sahetya et al. 2024). A direct comparison between the current treatment options and this targeted drug may not be possible. Therefore, in our study, we evaluated the therapeutic effect of the targeted drug by assessing the inflammatory cell count in BALF, measuring inflammatory cytokines in lung tissue, and performing HE staining of lung tissue to evaluate inflammatory cell aggregation, interstitial edema, and the integrity of the alveolar epithelium. The survival curves showed that the survival rate of ALI mice treated with lipid-encapsulated GW806742X was 50% on day 4, while the survival rate of the model group was only 10%. These findings suggest the promising therapeutic effect of this lipid-encapsulated nanomedicine.

Zhejiang University

Department of Critical Care Medicine of the Second Affiliated Hospital
Zhejiang University School of Medicine, Hangzhou, China

Bos, L. D. J. and L. B. Ware (2022). "Acute respiratory distress syndrome: causes, pathophysiology, and phenotypes." *Lancet* **400**(10358): 1145-1156.

Gorman, E. A., C. M. O'Kane and D. F. McAuley (2022). "Acute respiratory distress syndrome in adults: diagnosis, outcomes, long-term sequelae, and management." *Lancet* **400**(10358): 1157-1170.

Grasselli, G., C. S. Calfee, L. Camporota, D. Poole, M. B. P. Amato, M. Antonelli, Y. M. Arabi, F. Baroncelli, J. R. Beitler, G. Bellani, G. Bellington, B. Blackwood, L. D. J. Bos, L. Brochard, D. Brodie, K. E. A. Burns, A. Combes, S. D'Arrigo, D. De Backer, A. Demoule, S. Einav, E. Fan, N. D. Ferguson, J. P. Frat, L. Gattinoni, C. Guérin, M. S. Herridge, C. Hodgson, C. L. Hough, S. Jaber, N. P. Juffermans, C. Karagiannidis, J. Kesecioglu, A. Kwizera, J. G. Laffey, J. Mancebo, M. A. Matthay, D. F. McAuley, A. Mercat, N. J. Meyer, M. Moss, L. Munshi, S. N. Myatra, M. Ng Gong, L. Papazian, B. K. Patel, M. Pellegrini, A. Perner, A. Pesenti, L. Piquilloud, H. Qiu, M. V. Ranieri, E. Riviello, A. S. Slutsky, R. D. Stapleton, C. Summers, T. B. Thompson, C. S. Valente Barbas, J. Villar, L. B. Ware, B. Weiss, F. G. Zampieri, E. Azoulay and M. Cecconi (2023). "ESICM guidelines on acute respiratory distress syndrome: definition, phenotyping and respiratory support strategies." *Intensive Care Med* **49**(7): 727-759.

Qadir, N., S. Sahetya, L. Munshi, C. Summers, D. Abrams, J. Beitler, G. Bellani, R. G. Brower, L. Burry, J. T. Chen, C. Hodgson, C. L. Hough, F. Lamontagne, A. Law, L. Papazian, T. Pham, E. Rubin, M. Siuba, I. Teliya, S. Patolia, D. Chaudhuri, A. Walkey, B. Rochweg and E. Fan (2024). "An Update on Management of Adult Patients with Acute Respiratory Distress Syndrome: An Official American Thoracic Society Clinical Practice Guideline." *Am J Respir Crit Care Med* **209**(1): 24-36.

Ramji, H. F., M. Hafiz, H. H. Altaq, S. T. Hussain and F. Chaudry (2023). "Acute Respiratory Distress Syndrome; A Review of Recent Updates and a Glance into the Future." *Diagnostics (Basel)* **13**(9).

Minor Comments:

1. Introduction:

The introduction reads like a review article. The authors can significantly shorten this section, focusing on current findings relating ALI/ARDS to necroptosis and associated treatment models, including standard therapies and their deficiencies. This will clarify the novelty and impact of the current study.

RESPONSE:

We appreciate the insightful comments from the reviewer. Based on your feedback, we further refined the introduction to focus primarily on current findings linking ALI/ARDS to necroptosis, which enhances the readability and conciseness of the manuscript while making it more aligned with the central theme of the article.

2. Figures and images:

Please include a bright field image for topography visualization in Figures 3J and S4.

RESPONSE:

Thank you for your suggestion. We have made revisions in response to your questions, as indicated in **Figure 3J** and **Figure S6B** of the manuscript.

For Figure 5D, include original images from which the zoomed cells were derived, along with a scale bar, bright field, and untreated cell images to help understand protein distribution before and after LPS treatment.

RESPONSE:

Thank you for your recommendation. We have revised the figure to include the original images from which the zoomed-in cell images were derived, along with the requested scale bar and untreated cell images, as shown in **Figure 5D** of the manuscript.

3. Abbreviations and clarity:

Define "SPC" upon first use.

RESPONSE:

SPC, also named as lung-associated surfactant protein C, is an extremely hydrophobic surfactant protein that is essential for lung function and homeostasis after birth. In response to your suggestion, we have added a definition of SPC to the original article, which appears at **Page 9 Line 236-237** in the article. We hope that this descriptive addition will make it easier for you.

Clarify the sentence on page 17: "The lung tissues were obtained in animal experiences for PCR assay."

RESPONSE:

The correct description in the original text should be: "The lung tissues or cells modelled and obtained by '*In vivo animal experiments*' or '*Cell Culture and LPS Stimulation*' were used for PCR assays." We have updated this in the revised manuscript, specifically in the "***Quantitative real-time polymerase chain reaction***" section on **Page 16 Line 452-455**.

4. Reproducibility:

Review all experimental methods to ensure they are reproducible. For example, "The next day, the immunoprecipitants were washed five times and boiled with 2 × SDS Loading Buffer. "next day" is not precise, and "boiled" without any indication of duration and temperature is highly subjective. Please specify the precise timing and temperature for boiling immunoprecipitants (page 19).

RESPONSE:

Thank you for your suggestions. We have thoroughly reviewed the methodology section of the original article and made corrections where clarity was lacking (**Page 18 Line 502-505**). We hope these revision will facilitate readers in replicating our study with ease.

5. Relevance of Procedures:

Ensure all described procedures are relevant to the study. Some procedures described in the manuscripts may not be relevant to the current study. For example, "Goat Anti-rabbit IgG (HRP) (Abcam , ab205718), FTIC (aatbio, 11060), CY3 (aatbio, 11065) and CY5 (aatbio, 11066) were used as fluorescent labelling" (page 19). First, what is FTIC? Do you mean FITC? Second, where are the images obtained with "FTIC" and CY3? Please clarify the usage of fluorescent labeling throughout the manuscript and include images obtained with FITC and CY3 if mentioned.

RESPONSE:

Thank you for the referee's review. We have revised the immunofluorescence steps in the original manuscript and found them to be vague and potentially misleading. Therefore, we have provided a more detailed description in the revised manuscript on **Page 19 Line 527-549** and **Page 20 Line 568-587**. In addition, the images obtained with FITC, CY3, and CY5 are shown in **Figure 3J** and **Figure 5D** of the original manuscript, and the colors in the images correspond to the colors of the fluorescent dyes used.

Clarify some experimental procedures. For example, on page 19, the authors state that "Hematoxylin-eosin (HE) staining was performed to explore damage in the lung tissue. Prior to histological analysis, lungs were immersed in 4% paraformaldehyde for 24 hours." Was this H&E staining applicable to injured lungs? If yes, please briefly explain why the permeation and fixation steps were performed at the end. Include Eosin staining time in the protocol.

RESPONSE:

Thank you for your questions. In our experiment, we used a 1% eosin staining solution for 2 seconds. After staining, xylene was used for tissue clarification. Xylene effectively dissolves certain substances in tissue sections, such as residual fats and incompletely removed impurities, making the sections clearer and more transparent for observation under a microscope. Additionally, xylene has a refractive index close to that of glass, allowing sections treated with xylene to match the refractive index of glass slides and cover slips. This enhances light transmission through the sections, improving the overall observation quality. Furthermore, xylene alters the surface properties of the tissue sections, making them more compatible with mounting media, such as mounting glue or resin. This ensures a secure seal between the slide and cover slip, which helps preserve the sections for long-term storage and observation. Sealing also prevents drying,

Zhejiang University

Department of Critical Care Medicine of the Second Affiliated Hospital
Zhejiang University School of Medicine, Hangzhou, China

which can cause cell shrinkage and tissue structure deformation, while enhancing contrast, clarity, and transparency for better visualization. We have also included the processing time for each step in the revised manuscript (**Page 18 Line 502-505**), and hope that this addresses your concerns.

6. Literature citation:

On page 7, the authors acknowledged that "necroptosis is present in ALI" but omitted citing the relevant literature (page 7).

RESPONSE:

Thank you very much for your reminder. We have added the citation of the aforementioned references on **page 6 Line 141-145 (Reference 29, 30)** of the revised manuscript.

There are many other instances with similar issues in the experimental section. For example, "Tissue Immunofluorescence: Samples from both ALI and control were treated following a previously protocol" (page 19). Please cite all previous methods mentioned in the manuscript.

RESPONSE:

Thank you for raising the issue. We have now made corrections to these two oversights, located on **Page 19 Line 527-530 and Page 20 Line 560-562** of the original text

7. Figures:

In Figure 5, "The results showed that p-RIPK1, p-RIPK3, and p-MLKL could be used as targets to precipitate the other two proteins, indicating their interaction or complex formation" (page 22). What are the "other two proteins"?

RESPONSE:

We have made more refined revisions to the description on **Page 24, Line 687-690**, to clearly convey the intended message, and we hope this will also facilitate your understanding as well as that of the readers.

Figure 7: Please provide clearer zoomed-in images to display infiltrating cells more distinctly; this will facilitate determining the cell types involved in ALL.

RESPONSE:

Thank you very much for your suggestion. We have reformatted the HE staining images to cover both high and low magnification fields of view, and have made the changes on **Figure 7J** of the original text. We hope this will clearly demonstrate the therapeutic effects.

Spelling and Grammar:

Multiple spelling and grammatical errors in the manuscript need correction.

Examples include:

Alleviates → Alleviate (Title page)

ncroptosis → necroptosis (page 4)

Danger-associated molecular patterns → Damage-associated molecular patterns (page 4)

Remove double periods (..) after "Figure 1B" (page 6)

necrodtatin-1 → necrostatin-1 (pages 8, three places on page 18)

PIPK1 → RIPK1 (page 22)

RESPONSE:

Thank you for your meticulous reading. We have made the corresponding revisions based on the issues you identified, and the revised locations are as follows:

Title corrected as "Targeting Alveolar Epithelial Cells with Lipid Micelle-Encapsulated Necroptosis Inhibitors to Alleviate Acute Lung Injury";

Zhejiang University

Department of Critical Care Medicine of the Second Affiliated Hospital
Zhejiang University School of Medicine, Hangzhou, China

Page 3 Line 68, corrected as “regulated cell death (RCD)”;

Page 5 Line 117, “..” corrected as “.”;

Page 5 Line 131, “group” corrected as “mice”;

Page 6 Line 139, “healthy” corrected as “control”;

Page 7 Line 169, “protein” corrected as “proteins”;

Page 10 Line 258, “PEG-DSPE” corrected as “DSPE-PEG”;

Page 16 Line 461-463, “m- TNF α ” corrected as “mTNF α ”, “m-IL1 β ” corrected as “mIL1 β ”;

Page 18 Line 513, “permeation” corrected as “clarification”;

Page 22 Line 633, “both ALI group” corrected as “ALI group”;

Page 23 Line 654, “PIPK1” corrected as “RIPK1”;

Page 23 Line 662, added “(green)”;

Page 23 Line 663, “um” corrected as “ μ m”;

Reviewer #3 (Remarks to the Author):

Fig 2. There is no mention of the timepoints in the results section or in the figure legend.

Figure 2]- quantification is missing across multiple animals as well as multiple images per animal.

RESPONSE:

We highly appreciate the referee's opinion, and we have added the LPS dose and duration of intervention for the mice in the original text (**Page 15 Line 423-424**). Additionally, we have provided the number of mice used in our immunofluorescence experiment in the figure legend (**Page 23 Line 649-652**).

Fig 3. Details regarding the HBE cells used are lacking from the methods. Are these primary cells? Obtained from where? How were they grown eg in ALI? Can the authors show the cells undergo cell lysis and/or cell death?

RESPONSE:

HBE is a cell line derived from human bronchial epithelial cells, which are immortalized through transfection with the E6 and E7 genes of the human papillomavirus (HPV) (Forbes, Shah et al. 2003, Fulcher, Gabriel et al. 2009). They are widely used in laboratory research to model the barrier function of airway epithelium.

We have added the results of the CCK8 cell viability assay in the original text (**Page 6 Line 150-154**) and **Figure S4** to demonstrate the damage or cell death is occurring when treated with 200 μ g/ml LPS for 24 hours.

Forbes, B., A. Shah, G. P. Martin and A. B. Lansley (2003). "The human bronchial epithelial cell line 16HBE14o- as a model system of the airways for studying drug transport." *Int J Pharm* **257**(1-2): 161-167.

Zhejiang University

Department of Critical Care Medicine of the Second Affiliated Hospital
Zhejiang University School of Medicine, Hangzhou, China

Fulcher, M. L., S. E. Gabriel, J. C. Olsen, J. R. Tatreau, M. Gentsch, E. Livanos, M. T. Saavedra, P. Salmon and S. H. Randell (2009). "Novel human bronchial epithelial cell lines for cystic fibrosis research." *Am J Physiol Lung Cell Mol Physiol* **296**(1): L82-91.

Figure 3J- quantification is missing across multiple images.

RESPONSE:

We appreciate the insightful comments from the reviewer. We have added the bar graph of the corresponding fluorescence intensity statistics for **Figure 3J** in the original text as **Figure S5**, and provided a description of it on **Page 6 Line 160-162** of the original manuscript.

Fig 4 – why is RIPK1 no longer expressed in nec1 treated cells. They should still express total RIPK1. Are the inhibitors toxic? Can the authors show toxicity isn't an issue? Inhibitor alone controls are missing.

RESPONSE:

Thank you very much for your question. Nec-1 is a specific inhibitor of RIPK1, so when it is used to intervene in HBE cells, the expression of RIPK1 will decrease obviously. We have added a clarification about this issue on **Page 7 Line 173-174** of the manuscript. To confirm that the dosage of the inhibitor does not cause toxicity to the cells, we performed a cell viability assay using CCK-8 and the results are shown in **Figure S6A**. A detailed description of this can be found on **Page 7 Line 175-176** of the manuscript.

Fig 6 – can the authors demonstrate TRIF and Myd88 knockdown?

RESPONSE:

We greatly appreciate the referee for bringing this to our attention. We performed PCR analysis to detect the corresponding RNA Trif and Myd88 in the siRNA knockdown HBE cells, and the results have been added to **Figure S8A-B** in

the original manuscript. A detailed description can be found on **Page 8 Line 216-218**.

Fig 7 – analysis of lung injury is very limited to HandE sections. More detailed analysis is warranted. A dose response would also be useful. The methods lack details regarding how lung injury was scored. Eg what does 1-9 mean? Was it done blindly? And by how many readers?

RESPONSE:

We appreciate the insightful comments from the reviewer. Lung injury score is modified based on the scoring criteria from **reference 72**. This is described in detail on **Page 18 Line 518-525** of the original manuscript. The scorer was blinded to the sample grouping and treatment. For each sample, three random fields were selected, and the scores from the three fields were averaged. The average score we entered into GraphPad Prism 8 for statistical analysis.

Fig 7 – critical data showing treatment inhibits MLKL is missing. Also details as to how mice were treated e.g. the route of administration is lacking.

RESPONSE:

Thank you very much for your comments. We have added the relevant data for the targeted drug treatment in mice in **Figure 7**, including bronchoalveolar lavage cell counts, survival curves, and inflammatory factor detection. Detailed descriptions can be found in **Page 10 Line 267-281**. The method of administering the treatment drug is the same as LPS, both via airway instillation, which is described in the *“In vivo animal experiments”* section on **Page 15 Line 417-424**.

Details regarding the statistical analysis tests are missing from the legends.

RESPONSE:

Zhejiang University

Department of Critical Care Medicine of the Second Affiliated Hospital
Zhejiang University School of Medicine, Hangzhou, China

Thank you for your suggestion. We have now provided detailed statistical methods in the figure legends for each figure, and the specific content can be found on **Pages 23-27** of the manuscript . Additionally, we have included a detailed description of the statistical methods in the “*Statistical analysis*” of the methodology section.

Overall, the methods and figure legends lack detail, making it difficult to assess the data and the conclusions.

RESPONSE:

Thank you very much for your feedback. We have provided detailed additions to the methodology and figure legends sections of the manuscript , such as Scale, fluorescence channel, group, sample size per group, and number of repetitions. Please look at **Pages 14-26** for the specific details.

The introduction is very long, almost 3 pages in length.

RESPONSE:

Thank you very much for your suggestions. We have streamlined the introduction of the manuscript and improved some vague statements (**Page 3-4**).

31 January 2025

Re: Resubmission of Revised Manuscript (**COMMSBIO-24-3412A**)

Dear reviewer,

We are pleased to resubmit our revised manuscript for your consideration for publication in *Communications Biology*. We sincerely appreciate the time and effort of both the editor and reviewers in evaluating our previous submission. Their insightful feedback has been instrumental in improving our work. The revisions are highlighted in the revised manuscript using **blue text** for ease of identification. Detailed responses to the reviewer comments are provided in the accompanying Response to reviewers document. We believe these modifications have significantly enhanced the rigor and clarity of our study, and we are confident that the revised manuscript now meets the high standards of *Communications Biology*.

Reviewer #4 (Remarks to the Author):

The authors have not adequately addressed the following comments:

Fig 7 – analysis of lung injury is very limited to HandE sections. More detailed analysis is warranted.

RESPONSE: Thank you for bringing this issue to our attention. In addition to the original descriptions of HE pathology and inflammatory factor detection, we have added new data in the revised manuscript to better illustrate this issue. Epithelial cells are interconnected through a series of protein complexes, such as adhesion molecules (E-cadherin) and tight junction proteins, forming a crucial barrier that resists external infections, thereby protecting local tissues and preventing inflammatory damage (Wang, Xue et al. 2024). To present the therapeutic effects of

Zhejiang University

Department of Critical Care Medicine of the Second Affiliated Hospital
Zhejiang University School of Medicine, Hangzhou, China

our modified drug from multiple perspectives, we used the E-cadherin antibody to perform immunohistochemical staining on lung tissues from the Control, ALI, and SPC-M-GW treatment mice. The results show that the expression of E-cadherin protein is similar in the Control and SPC-M-GW treated ALI mice, whereas in the ALI mice, the expression of E-cadherin protein is obviously absent, which are shown in **Figure 7K**, with a detailed description provided on **Page 10, Line 275-280** of the manuscript. We hope this result provides you and the readers with a deeper understanding of the therapeutic effects.

Reference:

Wang, J., X. Xue, X. Zhao, L. Luo, J. Liu, S. Dai, F. Zhang, R. Wu, Y. Liu, C. Peng and Y. Li (2024). "Forsythiaside A alleviates acute lung injury by inhibiting inflammation and epithelial barrier damages in lung and colon through PPAR- γ /RXR- α complex." *J Adv Res* **60**: 183-200.

Fig 7 – critical data showing SPC-M-GW treatment inhibits MLKL in vivo is missing. This data is very critical with regard to validating the conclusions as there is currently no evidence it does inhibit MLKL.

RESPONSE: We sincerely appreciate your feedback. The active ingredient GW806742X used to formulate our targeted drug inhibits the membrane translocation of MLKL (Hildebrand, Tanzer et al. 2014). Therefore, our modified drug also exerts its inhibitory effect by blocking the process of MLKL translocating to the cell membrane. Additionally, MLE cells express a certain amount of SPC protein on their surface (**Figure S9**)(Kumar, Madhurakkat Perikamana et al. 2022). As such, MLE was used to model alveolar type II cells (AT II), and were treated with the targeted drug following LPS intervention. The resulting cells were stained with MLKL antibody (green) and Tubulin antibody (red), followed by confocal imaging. As shown in **Figure 7F**, MLE subjected to LPS intervention alone displayed a linear green fluorescence band at the cell membrane, indicating MLKL aggregation at the membrane. However,

Zhejiang University

Department of Critical Care Medicine of the Second Affiliated Hospital
Zhejiang University School of Medicine, Hangzhou, China

no such fluorescence was observed in the Control, GW806742X, or SPC-M-GW treatment groups. The results provided on **Page 10, Lines 258-267**, further demonstrate the efficacy of our targeted drug in inhibiting MLKL.

Reference:

Hildebrand, J. M., M. C. Tanzer, I. S. Lucet, S. N. Young, S. K. Spall, P. Sharma, C. Pierotti, J. M. Garnier, R. C. Dobson, A. I. Webb, A. Tripaydonis, J. J. Babon, M. D. Mulcair, M. J. Scanlon, W. S. Alexander, A. F. Wilks, P. E. Czabotar, G. Lessene, J. M. Murphy and J. Silke (2014). "Activation of the pseudokinase MLKL unleashes the four-helix bundle domain to induce membrane localization and necroptotic cell death." Proc Natl Acad Sci U S A **111**(42): 15072-15077.

Kumar, V., S. K. Madhurakkat Perikamana, A. Tata, J. Hoque, A. Gilpin, P. R. Tata and S. Varghese (2022). "An In Vitro Microfluidic Alveolus Model to Study Lung Biomechanics." Front Bioeng Biotechnol **10**: 848699.

Please note, Figure S6 to S9 are incorrectly labelled in the figure legends.

RESPONSE: Thank you very much for your reminder. We have corrected the labeling of **Figures S6 to S10** in the revised manuscript.

Zhejiang University

Department of Critical Care Medicine of the Second Affiliated Hospital Zhejiang
University School of Medicine, Hangzhou, China

28 February 2025

Dear reviewers,

Re: Resubmission of Revised Manuscript (COMMSBIO-24-3412C)

REVIEWERS' COMMENTS:

See the Nature Portfolio author and referees' website at www.nature.com/authors for information about policies, services and author benefits.

RESPONSE:

We would like to express our sincere gratitude to you and the reviewers for the thoughtful feedback on our manuscript entitled "**Targeting Alveolar Epithelial Cells with Lipid Micelle-Encapsulated Necroptosis Inhibitors to Alleviate Acute Lung Injury**".

These revisions include addressing all concerns raised and ensuring the manuscript complies with the journal's formatting and style guidelines. We have submitted the revised manuscript along with the updated supplementary materials. We hope that these revisions meet the expectations of the editorial team and the reviewers, and that the manuscript is now suitable for publication in *Communications Biology*.

We truly appreciate your time and consideration, and we look forward to hearing from you soon regarding the next steps in the review process.

Sincerely yours,

Bao-ping Tian, MD, PhD, Department of Critical Care Medicine, The Second Affiliated Hospital,
Zhejiang University School of Medicine, 88 Jiefang Rd., Hangzhou 310009, China. Email:

TianBP@zju.edu.cn

Yang Du, PhD, Department of Hepatobiliary and Pancreatic Surgery, The Second Affiliated Hospital,
Zhejiang University School of Medicine, 88 Jiefang Rd., Hangzhou 310009, China.

Email: yangdu@zju.edu.cn

Zhao-cai Zhang, MD, PhD, Department of Critical Care Medicine, The Second Affiliated Hospital,
Zhejiang University School of Medicine, 88 Jiefang Rd., Hangzhou 310009, China. Email:

2313003@zju.edu.cn

Lipid micelle-encapsulated necroptosis inhibitor targets epithelial cells to effectively alleviates acute airway inflammatory injury

Zhi-ying Kang^{1#}, Nan-xia Xuan^{1#}, Qi-chao Zhou¹, Qian-yu Huang¹, Gen-sheng Zhang¹, Wei Cui¹, Zhao-cai Zhang^{1*}, Yang Du^{2*}, Bao-ping Tian^{1*}

- 1 Department of Critical Care Medicine, The Second Affiliated Hospital, Zhejiang University School of Medicine, Hangzhou, Zhejiang, 310009, China.
- 2 Department of Hepatobiliary and Pancreatic Surgery, The Second Affiliated Hospital, Zhejiang University School of Medicine, Hangzhou, Zhejiang, 310009, China.

***Corresponding author:**

Bao-ping Tian, MD, PhD, Department of Critical Care Medicine, The Second Affiliated Hospital, Zhejiang University School of Medicine, 88 Jiefang Rd., Hangzhou 310009, China. Email: TianBP@zju.edu.cn; Tel: 86-571-87767018

Yang Du, PhD, Department of Hepatobiliary and Pancreatic Surgery, The Second Affiliated Hospital, Zhejiang University School of Medicine, 88 Jiefang Rd., Hangzhou 310009, China. Email: yangdu@zju.edu.cn

Zhao-cai Zhang, MD, PhD, Department of Critical Care Medicine, The Second Affiliated Hospital, Zhejiang University School of Medicine, 88 Jiefang Rd., Hangzhou 310009, China. Email: 2313003@zju.edu.cn

Zhi-ying Kang and Nan-xia Xuan contributed equally to this work.

Conflict of interest

The authors declare no conflict of interest.

Abstract: Acute lung injury (ALI) or its more severe form, acute respiratory distress syndrome (ARDS), represents a critical condition characterized by extensive inflammation within the airways. Necroptosis, a form of cell death, has been implicated in the pathogenesis of various inflammatory diseases. However, the precise characteristics and mechanisms of necroptosis in ARDS remain unclear. Thus, our study seeks to elucidate the specific alterations and regulatory factors associated with necroptosis in ARDS and to identify potential therapeutic targets for the disease. We discovered that necroptosis mediates the progression of ALI through the activation and formation of the RIPK1/RIPK3/MLKL complex. Moreover, our research substantiated the involvement of both Myd88 and TRIF in the activation of the TLR4 signaling pathway in ALI. Furthermore, we developed a lipid micelle-encapsulated drug targeting MLKL in alveolar type II epithelial cells and successfully applied it to treat ALI in mice. This targeted nanoparticle effectively mitigated epithelial cell damage by selectively inhibiting necroptosis, thereby achieving the objective of attenuating inflammatory injury. Our study delves into the specific mechanisms of necroptosis in ALI and proposes novel targeted therapeutic agents, presenting innovative strategies for the management of ARDS.

Keywords: ALI/ARDS; Necroptosis; Inflammatory injury; Lipid micelle; Targeted therapy

1. Introduction

Acute lung injury (ALI) or its more severe form, acute respiratory distress syndrome (ARDS), is a clinical syndromes characterised by diffuse alveolar injury, non-cardiogenic pulmonary oedema, and refractory hypoxaemia. Its main pathophysiological alteration is inflammation-mediated acute, progressively exacerbating diffuse alveolar injury¹. Alveolar epithelial cells play a crucial role in maintaining the stability of alveolar structures and actively participate in the immune response of the lungs^{2, 3}. They form a barrier between the alveolar space and surrounding tissues, helping to prevent the entry of pathogens and harmful substances. Additionally, alveolar epithelial cells produce various immune mediators and cytokines that regulate the immune response and contribute to the defense against infections and inflammatory processes in the lungs. Epithelial cell damage and dysfunction are not only key features of alveolar injury in ARDS, but also play an important role in the development and progression of ARDS⁴. The disruption of the alveolar epithelial barrier leads to increased permeability, inflammatory cell infiltration, and pulmonary edema, ultimately resulting in impaired gas exchange and severe hypoxemia. Moreover, the release of pro-inflammatory mediators and cytokines by damaged epithelial cells can further exacerbate the inflammatory response and contribute to the pathogenesis of ARDS⁵. It has been found that in ARDS caused⁵ by diseases such as bacterial and viral infections, which can cause direct acute lung injury, activation of the innate immune response was detected along with a more pronounced elevation of biomarkers of epithelial damage, suggesting that there is both an active immune response and a higher level of epithelial damage compared with indirect damage⁶. Given the role of inflammation and epithelial cell damage in the development of ARDS, finding ways to reduce the inflammatory response while also protecting and repairing the alveolar epithelium is a potential therapeutic approach. It may involve targeting specific inflammatory pathways, such as cytokines or immune cell activation, to dampen the excessive immune response. The goal is to mitigate lung injury, improve gas exchange, and ultimately improve the outcomes of patients with ARDS.

Currently, cell death is divided into regulated cell death (RCD) and non-regulated cell death. RCD is generally referred to as "apoptosis" and is characterised by cytoplasmic shrinkage, membrane vesiculation, chromatin condensation and DNA fragmentation^{7,8}, whereas non-RCD is generally referred to as "necroptosis", which is the opposite of RCD in that it is characterised by cellular swelling and plasma membrane damage leading to leakage of cytoplasmic constituents to the outside of the cell⁹. Necroptosis is in a special position among all cell death processes, it has been shown to be a type of RCD, but exhibits non-RCD similar morphological features, triggers an inflammatory response in the body and is part of autoimmune or inflammatory diseases¹⁰. The regulation and interactions between different cell death pathways are highly complex and context-dependent. The interplay between apoptosis, necroptosis, and other cell death processes can be influenced by multiple factors. It is now known that when levels of RIPK3, a key regulator of necroptosis, are elevated, it can inhibit the activation of caspase-8 and shift the cell towards necroptosis rather than apoptosis¹¹⁻¹³. When caspase-8 or cellular inhibitor of apoptosis proteins (cIAPs) are in a high level state, intracellular and extracellular stimuli activate the apoptosis pathway via TNFR. And when caspase-8 or cIAPs are absent, external stimuli promote the activation of necroptosis⁹. Toll-like receptors (TLRs) play a role in the innate immune response by detecting changes in the internal environment of an organism, such as the presence of pathogen-associated molecular patterns (PAMPs) or danger-associated molecular patterns (DAMPs)^{9, 14}. Activation of TLRs recruits the bridging proteins FADD, TRADD and TRIF, which interact with RIPK1 and caspase-8¹⁵. RIPK1 recruits RIPK3 to form a complex after activation of the NF- κ B inflammatory pathway through a process of ubiquitination and deubiquitination. Its complex then recruits MLKL and undergoes phosphorylation. The phosphorylated MLKL then oligomerises and translocates into the plasma membrane to form a pore that mediates the influx of extracellular material into the cell, causing it to swell and rupture¹⁴. Notable advances on the role of necroptosis in disease progression and corresponding interventions have been made in the tumor, skin and cardiovascular fields^{11, 13, 16-24}.

Activation of the necroptosis pathway during ARDS can indeed lead to the

release of pro-inflammatory signals. This can contribute to the overall inflammatory response in ARDS, amplifying the inflammatory cascade and worsening the condition²⁵⁻²⁷. Moreover, cell death resulting from necroptosis can exacerbate tissue damage and inflammation in ARDS. The release of intracellular contents from dying cells, along with the activation of immune cells in response to necroptotic cell death, can further contribute to tissue injury, inflammation, and the progression of ARDS. These dual effects, both the release of pro-inflammatory signals and the exacerbation of tissue damage, highlight the significant role that necroptosis can play in the development and severity of ARDS. By understanding the intricate process of necroptosis in ARDS, we can gain valuable insights into the underlying mechanisms of the disease. This understanding opens up opportunities to identify and target specific molecules or pathways involved in necroptosis, which in turn can lead to the development of novel therapeutic strategies for ARDS.

In this study, we suggest that necroptosis is involved in acute lung injury. This is characterized by the phosphorylation and complex formation of RIPK1, RIPK3, and MLKL. Additionally, the activation of TLR4 downstream signaling pathways including TRIF and Myd88 are also implicated. To target necroptosis for the treatment of acute lung injury, we have developed a lipid micelle encapsulated MLKL inhibitor, GW806742X, which can effectively alleviate airway inflammation and injury by specifically binding to alveolar type II epithelial cell as a targeted therapy. Our research has uncovered new pathological mechanisms underlying acute respiratory distress syndrome and has paved the way for novel targeted treatment strategies.

2. Results

2.1. Transcriptomic analysis revealed the activation of the necroptosis pathway in acute lung injury

To investigate whether the necroptosis pathway is activated in the pre-ARDS phase, we administered airway drops to mice using **Lipopolysaccharides** (LPS) to simulate the acute lung injury that occurs during the early phase of ARDS in the clinic. After 24 hours, lung tissue was taken for RNA-sequencing, western blotting and tissue

immunofluorescence. Upon visualisation of the sequencing results, genes associated with necroptosis, Myd88, Trif, Ripk1, Ripk3, Mlkl, Tlr4, were not only all up-regulated, but were also located in bands with p-values <0.05 and $\log_2FC >1$, as observed in the lung tissues of mice with ALI (**Figure S1A**). To gain a comprehensive understanding of the necroptosis pathway, a full list of genes associated with necroptosis was downloaded from the Kyoto Encyclopedia of Genes and Genomes (KEGG). This list was then cross-referenced with extensive literature review to ensure its accuracy and completeness. Subsequently, this curated gene list was utilized to generate a heat map representing mRNA expression levels in both ALI models and control mice (**Figure 1A**). This analysis helps to elucidate the changes occurring in other genes within the necroptosis pathway and provides insights into their potential involvement in ALI. The results of the heatmap showed that the vast majority of the genes visualised showed a significant up-regulation in expression, suggesting the possibility of high activation of the pathway in which they are located. Following the bulk-RNA sequencing analysis of differential genes, a Gene Ontology (GO) enrichment analysis was conducted (**Figure S1B**). This analysis revealed the presence of eight functional clusters associated with death, three of which showed significant relevance to the necroptosis pathway. Additionally, Gene Set Enrichment Analysis (GSEA) was performed on these gene sets, further confirming the presence and activation of the necroptosis (**Figure 1B**). Meanwhile, based on GSEA analysis of GO, we found five most statistically significant death-associated functional pathways, and the fifth one was extremely strongly associated with necroptosis (**Figure S1C**). Moreover, the representative genes such as Myd88, Trif, Ikk, Irf3, as well as Ripk1, Ripk3, Mlkl were analyzed for interaction using correlation analysis based on their Fragments Per Kilobase of exon model per Million mapped fragments (FPKM) (**Figure 1C**). The results revealed a substantial correlation among these genes, suggesting an interaction and influence between them.

2.2. Necroptosis activation and expression in acute lung injury

We further examined the protein expression of key factors in the necroptosis pathway. The western blotting analysis of mice lungs revealed a significant increase in the

expression of phosphorylated RIPK1 (**Figure 2A-C**), RIPK3 (**Figure 2D-F**), and MLKL (**Figure 2G-I**) proteins in the ALI group compared to the control group. Additionally, the paraffin immunofluorescence results of mouse lungs showed a marked increase in fluorescence intensity of p-MLKL, p-RIPK1, and p-RIPK3 in the ALI model samples in comparison to normal mice (**Figure 2J**), further supporting the activation of necroptosis in ALI. The results from the western blotting and immunofluorescence analyses indicate that the critical mediators RIPK1, RIPK3, and MLKL exhibit significantly increased levels of phosphorylation modification in the ALI model. In contrast, Caspase8 levels decreased as shown in western blotting in the ALI model (**Figure S2A-C**). Previous findings have suggested that phosphorylation modification of RIPK1, RIPK3, and MLKL proteins and decreased expression of Caspase 8 are essential and indispensable parts of the necroptosis pathway¹³, and these results are consistent with previous studies suggesting that there is activation of necroptosis early in the development of ARDS.

2.3. LPS induced phosphorylation of RIPK1/RIPK3/MLKL in epithelial cells

While it has been established that necroptosis is present in ALI, it remains unclear whether necroptosis is directly associated with epithelial cell damage and death in this condition. To investigate the presence of necroptosis within epithelial cells, we conducted an experiment where human airway epithelial (HBE) cells were exposed to LPS intervention. Specifically, we tested LPS at a concentration of 200 µg/ml for 24 hours in HBE cells, and assessed the protein expression levels using a Western blot analysis. The results revealed that only the phosphorylation levels of RIPK1 and RIPK3 were significantly elevated (**Figure S3A-G**), while the phosphorylation levels of MLKL did not show significant changes (**Figure S3H-K**). We thought that it might be the changes of cytokines in the internal environment after LPS stimulation, and the in vitro cultivation could not fully replicate the changes of these factors which led to the delay of necroptosis activation. To further investigate the potential activation of necroptosis within HBE cells under LPS intervention, we extended the exposure time to 48 hours using the same concentration of LPS. Western blot analysis revealed that

the protein content of p-RIPK1 (**Figure 3A-C**), p-RIPK3 (**Figure 3D-F**), and p-MLKL (**Figure 3G-I**) was significantly increased in the LPS-intervened group compared to the control group, as evidenced by the significantly darker bands observed in the former. From the immunofluorescence results of HBE cells, it can also be seen that there was a significant increase in the expression of p-MLKL, p-RIPK1 and p-RIPK3 in HBE cells after LPS intervention (**Figure 3J**). These results provide additional support for the existence of the necroptosis pathway in LPS-intervened HBE cells. However, it is important to note that there might be a delay in the activation of necroptosis in vitro. The delay in this case can be attributed to the fact that the changes occurring in the internal environment after an LPS intervention are not fully replicated in in vitro cultivation.

2.4. Both of RIPK1 and RIPK3 involve in MLKL phosphorylation in LPS treated epithelial cells

MLKL is located downstream of both RIPK1 and RIPK3 proteins in the necroptosis pathway. Phosphorylation of either RIPK1 or RIPK3 has now been found to initiate necroptosis in other diseases, but it has not been clarified in ALI exactly which initiates the process of necroptosis ^{28, 29}. To check the above, we used two inhibitors, Necrodtatin-1 (Nec-1) and GSK'872, to inhibit RIPK1 and RIPK3, independently. When HBE cells were intervened with Nec-1, GSK872 and LPS, samples from 48 hours were selected for WB assay, and the results showed that both inhibitors could cause the bands of the p-RIPK1, p-RIPK3 and p-MLKL to be significantly weakened (**Figure 4A-R**), suggesting that the amount of the p-RIPK1, p-RIPK3 and p-MLKL showed a significant decrease in the experimental group. The expression of p-MLKL in HBE induced by LPS decreases after intervention with inhibitors Nec-1 and GSK'872, respectively, according to fluorescence staining (**Figure S4**). These results indicate that the phosphorylation of both RIPK1 and RIPK3 is essential for the activation of MLKL.

2.5. The co-reaction between RIPK1, RIPK3 and MLKL in LPS induced necroptosis

Considering the currently proposed viewpoint that the combination of p-RIPK1 and p-RIPK3 recruits MLKL and triggers the phosphorylation process³⁰, and combining with the above experimental results showing that inhibition of both affects the phosphorylation of the other, we assume that there is a state in which p-RIPK1, p-RIPK3, and p-MLKL bind to each other to form a complex during the process of necroptosis. To validate our hypothesis, we performed immunoprecipitation assays targeting p-MLKL, p-RIPK1, and p-RIPK3 proteins individually (**Figure 5A-C**). The results demonstrated that all three proteins exhibited bands at their respective molecular weights, indicating their presence as a complex within the cells. These findings provide further evidence that p-MLKL, p-RIPK1, and p-RIPK3 can form complexes, supporting the notion that they potentially interact with each other to regulate necroptosis in HBE cells under LPS intervention. When immunoprecipitation was done with p-MLKL as the target, the expression of the three proteins was comparable in the co-precipitation group. In contrast, when immunoprecipitation was done with p-RIPK1 and p-RIPK3 as targets, the exposure of the bands of both in the co-precipitated group was significantly higher than that of the control group, whereas the bands of p-MLKL were equivalent or even lighter than that of the control group. Immunofluorescence staining performed on LPS-intervened HBE cells demonstrated the co-localization of p-RIPK1, p-RIPK3 and p-MLKL (**Figure 5D, Figure S5A**). The protein structures of p-RIPK1, p-RIPK3, p-MLKL were obtained from the Protein Data Bank (PDB), and the 3D conformations of the complexes formed by their binding were predicted using the cluspro website and the minimum binding energies were obtained for the relevant structures (**Figure S5B**), and finally their protein-binding sites were demonstrated by Pymol (**Figure 5E**). All of these results support that the protein of p-RIPK1, p-RIPK3, p-MLKL bind to each others.

2.6. Myd88 and TRIF are essential for necroptosis activation

MyD88 and TRIF are different adapter proteins that are recruited upon TLR4 activation. They enable signal transduction by mediating the recruitment and activation of downstream effector molecules. These two factors contribute significantly to the overall

cellular response to LPS stimulation^{31,32}. In order to investigate the role of TRIF and Myd88 in LPS induced necroptosis, we used siRNA to silence the expression of these genes individually. Samples were collected thereafter and subjected to a western blot assay after 48 hours. The results revealed a decrease in the expression of p-RIPK1 (**Figure 6A-C**), p-RIPK3 (**Figure 6D-F**), and p-MLKL (**Figure 6G-I**) in cells treated with siRNA-Myd88 or siRNA Trif (**Figure 6J-R**) compared to cells treated with LPS alone. In contrast, WB assays were performed on samples from LPS-intervened HBE cells at 6 h, 12 h and 24 h. We found that the levels of both p-IKK and p-IRF3 were elevated to variable degrees at 6 and 12 hours (**Figure 5S-T, Figure S6A-D**), suggesting that the pathways in which they occur are activated during necroptosis progression. These results suggest that both MYD88 and TRIF play a role in the activation of the necroptosis pathway.

2.7. Alveolar type II-targeted delivery of GW806742X via SPC antibody-modified lipid micelle

Given that one of the pathological features of ARDS is diffuse alveolar epithelial damage and that the alveolar epithelium plays a crucial role in maintaining gas exchange. When alveolar type I epithelial cells become damaged, alveolar type II epithelial cells can differentiate into alveolar type I epithelial cells, and take on the responsibility of secreting alveolar surface-active substances and reducing alveolar surface tension. Therefore, our objective was to target MLKL, a downstream molecule of RIPK1 and RIPK3, in order to inhibit necroptosis and ultimately treat airway inflammation. GW806742X, an MLKL inhibitor, has demonstrated the ability to inhibit necroptosis, indicating its potential as a treatment for inflammatory injuries³³⁻³⁶. However, its clinical application is hindered by its limited water solubility and low bioavailability. Consequently, there is a necessity for a novel targeted carrier system to enhance its therapeutic effectiveness. Therefore, we thought to prepare an SPC antibody-modified and GW806742X-loaded lipid nanoparticle targeting ATII cells, abbreviated as SPC-M-GW, with the abbreviation used to represent this targeted drug. We combined GW806742, DSPE-mPEG2000, and DSPE-PEG2000-MAL powders

and dissolved them in methanol. The lipids were then self-assembled around the inhibitor by dialysis displacement using PBS buffer. Subsequently, the antibodies were coupled to the lipids, and any unbound SPC antibodies were removed through dialysis. In which, SPC antibodies can target ATII cells, so it was attached to lipid micelles (**Figure 7A**). Finally, the lipid micelle was concentrated using ultrafiltration tubes, and its SPC-M-GW loading efficiency was determined using high-performance liquid chromatography (HPLC). The electron microscopy analysis revealed the distinctive features of the lipid micelle particles. They exhibited a spherical shape, maintaining a uniform and consistent size throughout. Additionally, the hydrodynamic particle size were evenly dispersed without any signs of aggregation (**Figure 7B**). The nanosizer of micelles was determined to be an average of 110nm using a nanosizer point analyser. Furthermore, micelles conjugated with antibodies exhibited a slightly larger diameter of 140 nm. Additionally, the micelles possessed a negative potential, indicating a balanced and stable system (**Figure 7C**). The use of fluorescent dye Cy5 to label micelles allows for visualization of their localization. Micelles modified with SPC antibodies show significantly enhanced cellular enrichment compared to unmodified micelles. To validate the targeted effect of the prepared lipid micelle, we conducted immunofluorescence staining of lung tissue and found that SPC-modified PEG-DSPE micelle containing CY5 can specifically bind to type II alveolar epithelial cells (**Figure 7D**). In the LPS-induced ALI mouse model, SPC-M-GW (1mg/kg) or GW806742X (1mg/kg) was administered via the airway 6 hours after LPS administration. Samples were collected 24 hours after the model was established. PCR analysis of mRNA levels in lung tissues revealed a significant elevation of inflammatory factors such as IL-1 β , IL-6, and TNF α in the ALI model. However, after treatment with SPC-M-GW, there was a notable decrease in these factors, further confirming the effective alleviation of airway inflammation by the drug (**Figure. 7E**). Through HE staining of lung tissue, we observed that the lung alveoli in the control group of mice displayed a normal structure with thin alveolar walls and no presence of inflammatory cells in the alveolar cavity. In contrast, the mice in the ALI group showed significant alveolar collapse, loss of epithelial cells, accumulation of inflammatory cells in the alveolar lumen, and

disruption of the continuity of the hyaline membrane, along with localized thickening (**Figure 7F**). No inhibitory effect of GW806742X was observed in alleviating inflammation-induced damage in lung tissues in 1mg/Kg, and unmodified antibody M-GW only has a weak effect. Surprisingly, administration of SPC-M-GW at a dose of only 1 mg/kg demonstrated significant inhibition of inflammatory cell aggregation, interstitial edema, and improvement in capillary dilatation, congestion, and haemorrhage (**Figure 7F-G**). These results indicate that AT2-targeted delivery of necroptosis inhibitor via SPC antibody-modified lipid micelle can significantly alleviate airway inflammatory injury.

3. Discussion

ALI/ARDS is a condition characterized by widespread damage to the airways and an inflammatory response. At the same time, necroptosis refers to a type of cell death that can be modified and is also accompanied by a severe inflammatory response. As a result, the damage to the airways and the inflammatory reactions mutually exacerbate each other, leading to the progression of the disease. In this study, we have shown that necroptosis is present in ALI and triggers downstream responses by phosphorylating RIPK1 and RIPK3 simultaneously (**Figure 8**). We also elucidated the presence of p-RIPK1, p-RIPK3, and p-MLKL complexes, further indicating the interaction of these phosphorylated proteins. Additionally, our findings indicated that in LPS-induced ARDS, TLR4 triggers necroptosis by the combined action of two downstream factors, MYD88 and TRIF. We have also identified the most effective targeted therapy for inhibiting the necroptosis with lipid micelle capsulated GW806742X for ALI.

It was proposed that when cells undergo necroptosis, it is LPS that activates its downstream TIRAP and MYD88 in the plasma membrane or activates the TRAM and TRIF pathways by stimulating the TLR4 receptor³⁷⁻³⁹. However, it is noteworthy that dual signalling mediated by TRIF and MYD88 has received little attention in previous studies⁴⁰⁻⁴². Moreover, it has been clearly demonstrated in previous studies that TRIF-mediated signalling is non-MYD88-dependent and that the signalling cascade it mediates occurs after endocytosis, whereas TLR4-mediated receptor endocytosis

blocks MYD88-dependent signalling^{37, 43}. However, we observed inhibition of the necroptosis pathway in HBE cells when either MYD88 or TRIF was blocked. Additionally, in our study, we observed an increase in the phosphorylation of IKK and IRF3 downstream of MYD88 and TRIF after the intervention with LPS⁴⁴. The concurrent activation of both pathways may be associated with the ability of LPS to evade endosomes created through TLR4-mediated endocytosis, or with the reduction of intracellular CD14. Research has revealed that the activation of TRIF relies heavily on intracellular CD14, and when intracellular CD14 levels are depleted, cellular TLR4 signaling shifts towards downstream transmission via the MYD88 pathway^{37, 45, 46}. However, the sequencing of activation for the MYD88 and TRIF pathways in the two aforementioned theories contradicts the proposition put forth by Reynoso et al. They suggest that MYD88 is responsible for initiating early responses, while TRIF is involved in later responses⁴⁷. These disparities in results could arise from variances in cell types, the timing of interventions, or even variations in the dosage of medication, potentially contributing to divergent outcomes.

In the present study, we demonstrate that phosphorylation of both RIPK1 and RIPK3 is an integral part of necroptosis in the development of ARDS, and that intervention in either of the two can have a dramatic effect on the other. And as the published papers discovered that not only RIPK1 is not essential for necroptosis, but also inhibition of the kinase activity of RIPK1, which converts the necroptosis pathway to RIPK1-independent necroptosis, results in the production of large amounts of DAMP by the cells, which induces a severe inflammatory response^{14, 48-51}. It has also been pointed out that RIPK3 is involved in the transmission and activation of various signalling pathways, and its inhibition will promote the inflammation-related STING pathway, while the inhibition of MLKL membrane translocation can inhibit the STING signalling pathway, which can inhibit the body's inflammatory response to a certain extent^{29, 51}.

Currently, lipid micelles serve as a widely employed drug delivery technology due to their lipophilic properties. This technology significantly enhances the efficiency of cellular drug uptake and simultaneously minimizes the undesired degradation of

drugs within the body^{52, 53}. Lipid micelles targeted delivery also enhances the absorption efficiency of drugs by the target cells, leading to lower dosages and reduced side effects in the treatment of diseases⁵⁴. Lipid micelles can be further enhanced in targeting specificity through chemical modification by attaching targeting ligands. In our study, GW806742X, an MLKL inhibitor, was incorporated into SPC antibody-modified lipid micelles for targeted delivery to alveolar type II epithelial cells using SPC antibody named SPC-M-GW. The micelles allowed for efficient release of a concentrated dose of GW806742X into the cytoplasm of these cells, enhancing intracellular uptake. In addition, due to the targeted function of the micelles, SPC-M-GW can effectively suppress MLKL membrane translocation, thereby achieving the purpose of inhibiting the process of necroptosis. And finally, it was demonstrated that this targeted medicine successfully alleviated airway inflammation-associated damage by specifically inhibiting necroptosis in acute lung injury. This provides a very promising approach to the treatment of acute lung injury.

4. Conclusion

Necroptosis acts as a double-edged sword under various pathophysiological conditions. On the one hand, necrotic cell death is a conservative defense mechanism that can prevent pathogen invasion by limiting the lifespan of infected cells. On the other hand, excessive necroptosis can cause damage to the organism and promote the progression of diseases, such as airway inflammatory diseases^{22, 55}. Therefore, exploring the pathogenesis of diseases by targeting the process of necroptosis and developing targeted drugs holds significant scientific value and clinical significance. The injury to airway epithelial cells is an important pathological feature of acute respiratory distress syndrome. Through RNA-seq, we identified the activation of the necroptosis pathway in the lung tissue of a mouse model of ALI induced by LPS. Furthermore, using the ALI mouse model and lung epithelial cell lines, we found that necroptosis mediated the progression of the disease, with the activation and formation complex of RIPK1, RIPK3, and MLKL. Our research also confirmed the involvement of both Myd88 and TRIF in the activation of LPS-TLR4 signaling pathway. Based on this, we developed a novel

lipid micelle based-drug deliver system targeting MLKL in alveolar type II epithelial cells, and applied it to the LPS-induced ALI model. This system effectively alleviated epithelial cell damage by selectively inhibiting necroptosis, achieving the goal of treating inflammatory injury. Our study not only revealed the role of necroptosis in ALI/ARDS but also provided a new targeted therapy, offering a new strategy for the treatment of ALI/ARDS.

5. Experimental Section

Mice: Male C57BL/6 mice, aged 6-8 weeks, weighing between 16-24 grams, were purchased from SHANGHAI SLAC LABORATORY ANIMAL CO.LTD., and were acclimated for one week in the animal laboratory of the Clinical Research Center of the Second Affiliated Hospital of Zhejiang University School of Medicine. The mice were housed under specific pathogen-free (SPF) conditions, with a temperature of $25 \pm 1^\circ\text{C}$, humidity of $60 \pm 5\%$, a 12 hour light/dark cycle, and free access to water. All animal experiments were approved by the Ethics Review Committee of the Second Affiliated Hospital of Zhejiang University School of Medicine (No. 2022.050).

Preparation Of SPC-M-GW: Since GW806742X is not readily dissolved in water, a solvent exchange method was used to prepare the lipid micelle. The lipid compositions used were as follows: DSPE-PEG2000 (Tanshui Technology Co , a80010201-2000-1g) and DSPE-PEG2000- MAL (Yusi ,YS-D223025) (mass ratio,4:1). All lipid materials and GW806742X (mass ratio, 20:1) was dissolved in methanol, and put in dialysis bag (Molecular weight cut-off, MWCO 100-500, Yuanye, SP131057-0.5m) at 4°C to be dialyzed against PBS, pH 7.5. The outer phase was replaced with fresh buffer solution at 2, 4,6, 8, and 12 h. After 24 h, the lipid materials and GW806742X were self-assembled into lipid micelle. The SPC Antibody-(#DF6647) were dissolved in the HEPES buffer (1 M, pH 7.5), and the lipid micelle was added to a 50 per cent glucose solution so that it ultimately contains 5 percent glucose⁵⁶. Antibody dilutions were mixed with glucose-containing lipid micelle in the ratio of 1:2 molar ratio of antibody to DSPE and placed at 4°C on 3D suspension instrument. Sample was collected after 6

hours and placed in a dialysis bag (MWCO 1000K, Yuanye, SP131486-0.5m) at 4 °C for 24 hours. The outer phase was also replaced with fresh buffer solution at 2, 4, 6, 8, and 12 h. The collected samples were finally concentrated and purified by placing them in ultrafiltration tubes with a pore size of 3KDa.

Characterization Of SPC-M-GW: The drug loading of SPC-M-GW was determined by high performance liquid chromatography (HPLC). The SPC-M-GW were solubilized with methanol before HPLC analysis. The Daojin LC-20AD HPLC system equipped with a Ultimate Plus-C18 4.6*100mm 3.5um column was used. The column was eluted with 0.1%TFA acetonitrile at a flow rate of 1.0 mL/min. PCT was detected at 254 nm. The SPC-M-GW drug loading capacity was determined using a calibration curve obtained in the same conditions using standard concentrations of GW806742X (ranging between 10 and 60 µg/mL); the correlation coefficient R² was 0.9981. The particle size and zeta potential were measured using a laser particle sizer (Malvern, ZSV3100) and photographed using an electron microscope (HITACHI, H-7650), and the particle sizes were measured to be 110 and 140 nm, and the potentials to be -27.35 mV and -19.13 mV for M-GW and SPC-M-GW respectively. SPC-M-GW loading efficiency (LE) was calculated as follows:

$$LE (\%) = (\text{Mass of GW806742X in 1 ml of solution} / \text{Mass of SPC-M-GW in 1 ml of solution}) \times 100\%$$

To confirm the successful connection of SPC antibody, a BCA detection kit (yamei, ZJ101) was used. Calculations of grafting rate (GR) were performed by the following methods:

$$GR (\%) = (\text{Total mass of antibody} / \text{initial mass of antibody input}) \times 100\%.$$

The final test results were: LE=28.67%, GR=30.23%.

Preparation And Characterization Of SPC-M-CY5: In accordance with the above method, DSPE-PEG2000 (Tanshui Technology Co, a80010201-2000-1g), DSPE-PEG2000-MAL (Yusi, YS-D223025), DSPE-PEG2000-NHS (Ponsure Biological, PS2-HE1-2K-100mg) and CY5-NH₂ (MeloPEG, 431201-2) were added according to a

molar ratio of 20:5:1:1 to prepare and assay SPC-M-CY5.

In vivo animal experiments: Mice were anesthetized with intraperitoneally 10mL/kg of pentobarbital. For the acute lung injury, mice were administrated with 50ul solution of LPS (Sigma-Aldrich, L9143) in a concentration of 10mg/kg via the airway. To treat ALI model, mice were administrated with 60ul solution of SPC-M-GW in a concentration of 2mg/kg, 1mg/kg, 0.5mg/kg or via the airway after LPS intervention 6 hours. For the control group, mice were treated with an equal volume of PBS. Lung tissues were collected after 24 hours of LPS, GW806742X or PBS treatment for the further experiments.

Cell Culture and LPS Stimulation: Human Bronchial Epithelial cells (HBE) were cultured in 1640 medium (Gibco, C11875500BT) supplemented with 10% fetal bovine serum (FBS, meilunbio, 9048-46-8) in 10 cm culture dishes. The cells were incubated in a constant temperature CO₂ incubator at 37°C with 5% CO₂. The experimental group was treated with 200 µg/ml LPS (Sigma-Aldrich, L9143), while the control group received the same volume PBS. The cells were stimulated for 24 or 48 hours before sample collection.

siRNA Transfection: The siRNA of MYD88 and TRIF were transfected into cells using GP-transfect-Mate. After 6 hours, the liquid was changed to 1640 medium containing 10% FBS to continue incubation for 12 hours, then 1640 medium containing 200ug/ml LPS and 10% FBS was given to the LPS intervention group and siRNA intervention group as described above, and the samples were collected after 48 hours of incubation, and stored at -80°C. (TRIF-S:CGGAACAGA AAUUCUAUAATT; TRIF- AS:UUA UAGAAUUUCUGUCCGTT; MYD88- S:GACUUUGAGUACUUGGAGATT; MYD88- AS:UCUCCA AGUACUCAAGUUCTT).

Quantitative real-time polymerase chain reaction: The lung tissues were obtained in animal experiences for PCR assay. Trizol reagent (Vazyme, R411-01) was used to

extract total RNA from lung tissue according to the manufacturer's instructions. The PrimeScript RT Reagent Kit was used to make cDNA (AG, AG11706). On QuantStudio 5 PCR equipment, qRT-PCR was performed using AceQ Universal SYBR qPCR Master Mix (Vazyme, Q111). The following were the primers utilized in this study: mGAPDH-F: CATCACTGCCACCCAGAAGACTG; mGAPDH-R: ATGCCAGTGAGCTTCCCGTTCAG; mL-6-F: TACCACTTCACAAGTCGGAGG C; mL-6-R: CTGCAAGTGCATCATCGTTGTTC; m-TNF α -F: GGTGCCTATGTCTCA GCCTCTT; m-TNF α -R: GCCATAGAAGTATGATGAGAGGGAG; m-IL1 β -F: TGGA CCTTCCAGGATGAGGACA; m-IL1 β -R: GTTCATCTCGGAGCCTGTA GTG. Relative quantification was determined using the $2^{-\Delta\Delta C_t}$ method.

Inhibitor Intervention: The cells were randomly divided into control, LPS intervention and GSK'872 or Necrodtatin-1 intervention group. GSK'872 (selleck, S8465) and Necrodtatin-1 (selleck, S8037) was dissolved in DMSO at a concentration of 50 mM and diluted to 5 μ M and 10 μ M in 1640 medium containing 200ug/ml LPS and 10% FBS for intervention group. Cellswere treated with 5 μ M GSK'872 or 10 μ M Necrodtatin-1 for 48 h and were subsequently collected for the next experiments.

Western Blotting: To extract total protein, HBE cells and lung from mice were lysed using RIPA buffer (sparkjade, EA0002) containing protease inhibitor (beyotime, ST507-10ml) and phosphatase inhibitor (Biosharp, BL615A) and quantified using the BCA Protein Quantification Kit (Yamei, ZJ101). The concentration of protein stock solution was adjusted to 3ug/ul after fully denaturing the protein using loading buffer (fude biological, FD002). The sample was separated by 10% sodium dodecyl sulfate-polyacrylamide gel electrophoresis (SDS-PAGE) and transferred onto 0.45 μ m polyvinylidene difluoride (PVDF) membranes (Merck, IPVH00010). The membranes were blocked with 5% FBS and incubated with primary antibodies to RIPK1 (CST, #3493), p-RIPK1 (affinity, af2398), RIPK3 (Abcam, ab62344) , p-RIPK3 (Abcam, ab209384) , MLKL (Merck, SAB5700808), p-MLKL (Abcam, ab196436 and affinity, AF7420), Casp8 (proteintech, 13423-1-AP). Then the membranes were incubated with

HRP-conjugated Affinipure Goat Anti-Mouse IgG (H+L) (proteintech,SA00001-1) and HRP-conjugated Affinipure Goat Anti-Rabbit IgG (H+L) (proteintech,SA00001-2). The antibody to β -Actin (sigma, A2228), GAPDH (Abcam, ab8245) were used as internal control for protein quantification. Band intensity was quantified using a chemiluminescence imaging system (BIO-RAD), Image J software, and GraphPad Prism 9.

Co-Immunoprecipitation: HBE were cultured in 1640 medium (Gibco, C11875500BT) containing 10% FBS (meilunbio) and 200ug/ml LPS in 10 cm culture dishes. After 48 h, cells were washed with cold PBS and fully lysed using IP lysis solution (beyotime, P0013) containing protease inhibitor (beyotime, ST507-10ml) and phosphatase inhibitor (Biosharp, BL615A). After centrifuged at 13000 \times g for 15min at 4 °C, per 750ug protein was added with 25 μ L Pierce™ protein A/G Agarose (thermo fisher, 88802) and IgG (CST, 3900S) or p-MLKL (Abcam, ab196436 and affinity, AF7420) or p-RIPK1 (affinity, af2398) or p-RIPK3 (Abcam, ab209384) overnight at 4 °C. The next day, the immunoprecipitants were washed five times and boiled with 2 \times SDS Loading Buffer. Western blotting assay was conducted according to the protocol above.

Tissue Immunofluorescence: Samples from both ALI and control were treated following a previously protocol. The p-RIPK1 (affinity, af2398), p-RIPK3 (Abcam, ab209384), p-MLKL (Abcam, ab196436, and affinity, af7420), and incubated with Goat Anti-rabbit IgG (HF7420) were used as primary antibody and Goat Anti-rabbit IgG (HRP) (Abcam , ab205718), FTIC (aatbio, 11060), CY3 (aatbio, 11065) and CY5 (aatbio, 11066) were used as fluorescent labelling. After the final staining, the samples were scanned using fluorescence microscope (Olympus#BX53) and processed by CaseViewer (2.4).

Cell Immunofluorescence: HBE cells from LPS, GW806742X, and control were treated following a previously protocol. The p-RIPK1 (affinity, af2398), p-RIPK3 (Abcam, ab209384), p-MLKL (affinity, af7420), and incubated with Goat Anti-rabbit IgG

(HF7420) were used as primary antibody and Goat Anti-rabbit IgG (HRP) (Abcam, ab205718), FTIC (aatbio, 11060), CY3 (aatbio, 11065) and CY5 (aatbio, 11066) were used as fluorescent labelling. After the final staining, the samples were scanned using fluorescence microscope (Olympus#BX53) and processed by CaseViewer (2.4).

Protein-Protein Docking: The 3D structures of RIPK1(6nw2) protein⁵⁷ and RIPK3-MLKL (7mon) protein⁵⁸ complex and their related information were obtained by searching the PDB database⁵⁹⁻⁶¹. The local files of the two 3D structures were uploaded to the cluspro website, and the protein docking results containing 10 docking models with Job Details of 991258 were obtained, and their docking model scores were compared, and the model with the lowest energy value and the lowest central energy value were selected⁶²⁻⁶⁵. The docking keys were visualised using PyMOL (Anaconda 2, 64-bit).

Hematoxylin-eosin Staining: Hematoxylin-eosin (HE) staining was performed to explore damage in the lung tissue. Prior to histological analysis, lungs were immersed in 4% paraformaldehyde for 24 hours. Paraffin-embedded sections (3 µm thickness) were prepared and subjected to HE staining. The sections were placed in aqueous hematoxylin solution for 3-5 minutes. Then, they were washed with running water for 1-2 minutes, dehydrated in alcohol for 1-2 minutes. Finally, sections were treated with permeation and fixation. Light microscopy images were obtained using an microscope (Olympus, CX-31). The degree of congestion, edema, exudation, and tissue damage in the lung tissue could be evaluated.

Bulk RNA-sequencing: Control and Acute Lung Injury group mouse lung tissue samples were used for transcriptome sequencing. Raw reads were obtained using the BGISEQ platform and filtered to remove low-quality reads, adapter contamination, and an unknown base N content. Then, clean reads were then mapped to gene symbols obtained from the National Center for Biotechnology Information (NCBI) database. DESeq2 package in R (version 4.2.0) was used to identify differentially expressed

genes (DEGs) and their values. Genes related to the Necroptosis pathway were obtained from the KEGG database. Heatmaps were generated by intersecting the Necroptosis pathway genes with the DEGs and plotting their fpkm values. Volcano plots were generated using the ggplot2 package to visualize DEGs with \log_2 Fold Change < -1 or > 1 and $p\text{-value} < 0.05$. Key genes in the Necroptosis pathway were fluorescently labeled. GSEA enrichment analysis of differential genes was performed using clusterProfiler and ReactomePA package, and the data of the pathway named necroptosis was extracted and visualised using the gseaplot2 function to obtain a GSEA biplot, while the pathway related to death was extracted and plotted as a bolliplot.

Statistical analysis: Data were presented as mean \pm standard deviation (SD) of minimum three replicates unless otherwise indicated. Comparisons between two groups were using the two-tailed Student's t-test, and a $p\text{-value} < 0.05$ was considered statistically significant. Comparisons among multiple groups were conducted by one-way ANOVA and P-values of less than 0.05 were considered statistically different. Data were analyzed using Prism software (GraphPad 9.0, San Diego, CA). For RNA sequencing data, statistical analyses were described as above.

Figure-1. RNA-seq analysis of lung tissues of ALI. (A) Heat mapping of Bulk RNA-sequence results revealed that the expression of genes involved in the necroptosis pathway was altered in both ALI groups. (B) GSEA analysis using Bulk RNA-sequence data demonstrated that the necroptosis pathway was significantly activated in the lung tissue of mice in the ALI group ($p < 0.05$). (C) Heatmap after correlation analysis based on Fragments Per Kilobase of exon model per Million mapped fragments (FPKM) of genes. Positive correlations are shown in red, negative correlations in blue, and the size of the circles and numbers represent the magnitude of the correlation. Samples from 5 mice per group were analyzed.

Figure-2. Expression of necroptosis in LPS-induced acute lung injury. (A, D, G) Validation of protein expression levels of RIPK1, p-RIPK1, RIPK3, p-RIPK3, MLKL, and p-MLKL in mouse lung tissue with ALI using Western blotting. (B,C,E,F,H,I) Quantitative statistical analysis of Western blotting results using image J software. (J) Fluorescence microscope images of paraffin tissue sections from mouse lung tissues stained with anti-p-RIPK1 (green), anti-p-RIPK3 (grey), anti-p-MLKL (red), and DAPI (blue) for visualization, with a scale bar indicating 50 μm . ns: no

significant; *P<0.05, ***P<0.001.

Figure-3. PIPK1, RIPK3 and MLKL expression in LPS treated HBE cells in vitro. (A, D, G) Western blotting was performed to verify the expression of RIPK1, p-RIPK1, RIPK3, p-RIPK3, MLKL, and p-MLKL in HBE cells treated with 200 µg/ml of LPS for 48 hours. (B, C, E, F, H, I) Quantitative statistical analysis was conducted to analyze the results of Western blotting. (J) Fluorescence microscope images were captured for HBE cells treated with anti-p-RIPK1 (green), anti-p-RIPK3 (red), anti-p-MLKL (pink), and DAPI (blue) in both the control and LPS intervention groups. Scale markers of 30 µm were included in the images. ns: no significant; *P<0.05, **P<0.01; ***P<0.001.

Figure-4. The expression of necroptosis following the inhibition of RIPK1 and RIPK3 respectively. (A-C) Western blotting was performed to verify the expression of RIPK1, p-RIPK1, RIPK3, p-RIPK3, MLKL, and p-MLKL in HBE cells treated with LPS at a concentration of 200 µg/ml or LPS+Nec-1 intervention for 48 hours. (D-I) Quantitative statistical analysis was conducted using Image J software to analyze the results of the Western blotting. (J-L) Western blotting was performed to verify the expression of RIPK1, p-RIPK1, RIPK3, p-RIPK3, MLKL, and p-MLKL in HBE cells treated with LPS at a concentration of 200 µg/ml or LPS+GSK'872 intervention for 48 hours. (M-O) Quantitative statistical analysis of the Western blotting results was performed using Image J software for RIPK1, p-RIPK1, RIPK3, p-RIPK3, MLKL, and p-MLKL expression in the HBE cells with the respective interventions. ns: no significant; *P<0.05, **P<0.01; ***P<0.001.

Figure-5. The co-location of p-RIPK1, p-RIPK2 and p-MLKL during necroptosis process. (A-C) Immunoprecipitation assays were conducted on HBE cells treated with LPS at a concentration of 200 µg/ml for 48 hours, using p-MLKL, p-RIPK1, and p-RIPK3 as targets. The results showed that p-RIPK1, p-RIPK3, and p-MLKL could be used as targets to precipitate the other two proteins, indicating their interaction or complex formation. (D) Fluorescence microscopy images were captured for HBE cells treated with LPS, showing the localization of p-RIPK1 (green), p-RIPK3 (red), p-MLKL (pink), and DAPI (blue) as markers. The images revealed the co-localization of these proteins within the cells. (E) A 3D model was generated to simulate the binding states of RIPK1 (blue) and the RIPK3-MLKL complex (pink). The model was obtained from the Protein Data Bank (PDB) database using the ClusPro website, which predicts protein-protein interactions and complex structures.

Figure-6. The downstream signaling in LPS induced necroptosis. (A,D,G) Western blotting to verify RIPK1, p-RIPK1, RIPK3, p-RIPK3, MLKL and p-MLKL expression in HBE cells with LPS 200µg/ml or LPS+MYD88 siRNA-treatment, (B,C,E,F,H,I) quantitative statistical analysis of

western blotting using image J respectively. (J,M,P)Western blotting to verify RIPK1, p-RIPK1, RIPK3, p-RIPK3, MLKL and p-MLKL expression in HBE cells with LPS 200 µg/ml or LPS+TRIF siRNA, (K,L,N,O,Q,R) quantitative statistical analysis of western blotting using imageJ. (S,T) Western blotting was performed to verify the expression of IKK, p-IKK, IRF3, and p-IRF3 in HBE cells treated with LPS at a concentration of 200 µg/ml for 6, 12, and 24 hours, respectively. ns: no significant; *P<0.05, **P<0.01; ***P<0.001.

Figure-7. The effect of lipid micelle-encapsulated GW806742X targeting MLKL in alveolar type II epithelial cells in acute lung injury. (A) The process of preparing SPC-modified PEG-DSPE micelle containing GW806742X(SPC-M-GW) targeting alveolar type II epithelial cells. (B) Transmission electron micrographs of lipid nanomicelles with unconjugated antibody (M-GW) and conjugated antibody (SPC-M-GW). (C) Statistical plots of the hydration diameters and zeta potentials of the M-GW and the SPC-M-GW. (D) Fluorescence microscope images of mouse lung tissue stained with DAPI, SPC and Cy5 can be seen to appear in aggregates in alveolar type II epithelial cells. (E) The changes of mRNA levels for IL-1 β , IL-6, TNF- α in the lung tissues in control, ALI and SPC-M-GW 1 mg/kg treatment group. (F) Microscope images of HE-stained lung tissues of mice with LPS, GW (GW806742X), M-GW or SPC-M-GW, the dose for all three groups of GW is 1mg/kg, lung injury was scored (G). Samples from 5-6 mice per group were analyzed.

Figure-8. Schematic illustrating the potential targeted therapy for necroptosis in acute lung injury. The pathological characteristics of ALI/ARDS involve diffuse inflammatory injury of the airway epithelial/endothelial cells, which is mediated by necroptosis. During this process, phosphorylation and complex formation of RIPK1, RIPK3, MLKL, as well as activation of TLR4 downstream signaling pathways involving TRIF and Myd88 are involved. The lipid micelle-encapsulated GW806742X (SPC-M-GW) that specifically targets MLKL in AT2 cells have successfully used it to treat acute lung injury mice.

Supporting Information

Figure-S1. Analysis of lung tissue sequencing results in control and ALI mice. (A) Volcano plots were generated using Bulk RNA-seq data and analysis of variance. Gene names highlighted in black represent those with a p-value < 0.05 and a log₂FoldChange > 1, while gene names in grey represent those with a p-value < 0.05 and a log₂FoldChange < 1. (B) A Lollipop plot was created using Bulk RNA-seq data and GO enrichment analyses. Specifically, we extracted and displayed the functional set related to death. (C) A GSEA map was generated using Bulk RNA-seq data and GO-based GSEA

enrichment analysis. From this map, we extracted the top five functional sets associated with death for presentation.

Figure-S2. Expression levels of caspase-8 were significantly higher in lung tissues of ALI mice.

(A-C) The expression of Caspase-8 and cleaved-Caspase-8 in mouse lung tissues in ALI was confirmed through western blotting. Quantification of their expression using image analysis software J demonstrated a significant reduction in the levels of Caspase-8.

Figure-S3. ALI mice showed significant changes in p-RIPK1 and p-RIPK3 in 24-hour lung tissue samples. The expression of RIPK1, p-RIPK1(A-D), RIPK3, p-RIPK3 (E-G), MLKL, p-MLKL (H-K), in HBE cells was confirmed using western blotting after 24 hours of LPS intervention.

Figure-S4. GSK'872 and Nec-1 significantly reduced p-MLKL levels in HBE. Fluorescence microscope images were captured to observe HBE cells after treatment with p-MLKL (green) and DAPI (blue) as markers. Scale markers of 100 μ m were included for reference.

Figure-S5. p-RIPK1, p-RIPK3 and p-MLKL can be present as complexes. (A) Fluorescence microscopy images of HBE cells intervened with LPS were processed with p-RIPK1 (green), p-RIPK3 (red), p-MLKL (pink), and DAPI (blue) as markers. The fluorescence intensity was quantified using imageJ, and the results were plotted using Prism. Consistent fluorescence peaks were observed for p-RIPK1 (green), p-RIPK3 (red), and p-MLKL (pink). (B) The binding energies required for the formation of RIPK1 and RIPK3-MLKL complexes were determined using the first 10 binding modes provided by the ClusPro website.

Figure-S6. The phosphorylation levels of IKK and IRF3 were significantly increased by 6 h of LPS intervention in HBE. Quantitative statistical analysis of western blotting results for the expression levels of p-IKK (A), p-IRF3 (B), IKK (C), and IRF-3 (D) in HBE cells following LPS treatment at various time points.

Funding Declaration

This work was supported by the National Key Research and Development Program of China (2021YFC2501800), the National Natural Science Foundation of China (82172163, 82272182, 82372185, 82302485), and the Medical and Health Research Program of Zhejiang Province (No. 2023KY769).

Acknowledgements

We thank Ms Yuqiong Xie, Ms Meirong Yu, Ms Yonglan Zhu, Ms Amin Liu, Ms Yuelan Chen, Clinical Research Center, The Second Affiliated Hospital of Zhejiang University for laboratory management and technical support.

Authors' contributions: TBP contributed to the overall conceptualization and design of the research, DY was responsible for designing experiments related to pharmaceutical materials. KZY, ZQC and HQY carried out the animal experiments, in vitro cell experiments and bioinformatics analysis. TBP, DY, ZZC, XNX, and KZY analyzed the data and provided explanations. KZY, XNX and TBP prepared the figures and drafted the manuscript. ZZC, CW, ZGS and XNX performed the critical reading of the manuscript. TBP, DY and ZZC edited and revised the manuscript. All authors have read and agreed to the published version of the manuscript.

Conflict of Interest

The authors declare that there is no conflict of interest.

Data Availability Statement

The data that support the findings of this study are available from the corresponding author upon reasonable request.

Reference:

1. Thompson BT, Chambers RC, Liu KD: Acute Respiratory Distress Syndrome. The New England journal of medicine. 2017, 377(6):562–572. doi:10.1056/NEJMra1608077
2. Alysandratos KD, Herriges MJ, Kotton DN: Epithelial Stem and Progenitor Cells in Lung Repair and Regeneration. Annual review of physiology. 2021, 83:529–550. doi:10.1146/annurev-physiol-041520-092904
3. Hewitt RJ, Lloyd CM: Regulation of immune responses by the airway epithelial cell landscape. Nature reviews Immunology. 2021, 21(6):347–362. doi:10.1038/s41577-020-00477-9
4. Zhou Y, Li P, Goodwin AJ, Cook JA, Halushka PV, Chang E *et al*: Exosomes from endothelial progenitor cells improve outcomes of the lipopolysaccharide-

- induced acute lung injury. *Critical care* (London, England). 2019, 23(1):44. doi:10.1186/s13054-019-2339-3
5. Bos LDJ, Ware LB: Acute respiratory distress syndrome: causes, pathophysiology, and phenotypes. *Lancet* (London, England). 2022, 400(10358):1145-1156. doi:10.1016/s0140-6736(22)01485-4
 6. Wang P, Luo R, Zhang M, Wang Y, Song T, Tao T *et al*: A cross-talk between epithelium and endothelium mediates human alveolar-capillary injury during SARS-CoV-2 infection. *Cell death & disease*. 2020, 11(12):1042. doi:10.1038/s41419-020-03252-9
 7. Peng F, Liao M, Qin R, Zhu S, Peng C, Fu L *et al*: Regulated cell death (RCD) in cancer: key pathways and targeted therapies. *Signal transduction and targeted therapy*. 2022, 7(1):286. doi:10.1038/s41392-022-01110-y
 8. Tang D, Kang R, Berghe TV, Vandenabeele P, Kroemer G: The molecular machinery of regulated cell death. *Cell research*. 2019, 29(5):347-364. doi:10.1038/s41422-019-0164-5
 9. Choi ME, Price DR, Ryter SW, Choi AMK: Necroptosis: a crucial pathogenic mediator of human disease. *JCI insight*. 2019, 4(15). doi:10.1172/jci.insight.128834
 10. Zhang X, Fan C, Zhang H, Zhao Q, Liu Y, Xu C *et al*: MLKL and FADD Are Critical for Suppressing Progressive Lymphoproliferative Disease and Activating the NLRP3 Inflammasome. *Cell reports*. 2016, 16(12):3247-3259. doi:10.1016/j.celrep.2016.06.103
 11. Galluzzi L, Kepp O, Chan FK, Kroemer G: Necroptosis: Mechanisms and Relevance to Disease. *Annual review of pathology*. 2017, 12:103-130. doi:10.1146/annurev-pathol-052016-100247
 12. Weinlich R, Oberst A, Beere HM, Green DR: Necroptosis in development, inflammation and disease. *Nature reviews Molecular cell biology*. 2017, 18(2):127-136. doi:10.1038/nrm.2016.149
 13. Gao W, Wang X, Zhou Y, Wang X, Yu Y: Autophagy, ferroptosis, pyroptosis, and necroptosis in tumor immunotherapy. *Signal transduction and targeted therapy*. 2022, 7(1):196. doi:10.1038/s41392-022-01046-3
 14. Bertheloot D, Latz E, Franklin BS: Necroptosis, pyroptosis and apoptosis: an intricate game of cell death. *Cellular & molecular immunology*. 2021, 18(5):1106-1121. doi:10.1038/s41423-020-00630-3
 15. Kearney CJ, Martin SJ: An Inflammatory Perspective on Necroptosis. *Molecular cell*. 2017, 65(6):965-973. doi:10.1016/j.molcel.2017.02.024
 16. Belavgeni A, Meyer C, Stumpf J, Hugo C, Linkermann A: Ferroptosis and Necroptosis in the Kidney. *Cell chemical biology*. 2020, 27(4):448-462. doi:10.1016/j.chembiol.2020.03.016
 17. Alphonse MP, Rubens JH, Ortines RV, Orlando NA, Patel AM, Dikeman D *et al*: Pan-caspase inhibition as a potential host-directed immunotherapy against MRSA and other bacterial skin infections. *Science translational medicine*. 2021, 13(601). doi:10.1126/scitranslmed.abe9887
 18. de Reuver R, Verdonck S, Dierick E, Nemegeer J, Hessmann E, Ahmad S *et al*:

- ADAR1 prevents autoinflammation by suppressing spontaneous ZBP1 activation. *Nature*. 2022, 607(7920):784–789. doi:10.1038/s41586-022-04974-w
19. Liu L, Li H, Hu D, Wang Y, Shao W, Zhong J *et al*: Insights into N6-methyladenosine and programmed cell death in cancer. *Molecular cancer*. 2022, 21(1):32. doi:10.1186/s12943-022-01508-w
 20. Tong X, Tang R, Xiao M, Xu J, Wang W, Zhang B *et al*: Targeting cell death pathways for cancer therapy: recent developments in necroptosis, pyroptosis, ferroptosis, and cuproptosis research. *Journal of hematology & oncology*. 2022, 15(1):174. doi:10.1186/s13045-022-01392-3
 21. Nozaki K, Li L, Miao EA: Innate Sensors Trigger Regulated Cell Death to Combat Intracellular Infection. *Annual review of immunology*. 2022, 40:469–498. doi:10.1146/annurev-immunol-101320-011235
 22. Pasparakis M, Vandenabeele P: Necroptosis and its role in inflammation. *Nature*. 2015, 517(7534):311–320. doi:10.1038/nature14191
 23. Zhang X, Ren Z, Xu W, Jiang Z: Necroptosis in atherosclerosis. *Clinica chimica acta; international journal of clinical chemistry*. 2022, 534:22–28. doi:10.1016/j.cca.2022.07.004
 24. Karunakaran D, Geoffrion M, Wei L, Gan W, Richards L, Shangari P *et al*: Targeting macrophage necroptosis for therapeutic and diagnostic interventions in atherosclerosis. *Science advances*. 2016, 2(7):e1600224. doi:10.1126/sciadv.1600224
 25. Schock SN, Chandra NV, Sun Y, Irie T, Kitagawa Y, Gotoh B *et al*: Induction of necroptotic cell death by viral activation of the RIG-I or STING pathway. *Cell death and differentiation*. 2017, 24(4):615–625. doi:10.1038/cdd.2016.153
 26. Brault M, Olsen TM, Martinez J, Stetson DB, Oberst A: Intracellular Nucleic Acid Sensing Triggers Necroptosis through Synergistic Type I IFN and TNF Signaling. *Journal of immunology (Baltimore, Md : 1950)*. 2018, 200(8):2748–2756. doi:10.4049/jimmunol.1701492
 27. Chen D, Tong J, Yang L, Wei L, Stolz DB, Yu J *et al*: PUMA amplifies necroptosis signaling by activating cytosolic DNA sensors. *Proceedings of the National Academy of Sciences of the United States of America*. 2018, 115(15):3930–3935. doi:10.1073/pnas.1717190115
 28. Wallach D, Kang TB, Dillon CP, Green DR: Programmed necrosis in inflammation: Toward identification of the effector molecules. *Science (New York, NY)*. 2016, 352(6281):aaf2154. doi:10.1126/science.aaf2154
 29. Zhang X, Wu J, Liu Q, Li X, Yang Y, Wu L *et al*: RIPK3-MLKL necroptotic signalling amplifies STING pathway and exacerbates lethal sepsis. *Clinical and translational medicine*. 2023, 13(7):e1334. doi:10.1002/ctm2.1334
 30. Wegner KW, Saleh D, Degterev A: Complex Pathologic Roles of RIPK1 and RIPK3: Moving Beyond Necroptosis. *Trends in pharmacological sciences*. 2017, 38(3):202–225. doi:10.1016/j.tips.2016.12.005
 31. Pereira M, Durso DF, Bryant CE, Kurt-Jones EA, Silverman N, Golenbock DT *et al*: The IRAK4 scaffold integrates TLR4-driven TRIF and MYD88 signaling pathways. *Cell reports*. 2022, 40(7):111225. doi:10.1016/j.celrep.2022.111225

32. Owen AM, Luan L, Burelbach KR, McBride MA, Stothers CL, Boykin OA *et al*: MyD88-dependent signaling drives toll-like receptor-induced trained immunity in macrophages. *Frontiers in immunology*. 2022, 13:1044662. doi:10.3389/fimmu.2022.1044662
33. Sun Y, Revach OY, Anderson S, Kessler EA, Wolfe CH, Jenney A *et al*: Targeting TBK1 to overcome resistance to cancer immunotherapy. *Nature*. 2023, 615(7950):158-167. doi:10.1038/s41586-023-05704-6
34. Hildebrand JM, Tanzer MC, Lucet IS, Young SN, Spall SK, Sharma P *et al*: Activation of the pseudokinase MLKL unleashes the four-helix bundle domain to induce membrane localization and necroptotic cell death. *Proceedings of the National Academy of Sciences of the United States of America*. 2014, 111(42):15072-15077. doi:10.1073/pnas.1408987111
35. Zhan Q, Jeon J, Li Y, Huang Y, Xiong J, Wang Q *et al*: CAMK2/CaMKII activates MLKL in short-term starvation to facilitate autophagic flux. *Autophagy*. 2022, 18(4):726-744. doi:10.1080/15548627.2021.1954348
36. Liu M, Lu J, Hu J, Chen Y, Deng X, Wang J *et al*: Sodium sulfite triggered hepatic apoptosis, necroptosis, and pyroptosis by inducing mitochondrial damage in mice and AML-12 cells. *Journal of hazardous materials*. 2024, 467:133719. doi:10.1016/j.jhazmat.2024.133719
37. Ciesielska A, Matyjek M, Kwiatkowska K: TLR4 and CD14 trafficking and its influence on LPS-induced pro-inflammatory signaling. *Cellular and molecular life sciences : CMLS*. 2021, 78(4):1233-1261. doi:10.1007/s00018-020-03656-y
38. Yu Z, Jiang N, Su W, Zhuo Y: Necroptosis: A Novel Pathway in Neuroinflammation. *Frontiers in pharmacology*. 2021, 12:701564. doi:10.3389/fphar.2021.701564
39. Li D, Wu M: Pattern recognition receptors in health and diseases. *Signal transduction and targeted therapy*. 2021, 6(1):291. doi:10.1038/s41392-021-00687-0
40. Hoebe K, Janssen EM, Kim SO, Alexopoulou L, Flavell RA, Han J *et al*: Upregulation of costimulatory molecules induced by lipopolysaccharide and double-stranded RNA occurs by Trif-dependent and Trif-independent pathways. *Nature immunology*. 2003, 4(12):1223-1229. doi:10.1038/ni1010
41. Shen H, Tesar BM, Walker WE, Goldstein DR: Dual signaling of MyD88 and TRIF is critical for maximal TLR4-induced dendritic cell maturation. *Journal of immunology (Baltimore, Md : 1950)*. 2008, 181(3):1849-1858. doi:10.4049/jimmunol.181.3.1849
42. Kaisho T, Takeuchi O, Kawai T, Hoshino K, Akira S: Endotoxin-induced maturation of MyD88-deficient dendritic cells. *Journal of immunology (Baltimore, Md : 1950)*. 2001, 166(9):5688-5694. doi:10.4049/jimmunol.166.9.5688
43. Pan X, Niu X, Li Y, Yao Y, Han L: Preventive Mechanism of Lycopene on Intestinal Toxicity Caused by Cyclophosphamide Chemotherapy in Mice by Regulating TLR4-MyD88/TRIF-TRAF6 Signaling Pathway and Gut-Liver Axis. *Nutrients*. 2022, 14(21). doi:10.3390/nu14214467
44. Vanaja SK, Russo AJ, Behl B, Banerjee I, Yankova M, Deshmukh SD *et al*:

- Bacterial Outer Membrane Vesicles Mediate Cytosolic Localization of LPS and Caspase-11 Activation. *Cell*. 2016, 165(5):1106–1119. doi:10.1016/j.cell.2016.04.015
45. Jiang Z, Georgel P, Du X, Shamel L, Sovath S, Mudd S *et al*: CD14 is required for MyD88-independent LPS signaling. *Nature immunology*. 2005, 6(6):565–570. doi:10.1038/ni1207
46. Zanoni I, Ostuni R, Marek LR, Barresi S, Barbalat R, Barton GM *et al*: CD14 controls the LPS-induced endocytosis of Toll-like receptor 4. *Cell*. 2011, 147(4):868–880. doi:10.1016/j.cell.2011.09.051
47. Reynoso M, Hobbs S, Kolb AL, Matheny RW, Jr., Roberts BM: MyD88 and not TRIF knockout is sufficient to abolish LPS-induced inflammatory responses in bone-derived macrophages. *FEBS letters*. 2023, 597(9):1225–1232. doi:10.1002/1873-3468.14616
48. Gong Y, Fan Z, Luo G, Yang C, Huang Q, Fan K *et al*: The role of necroptosis in cancer biology and therapy. *Molecular cancer*. 2019, 18(1):100. doi:10.1186/s12943-019-1029-8
49. Rickard JA, O'Donnell JA, Evans JM, Lalaoui N, Poh AR, Rogers T *et al*: RIPK1 regulates RIPK3-MLKL-driven systemic inflammation and emergency hematopoiesis. *Cell*. 2014, 157(5):1175–1188. doi:10.1016/j.cell.2014.04.019
50. Degtarev A, Ofengeim D, Yuan J: Targeting RIPK1 for the treatment of human diseases. *Proceedings of the National Academy of Sciences of the United States of America*. 2019, 116(20):9714–9722. doi:10.1073/pnas.1901179116
51. Newton K: RIPK1 and RIPK3: critical regulators of inflammation and cell death. *Trends in cell biology*. 2015, 25(6):347–353. doi:10.1016/j.tcb.2015.01.001
52. Haddadzadegan S, Dorkoosh F, Bernkop-Schnürch A: Oral delivery of therapeutic peptides and proteins: Technology landscape of lipid-based nanocarriers. *Advanced drug delivery reviews*. 2022, 182:114097. doi:10.1016/j.addr.2021.114097
53. Li J, Zheng H, Xu EY, Moehwald M, Chen L, Zhang X *et al*: Inhalable PLGA microspheres: Tunable lung retention and systemic exposure via polyethylene glycol modification. *Acta biomaterialia*. 2021, 123:325–334. doi:10.1016/j.actbio.2020.12.061
54. Huang SQ, Zhang HM, Zhang YC, Wang LY, Zhang ZR, Zhang L: Comparison of two methods for tumour-targeting peptide modification of liposomes. *Acta pharmacologica Sinica*. 2023, 44(4):832–840. doi:10.1038/s41401-022-01011-4
55. Sauler M, Bazan IS, Lee PJ: Cell Death in the Lung: The Apoptosis-Necroptosis Axis. *Annual review of physiology*. 2019, 81:375–402. doi:10.1146/annurev-physiol-020518-114320
56. Song X, Wan Z, Chen T, Fu Y, Jiang K, Yi X *et al*: Development of a multi-target peptide for potentiating chemotherapy by modulating tumor microenvironment. *Biomaterials*. 2016, 108:44–56. doi:10.1016/j.biomaterials.2016.09.001
57. Hamilton GL, Chen H, Deshmukh G, Eigenbrot C, Fong R, Johnson A *et al*: Potent and selective inhibitors of receptor-interacting protein kinase 1 that lack

- an aromatic back pocket group. *Bioorganic & medicinal chemistry letters*. 2019, 29(12):1497–1501. doi:10.1016/j.bmcl.2019.04.014
58. Meng Y, Davies KA, Fitzgibbon C, Young SN, Garnish SE, Horne CR *et al*: Human RIPK3 maintains MLKL in an inactive conformation prior to cell death by necroptosis. *Nature communications*. 2021, 12(1):6783. doi:10.1038/s41467-021-27032-x
59. Berman HM, Westbrook J, Feng Z, Gilliland G, Bhat TN, Weissig H *et al*: The Protein Data Bank. *Nucleic acids research*. 2000, 28(1):235–242. doi:10.1093/nar/28.1.235
60. Bernstein FC, Koetzle TF, Williams GJ, Meyer EF, Jr., Brice MD, Rodgers JR *et al*: The Protein Data Bank: a computer-based archival file for macromolecular structures. *Journal of molecular biology*. 1977, 112(3):535–542. doi:10.1016/s0022-2836(77)80200-3
61. Burley SK, Bhikadiya C, Bi C, Bittrich S, Chao H, Chen L *et al*: RCSB Protein Data Bank (RCSB.org): delivery of experimentally-determined PDB structures alongside one million computed structure models of proteins from artificial intelligence/machine learning. *Nucleic acids research*. 2023, 51(D1):D488–d508. doi:10.1093/nar/gkac1077
62. Desta IT, Porter KA, Xia B, Kozakov D, Vajda S: Performance and Its Limits in Rigid Body Protein-Protein Docking. *Structure (London, England : 1993)*. 2020, 28(9):1071–1081. e1073. doi:10.1016/j.str.2020.06.006
63. Vajda S, Yueh C, Beglov D, Bohnuud T, Mottarella SE, Xia B *et al*: New additions to the ClusPro server motivated by CAPRI. *Proteins*. 2017, 85(3):435–444. doi:10.1002/prot.25219
64. Kozakov D, Hall DR, Xia B, Porter KA, Padhorny D, Yueh C *et al*: The ClusPro web server for protein-protein docking. *Nature protocols*. 2017, 12(2):255–278. doi:10.1038/nprot.2016.169
65. Kozakov D, Beglov D, Bohnuud T, Mottarella SE, Xia B, Hall DR *et al*: How good is automated protein docking? *Proteins*. 2013, 81(12):2159–2166. doi:10.1002/prot.24403

A

B

C

A

B

C

D

E

F

G